# Communication-Efficient Adaptive Federated Bi-level Optimization with Data and System Heterogeneity

## Abstract

Bilevel optimization is a popular nested optimization model in machine learning. Federated bilevel optimization, which extends bilevel optimization to the Federated Learning setting, faces challenges such as complex nested sub-loops, high communication overhead, and a lack of adaptive mechanisms. To address these issues, this paper proposes an Adaptive Single-loop Federated Bilevel Optimization algorithm (ASFBO) in the presence of both data heterogeneity (Non-IID client data) and system heterogeneity (partial client participation per round and varying numbers of local iterations). By replacing nested sub-iterations with a single-loop architecture, ASFBO significantly reduces communication frequency and computational costs. It employs multiple adaptive learning rate variables to dynamically adjust the step sizes of upper-level variable updates, thereby speeding up the algorithm's convergence. Furthermore, a locally accelerated version of the algorithm (LA-ASFBO) that incorporates momentum-based variance reduction techniques is proposed to mitigate hyper-gradient estimation bias across distributed nodes effectively. Theoretical analysis shows that, under the classic setting of a non-convex upper-level and strongly convex lower-level, ASFBO and LA-ASFBO achieve convergence to an $\epsilon$-stationary point with only $\tilde{\mathcal{O}}(\epsilon^{-2})$ sample complexity and $\tilde{\mathcal{O}}(\epsilon^{-1})$ communication complexity. Experiments on federated hyper-representation learning tasks demonstrate the superiority of the proposed algorithm.

## 1 Introduction

In recent years, emerging directions in the field of machine learning—such as meta-learning, automatic hyperparameter optimization, reinforcement learning, and adversarial learning—have made significant strides by virtue of their shared nested optimization structure. These problems, commonly referred to as bi-level optimization (BLO) Bracken & McGill (1973), comprise an upper-level problem whose objective and variables depend on the solution of a lower-level auxiliary problem. Theoretical advances in stochastic bi-level optimization (SBO) Christiansen et al. (2001) have spurred interest in solving BLOs at scale in distributed systems. More recently, the advent of federated learning has given rise to federated bi-level optimization (FBO) Tarzanagh et al. (2022), which extends distributed and federated optimization paradigms to address tasks such as federated min-max optimization Sharma et al. (2022), federated meta-learning Chen et al. (2018), and federated hyper-parameter optimization Khodak et al. (2021).

Deterministic bi-level optimization has evolved from its early formulation as a leader-follower problem Bracken & McGill (1973) to three practical hyper-gradient estimation strategies that dominate contemporary research and applications. The first is *iterative differentiation* Franceschi et al. (2018), which explicitly unrolls the lower-level solver, treats it as a differentiable computation graph, and back-propagates through the iterations to obtain an approximate hyper-gradient without forming or inverting Hessian matrices. The second is *approximate implicit differentiation* Ghadimi & Wang (2018), which postpones differentiation until the lower-level variables have nearly converged and then applies iterative linear solvers—such as conjugate gradient or Neumann-series truncation—to approximate the inverse Hessian required by the implicit function theorem. The third is the *fully single-loop* paradigm Li et al. (2022a), which reformulates the hyper-gradient computation as a jointly optimized linear-system objective so that the upper-level variables, lower-level variables, and an auxiliary vector are updated in a single pass, markedly reducing per-iteration

Table 1: Comparison of federated bi-level optimization algorithms

| Algorithm | VR | MU | AS | SH | DH | PP | SC | CC |
|---|---|---|---|---|---|---|---|---|
| **FBO-AggITD** Xiao & Ji (2023) | | | | | ✓ | | $\epsilon^{-2}$ | $\epsilon^{-2}$ |
| **LocalBSGM** Gao (2022) | | ✓ | | | | | $\epsilon^{-2}n^{-1}$ | $\epsilon^{-1.5}$ |
| **LocalBSGVR** Gao (2022) | ✓ | ✓ | | | | | $\epsilon^{-1.5}n^{-1}$ | $\epsilon^{-1}$ |
| **FedNest** Tarzanagh et al. (2022) | ✓ | | | | ✓ | | $\epsilon^{-2}$ | $\epsilon^{-2}$ |
| **FedMBO** Huang et al. (2023) | | | | | ✓ | ✓ | $\epsilon^{-2}P^{-1}$ | $\epsilon^{-2}$ |
| **SimFBO** Yang et al. (2023) | | | | | ✓ | ✓ | $\epsilon^{-2}P^{-1}$ | $\epsilon^{-1}$ |
| **ShroFBO** Yang et al. (2023) | | | | ✓ | ✓ | ✓ | $\epsilon^{-2}P^{-1}$ | $\epsilon^{-1}$ |
| **FedBiOAcc** Li et al. (2023) | ✓ | ✓ | | | ✓ | | $\epsilon^{-1.5}n^{-1}$ | $\epsilon^{-1}$ |
| **AdaFBiO** Huang (2022) | ✓ | ✓ | ✓ | | ✓ | | $\epsilon^{-1.5}$ | $\epsilon^{-1}$ |
| **ASFBO (this paper)** | | ✓ | ✓ | ✓ | ✓ | ✓ | $\epsilon^{-2}P^{-1}$ | $\epsilon^{-1}$ |
| **LA-ASFBO (this paper)** | ✓ | ✓ | ✓ | ✓ | ✓ | ✓ | $\epsilon^{-2}P^{-1}$ | $\epsilon^{-1}$ |

*Abbreviations:* VR = Variance Reduction; MU = Momentum Update; AS = Adaptive Stepsize; SH = System Heterogeneity; DH = Data Heterogeneity; PP = Partial Participation; SC = Sample Complexity; CC = Communication Complexity.

computational and memory costs while retaining provable convergence guarantees. These three methods provide a flexible toolkit that supports the solution to bi-level optimization problems in federated scenarios and serves as the basis for methodology in federated meta-learning Liu et al. (2023), federated hyperparameter optimization Nakka et al. (2024), and other federated nested learning tasks.

## 1.1 Main Contributions

In this paper, we tackle the challenges of data heterogeneity and system heterogeneity in federated bi-level optimization by proposing a communication-efficient single-loop algorithm enhanced with momentum updates, adaptive learning rates, and personalized privacy protection. The comparison with existing works are shown in Table 1. Our main contributions are as follows:

- **ASFBO:** We introduce ASFBO, a single-loop federated bi-level optimization algorithm that incorporates momentum smoothing in local updates to mitigate the variance of stochastic hyper-gradient estimates. On the server side, we employ an adaptive learning-rate strategy combined with client re-weighting to suppress convergence errors induced by data and system heterogeneity.

- **LA-ASFBO:** Building on ASFBO, we develop a locally accelerated variant LA-ASFBO by integrating momentum-based variance reduction, which further improves computational efficiency and significantly enhances convergence stability under non-i.i.d. data distributions.

- **Theoretical Analysis:** We prove that both ASFBO and LA-ASFBO achieve an $\epsilon$-accurate stationary point for the nonconvex-strongly-convex bi-level problem within $\mathcal{O}(\epsilon^{-1})$ communication rounds, while their computational complexity scales as $\mathcal{O}(P^{-1}\epsilon^{-2})$, where $P$ denotes the total number of clients.

- **Empirical Validation:** We demonstrate the effectiveness and stability of ASFBO and LA-ASFBO on real-world federated hyper-representation learning tasks, showing superior convergence and communication efficiency compared to existing methods.

Our work provides both theoretical foundations and practical algorithmic designs for efficient bi-level optimization in heterogeneous federated environments, and lays the groundwork for integrating more advanced adaptive techniques and privacy-preserving mechanisms in future federated bi-level frameworks.

## 2 Related Works

**Bilevel optimization:** Bilevel optimization, first formulated by Bracken and McGill to model leader-follower decision processes Bracken & McGill (1973), has evolved from KKT-based single-level reformulations Hansen et al. (1992); Sinha et al. (2017) to three main solution paradigms: (1) implicit-function (IF) methods, which leverage the implicit function theorem Krantz & Parks (2002) and approximate the inverse Hessian via techniques such as truncated conjugate gradient Shaban et al. (2019), low-rank Woodbury updates Singh & Alistarh (2020), or Neumann series expansion Ghadimi & Wang (2018); (2) gradient-unrolling (GU) approaches, which treat the lower-level solver as a differentiable computation graph and apply forward or reverse-mode automatic differentiation with truncation to balance memory and accuracy Domke (2012); Baydin et al. (2018); Franceschi et al. (2017); Maclaurin et al. (2015); Luketina et al. (2016); (3) value-function (VF) schemes, which convert the lower-level optimum into upper-level constraints or penalty terms and solve the resulting single-level problem via barrier or augmented Lagrangian methods Liu et al. (2021); Shen & Chen (2023); Sow et al. (2022). More recently, stochastic bilevel optimization has been studied under two complementary frameworks: approximate implicit differentiation (AID), exemplified by BSA Ghadimi & Wang (2018) and its extensions TTSA Hong et al. (2020), STABLE Chen et al. (2022), MSTSA/SUSTAIN Khanduri et al. (2021), and MRBO Yang et al. (2021), which approximate Hessian inverses within a single-loop update; and iterative differentiation (ITD), which directly unrolls stochastic lower-level iterations and applies truncated forward/backward differentiation to obtain hypergradients with provable convergence guarantees Franceschi et al. (2018); Shaban et al. (2019).

**Federated bilevel optimization:** Federated bi-level optimization extends classical bi-level frameworks to federated settings, where modern tasks such as federated meta-learning and hyperparameter tuning naturally exhibit upper- and lower-level coupling Chen et al. (2018); Khodak et al. (2021). Early works under homogeneous (IID) assumptions—e.g., LocalBSGM Gao (2022) and FedBiO Li et al. (2022b)—adopt a fully single-loop approach Li et al. (2022a) with local hypergradient estimation and achieve asymptotic (even accelerated linear) convergence; their momentum-enhanced variants LocalBSGVR and FedBiOAcc further reduce variance and speed up convergence. In non-IID scenarios, communication-efficient AID-based FedNest employs FedSVRG in the inner loop and global hypergradient aggregation to attain the first non-asymptotic rates Tarzanagh et al. (2022), while AdaFBiO Huang (2022) integrates momentum variance reduction entirely on-device to cut synchronization costs at the expense of linear acceleration under heterogeneity. ITD-based FBO-AggITD unrolls a fixed number of local iterations and aggregates iterative-differentiation estimates in a single round Xiao & Ji (2023), and ShroFBO builds on a communication-efficient single-loop paradigm with client re-weighting to recover linear rates in heterogeneous environments Yang et al. (2023). Beyond centralized federated architectures, decentralized bi-level optimization (DBO) methods—such as AiPOD's stochastic primal-dual scheme Lu et al. (2022), gossip-based Gossip-BiO Yang et al. (2022), SLDBO's two-matrix-vector-multiply single-loop algorithm Dong et al. (2023), and the SPARKLE primal-dual framework Zhu et al. (2024)—as well as the recent decentralized algorithms by Chen et al. Chen et al. (2024), further broaden the applicability of federated bi-level techniques to heterogeneous, peer-to-peer networks.

## 3 Preliminary

### 3.1 Federated Bilevel Optimization

Formally, the federated bilevel optimization problem can be expressed as

$$\min_{x \in \mathbb{R}^p} \Phi(x) = F\big(x, y^*(x)\big) := \sum_{i=1}^{n} p_i f_i\big(x, y^*(x)\big) = \sum_{i=1}^{n} p_i \mathbb{E}_\xi\Big[f_i\big(x, y^*(x); \xi_i\big)\Big], \text{(upper)} \tag{1a}$$

$$\text{s.t.} \quad y^*(x) = \arg\min_{y \in \mathbb{R}^q} G(x, y) := \sum_{i=1}^{n} p_i g_i(x, y) = \sum_{i=1}^{n} p_i \mathbb{E}_\zeta\big[g_i(x, y; \zeta_i)\big], \text{(lower)} \tag{1b}$$

where, $n$ is the number of distributed devices (e.g. smartphones or edge nodes) participating in training, and $p_i \geq 0$ with $\sum_{i=1}^{n} p_i = 1$ denotes the relative importance of client $i$ (e.g., proportional to its data volume). The upper level variable $x \in \mathbb{R}^p$ typically represents global model parameters or hyperparameters, while

the lower level variable $y \in \mathbb{R}^q$ denotes auxiliary local parameters conditioned on $x$. The mapping $y^*(x)$ is defined by the optimal solution of the problem of the lower level in (1b). The function $\Phi(x) = F(x, y^*(x))$ measures the global performance (for example, aggregate loss) and depends on both $x$ and the solution at the lower level $y^*(x)$, while $G(x, y)$ at the lower level aggregates the weighted local objectives. Specifically, $f_i(x, y^*(x)) = \mathbb{E}_\xi[f_i(x, y^*(x); \xi_i)]$ and $g_i(x, y) = \mathbb{E}_\zeta[g_i(x, y; \zeta_i)]$ denote the expected local loss functions at client $i$, smoothed over the random samples $\xi_i$ and $\zeta_i$, respectively.

Efficiently solving the federated bilevel optimization problem (1) entails several key challenges, including the computation of the federated hyper-gradient $\nabla \Phi(x)$, the heterogeneity of the data and system levels, and the inherent complexity of the nested optimization structure. By applying the implicit function theorem Griewank & Walther (2008) under Assumptions 1–5, the hyper-gradient admits the form:

$$\nabla \Phi(x) = \sum_{i=1}^n p_i \nabla_x f_i(x, y^*) - \nabla_{xy}^2 G(x, y^*) \left[\nabla_{yy}^2 G(x, y^*)\right]^{-1} \sum_{i=1}^n p_i \nabla_y f_i(x, y^*), \tag{2}$$

where $\nabla_x f_i(x, y^*)$ and $\nabla_y f_i(x, y^*)$ denote the local gradients of $f_i$ and $\nabla_{xy}^2 G$, $\nabla_{yy}^2 G$ are the mixed Jacobian and Hessian of the aggregated lower-level objective Griewank & Walther (2008).

## 3.2 Federated Hypergradient Estimation

The direct computation of the federated hypergradient $\nabla \Phi(x)$ in (2) is intractable due to the implicit dependence of $y^*(x)$ on $x$ and the heterogeneity of the client side. Instead, we introduce a surrogate hypergradient:

$$\nabla \bar{F}(x, y, z) = \nabla_x F(x, y) - \nabla_{xy}^2 G(x, y) z, \tag{3}$$

where $y$ approximates the lower-level solution $y^*(x)$, and the auxiliary vector $z \in \mathbb{R}^{d_y}$ is defined to satisfy

$$z = \left[\nabla_{yy}^2 G(x, y)\right]^{-1} \nabla_y F(x, y), \tag{4}$$

thereby approximating the Hessian-inverse term in the true hypergradient.

To estimate $y$ and $z$ in a federated manner, we observe that solving the global linear system

$$\nabla_{yy}^2 G(x, y) z = \nabla_y F(x, y) \tag{5}$$

is equivalent to minimizing the quadratic objective

$$R(x, y, z) = \tfrac{1}{2} z^\top \nabla_{yy}^2 G(x, y) z - z^\top \nabla_y F(x, y), \tag{6}$$

since

$$\nabla_z R(x, y, z) = \nabla_{yy}^2 G(x, y) z - \nabla_y F(x, y). \tag{7}$$

Noting that

$$G(x, y) = \sum_{i=1}^n p_i \, g_i(x, y), \tag{8a}$$

$$F(x, y) = \sum_{i=1}^n p_i \, f_i(x, y), \tag{8b}$$

we decompose the global auxiliary objective into client-wise components

$$R(x, y, z) = \sum_{i=1}^n p_i \, R_i(x, y, z), \tag{9}$$

where

$$R_i(x,y,z) = \tfrac{1}{2}\, z^\top \nabla_{yy}^2 g_i(x,y)\, z - z^\top \nabla_y f_i(x,y), \tag{10}$$

and

$$\nabla_z R_i(x,y,z) = \nabla_{yy}^2 g_i(x,y)\, z - \nabla_y f_i(x,y). \tag{11}$$

Each client $i$ locally computes $\nabla_{yy}^2 g_i$ and $\nabla_y f_i$, minimizes $R_i$ to update $z$, and sends $\nabla_z R_i$ to the server, which aggregates to refine the global $z$.

Similarly, the global lower-level gradient

$$\nabla_y G(x,y) = \sum_{i=1}^{n} p_i\, \nabla_y g_i(x,y) \tag{12}$$

is obtained by aggregating local $\nabla_y g_i$ updates to drive $y$ toward $y^*(x)$.

Finally, with federated estimates $y$ and $z$, the surrogate hypergradient also decomposes as

$$\nabla \bar{F}(x,y,z) = \sum_{i=1}^{n} p_i\, \nabla \bar{f}_i(x,y,z), \tag{13}$$

where

$$\nabla \bar{f}_i(x,y,z) = \nabla_x f_i(x,y) - \nabla_{xy}^2 g_i(x,y)\, z. \tag{14}$$

This framework avoids explicit Hessian inversion, reduces computational complexity, and leverages distributed client computations to approximate the true federated hypergradient in a communication-efficient manner.

## 4   ASFBO: An Adaptive Single-Loop Federated Bilevel Optimization Framework

ASFBO (and its variance-reduced variant LA-ASFBO) solves the federated bilevel problem in a *single communication loop*: in every round the server broadcasts the current global variables, clients run a small number of local iterations with momentum, aggregate their updates, and the server performs a single global update. This section details the four key components of the framework.

### 4.1   Communication Rounds and Client Initialisation

Let $n$ be the total number of clients and $T$ the total number of communication rounds indexed by $t = 0, \ldots, T-1$. In round $t$ only a subset $C^{(t)} \subseteq \{1, \ldots, n\}$ is active; we assume $C^{(t)}$ is sampled uniformly without replacement. The server broadcasts the global variables $\{x^{(t)}, y^{(t)}, z^{(t)}\}$ to every $i \in C^{(t)}$, and each active client initialises its local copies by

$$x_i^{(t,0)} = x^{(t)}, \quad y_i^{(t,0)} = y^{(t)}, \quad z_i^{(t,0)} = z^{(t)}. \tag{15}$$

A common initial state mitigates bias caused by Non-IID data and facilitates global convergence.

Clients then bootstrap their momentum buffers using the current stochastic gradients:

$$u_i^{(t,0)} = \nabla \bar{f}_i(x_i^{(t,0)}, y_i^{(t,0)}, z_i^{(t,0)}; \bar{\xi}_i^{(t,0)}), \tag{16a}$$

$$v_i^{(t,0)} = \nabla_y g_i(x_i^{(t,0)}, y_i^{(t,0)}; \zeta_i^{(t,0)}), \tag{16b}$$

$$w_i^{(t,0)} = \nabla_z R_i(x_i^{(t,0)}, y_i^{(t,0)}, z_i^{(t,0)}; \psi_i^{(t,0)}). \tag{16c}$$

The local hyper-gradient surrogate that approximates the true federated hyper-gradient is

$$\nabla \bar{f}_i(x_i^{(t,k)}, y_i^{(t,k)}, z_i^{(t,k)}; \bar{\xi}_i) = \nabla_x f_i(x_i^{(t,k)}, y_i^{(t,k)}; \xi_i^{(t,k)}) - \nabla_{xy}^2 g_i(x_i^{(t,k)}, y_i^{(t,k)}; \zeta_i^{(t,k)})\, z_i^{(t,k)}. \tag{17}$$

For the auxiliary variable, we need the gradient of the local linear-system objective:

$$\nabla_z R_i(x_i^{(t,k)}, y_i^{(t,k)}, z_i^{(t,k)}; \psi_i^{(t,k)}) = \nabla_{yy}^2 g_i(x_i^{(t,k)}, y_i^{(t,k)}; \zeta_i^{(t,k)})\, z_i^{(t,k)} - \nabla_y f_i(x_i^{(t,k)}, y_i^{(t,k)}; \xi_i^{(t,k)}). \tag{18}$$

---

**Algorithm 1** ASFBO and LA-ASFBO

---

**Input**: initial global variables $\{x^{(0)}, y^{(0)}, z^{(0)}\}$; total clients $n$; total communication rounds $T$

**Parameter**: active-client set per round $C^{(t)}$; local iteration counts $\tau_i^{(t)}$; tuning parameters $\{\beta, \eta_x, \eta_y, \eta_z, \gamma_x, \gamma_y, \gamma_z\}$

**Output**: optimized global variables $\{x^{(T)}, y^{(T)}\}$

1: Initialise adaptive learning-rate factors $\{a^{(0)}\!=\!0,\, b^{(0)}\!=\!0,\, c^{(0)}\!=\!0\}$
2: **for** $t = 0$ **to** $T-1$ **do**
3:   Server broadcasts $\{x^{(t)}, y^{(t)}, z^{(t)}\}$ to every active client
4:   **for** $i \in C^{(t)}$ **(in parallel) do**
5:     Initialise local variables $\{x_i^{(t,0)}, y_i^{(t,0)}, z_i^{(t,0)}\}$ via (15)
6:     Initialise momenta $\{u_i^{(t,0)}, v_i^{(t,0)}, w_i^{(t,0)}\}$ via (16)
7:     **for** $k = 0$ **to** $\tau_i^{(t)} - 1$ **do**
8:       Update local variables using (19)
9:       Update momenta with basic momentum (20) (**ASFBO**)

        **or** momentum variance reduction (21) (**LA-ASFBO**)
10:     **end for**
11:     Aggregate local gradients to obtain $\{q_{x,i}^{(t)}, q_{y,i}^{(t)}, q_{z,i}^{(t)}\}$ via (22)
12:     Send aggregated gradients to the server
13:   **end for**
14:   Server aggregates global gradients $\{h_x^{(t)}, h_y^{(t)}, h_z^{(t)}\}$ via (26)
15:   Update adaptive learning rates via (27) and (28)
16:   Update global variables $\{x^{(t+1)}, y^{(t+1)}, z^{(t+1)}\}$ via (29)
17: **end for**

---

## 4.2 Client-Side Local Update and Aggregation

Each active client performs $\tau_i^{(t)}$ local iterations. At iteration $k$, it simultaneously updates all three variables using momentum directions:

$$x_i^{(t,k+1)} = x_i^{(t,k)} - \eta_x\, u_i^{(t,k)}, \tag{19a}$$

$$y_i^{(t,k+1)} = y_i^{(t,k)} - \eta_y\, v_i^{(t,k)}, \tag{19b}$$

$$z_i^{(t,k+1)} = z_i^{(t,k)} - \eta_z\, w_i^{(t,k)}. \tag{19c}$$

This single-loop design avoids nested solves and reduces communication frequency while controlling client drift through appropriate choices of $\tau_i^{(t)}$.

**ASFBO (basic momentum).** Momentum buffers are updated via an exponential moving average:

$$u_i^{(t,k+1)} = \beta\, \nabla \bar{f}_i(*^{(t,k+1)}) + (1-\beta)\, u_i^{(t,k)}, \tag{20a}$$

$$v_i^{(t,k+1)} = \beta\, \nabla_y g_i(*^{(t,k+1)}) + (1-\beta)\, v_i^{(t,k)}, \tag{20b}$$

$$w_i^{(t,k+1)} = \beta\, \nabla_z R_i(*^{(t,k+1)}) + (1-\beta)\, w_i^{(t,k)}, \tag{20c}$$

where $*$ denotes the related variable.

**LA-ASFBO (STORM variance reduction).** To suppress gradient noise the STORM rule is employed:

$$u_i^{(t,k+1)} = \nabla \bar{f}_i(*^{(t,k+1)}) + (1-\beta)\big(u_i^{(t,k)} - \nabla \bar{f}_i(*^{(t,k)})\big), \tag{21a}$$

$$v_i^{(t,k+1)} = \nabla_y g_i(*^{(t,k+1)}) + (1-\beta)\big(v_i^{(t,k)} - \nabla_y g_i(*^{(t,k)})\big), \tag{21b}$$

$$w_i^{(t,k+1)} = \nabla_z R_i(*^{(t,k+1)}) + (1-\beta)\big(w_i^{(t,k)} - \nabla_z R_i(*^{(t,k)})\big). \tag{21c}$$

STORM leverages gradient differences computed with the *same* mini-batch to achieve variance reduction and faster convergence under heterogeneous data.

After $\tau_i^{(t)}$ local iterations, client $i$ compresses its information into three aggregated gradients:

$$q_{x,i}^{(t)} = \sum_{k=0}^{\tau_i^{(t)}-1} u_i^{(t,k)} = \sum_{k=0}^{\tau_i^{(t)}-1} a_i^{(t,k)} \nabla \bar{f}_i(\cdot), \tag{22a}$$

$$q_{y,i}^{(t)} = \sum_{k=0}^{\tau_i^{(t)}-1} v_i^{(t,k)} = \sum_{k=0}^{\tau_i^{(t)}-1} a_i^{(t,k)} \nabla_y g_i(\cdot), \tag{22b}$$

$$q_{z,i}^{(t)} = \sum_{k=0}^{\tau_i^{(t)}-1} w_i^{(t,k)} = \sum_{k=0}^{\tau_i^{(t)}-1} a_i^{(t,k)} \nabla_z R_i(\cdot). \tag{22c}$$

Here $a_i^{(t,k)}$ encodes the effective weight of each stochastic gradient under the chosen momentum rule. The aggregated triplet $\{q_{x,i}^{(t)}, q_{y,i}^{(t)}, q_{z,i}^{(t)}\}$ is sent to the server, which averages them to update the global variables and adaptive learning rates. This completes one communication round of ASFBO/LA-ASFBO.

### 4.3 Server-Side Global Aggregation and Update

The server completes each communication round by aggregating the local gradients, adapting the learning rates, and updating the global variables.

**Global Gradient Aggregation.** Upon receiving $\{q_{x,i}^{(t)}, q_{y,i}^{(t)}, q_{z,i}^{(t)}\}_{i\in C^{(t)}}$ from the active clients, a naive aggregation uses the effective weights $\widetilde{p}_i = \frac{n}{|C^{(t)}|} p_i$:

$$q_x^{(t)} = \sum_{i\in C^{(t)}} \widetilde{p}_i \, q_{x,i}^{(t)}, \quad q_y^{(t)} = \sum_{i\in C^{(t)}} \widetilde{p}_i \, q_{y,i}^{(t)}, \quad q_z^{(t)} = \sum_{i\in C^{(t)}} \widetilde{p}_i \, q_{z,i}^{(t)}. \tag{23}$$

However, heterogeneous local iteration counts $\tau_i^{(t)}$ bias these sums. Following Wang et al. (2020), rewriting (23) as

$$q_x^{(t)} = \rho^{(t)} \sum_{i=1}^{n} w_i^{(t)} h_{x,i}^{(t)}, \quad q_y^{(t)} = \rho^{(t)} \sum_{i=1}^{n} w_i^{(t)} h_{y,i}^{(t)}, \quad q_z^{(t)} = \rho^{(t)} \sum_{i=1}^{n} w_i^{(t)} h_{z,i}^{(t)}, \tag{24}$$

with

$$\rho^{(t)} := \sum_j p_j \|a_j^{(t)}\|_1, \quad w_i^{(t)} := \frac{p_i \|a_i^{(t)}\|_1}{\rho^{(t)}}. \tag{25}$$

reveals the dependence on $\tau_i^{(t)}$ via $\|a_i^{(t)}\|_1$. ASFBO therefore aggregates the *reweighted* gradients

$$h_x^{(t)} = \sum_{i\in C^{(t)}} \widetilde{p}_i \frac{q_{x,i}^{(t)}}{\|a_i^{(t)}\|_1}, \quad h_y^{(t)} = \sum_{i\in C^{(t)}} \widetilde{p}_i \frac{q_{y,i}^{(t)}}{\|a_i^{(t)}\|_1}, \quad h_z^{(t)} = \sum_{i\in C^{(t)}} \widetilde{p}_i \frac{q_{z,i}^{(t)}}{\|a_i^{(t)}\|_1}, \tag{26}$$

thereby neutralizing the iteration-count bias.

**Adaptive Learning Rates.** Following AdaGrad-Norm Ward et al. (2020), the server maintains running norms

$$s_x^{(t+1)} = \varrho^{(t)} s_x^{(t)} + (1 - \varrho^{(t)}) \|h_x^{(t)}\|_2, \tag{27a}$$

$$s_y^{(t+1)} = \varrho^{(t)} s_y^{(t)} + (1 - \varrho^{(t)}) \|h_y^{(t)}\|_2, \tag{27b}$$

$$s_z^{(t+1)} = \varrho^{(t)} s_z^{(t)} + (1 - \varrho^{(t)}) \|h_z^{(t)}\|_2, \tag{27c}$$

and computes step sizes

$$\gamma_x^{(t)} = \frac{\gamma_x}{s_x^{(t+1)} + \rho}, \quad \gamma_y^{(t)} = \frac{\gamma_y}{s_y^{(t+1)} + \rho}, \quad \gamma_z^{(t)} = \frac{\gamma_z}{s_z^{(t+1)} + \rho}. \tag{28}$$

**Global Variable Update.** Finally, scaled gradients are used to update the global variables:

$$x^{(t+1)} = x^{(t)} - \rho^{(t)}\gamma_x^{(t)}h_x^{(t)}, \quad y^{(t+1)} = y^{(t)} - \rho^{(t)}\gamma_y^{(t)}h_y^{(t)}, \quad z^{(t+1)} = \mathcal{P}_r\left(z^{(t)} - \rho^{(t)}\gamma_z^{(t)}h_z^{(t)}\right), \tag{29}$$

where the projection

$$\mathcal{P}_r(z) = \begin{cases} z, & \|z\| \leq r, \\ \frac{r}{\|z\|}z, & \|z\| > r, \end{cases} \tag{30}$$

with $r = L_f/\mu_g$ keeps $z$ bounded, ensuring the smoothness assumptions required for convergence.

## 5 Theoretical Analysis

### 5.1 Definitions and Assumptions

We make the following standard definitions and assumptions for the outer- and inner-level objective functions, as also adopted in stochastic bilevel optimization Ji et al. (2021) and in federated bilevel optimization Huang et al. (2023).

**Definition 1** (*L*-Lipschitz Continuity Yang et al. (2023)). *A mapping $F : \mathbb{R}^p \to \mathbb{R}$ is said to be L-Lipschitz continuous if there exists $L \geq 0$ such that for all $x_1, x_2 \in \mathbb{R}^p$,*

$$\|F(x_1) - F(x_2)\| \leq L\|x_1 - x_2\|. \tag{31}$$

Since the overall objective $\Phi(x)$ is non-convex, the goal is expected to find an $\epsilon$-accurate stationary point defined as follows.

**Definition 2** ($\epsilon$-Accurate Stationary Point Yang et al. (2023)). *A point $x$ is an $\epsilon$-accurate stationary point of $\Phi(x)$ if*

$$\mathbb{E}\left[\|\nabla\Phi(x)\|^2\right] \leq \epsilon, \tag{32}$$

*where $\nabla\Phi(x)$ is the gradient of $\Phi$ at $x$ and $\epsilon > 0$ is a prescribed tolerance.*

The following assumption characterizes the geometries of the objective functions.

**Assumption 1** (Twice Continuously Differentiable Yang et al. (2023)). *For all $i \in \{1, \ldots, n\}$ and any $(x, y) \in \mathbb{R}^{d_x} \times \mathbb{R}^{d_y}$, both $f_i(x, y)$ and $g_i(x, y)$ are twice continuously differentiable.*

**Assumption 2** ($\mu_g$-Strong Convexity in $y$ Yang et al. (2023)). *For each $i$ and any $x$, $g_i(x, \cdot)$ is $\mu_g$-strongly convex in $y$, i.e.,*

$$\nabla_{yy}^2 g_i(x, y) \succeq \mu_g I, \quad \mu_g > 0. \tag{33}$$

**Assumption 3** (Lipschitz Continuity of $f_i$, $g_i$ and Their Derivatives Yang et al. (2023)). *There exist constants $L_f, L_1, L_2, L_3 \geq 0$ such that for all $i$ and any $(x_1, y_1), (x_2, y_2)$, it holds that*

$$\|f_i(x_1, y_1) - f_i(x_2, y_2)\| \leq L_f \ d, \tag{34a}$$

$$\|\nabla f_i(x_1, y_1) - \nabla f_i(x_2, y_2)\| \leq L_1 \ d, \tag{34b}$$

$$\|\nabla g_i(x_1, y_1) - \nabla g_i(x_2, y_2)\| \leq L_1 \ d, \tag{34c}$$

$$\|\nabla^2 f_i(x_1, y_1) - \nabla^2 f_i(x_2, y_2)\| \leq L_2 \ d, \tag{34d}$$

$$\|\nabla^2 g_i(x_1, y_1) - \nabla^2 g_i(x_2, y_2)\| \leq L_2 \ d, \tag{34e}$$

$$\|\nabla^3 g_i(x_1, y_1) - \nabla^3 g_i(x_2, y_2)\| \leq L_3 \ d, \tag{34f}$$

*where $d := \sqrt{\|x_1 - x_2\|^2 + \|y_1 - y_2\|^2}$ is the Euclidean distance between $(x_1, y_1)$ and $(x_2, y_2)$.*

The Lipschitz continuity of the third-order derivative is necessary here to ensure the smoothness of $z^*(x)$, which guarantees the descent in the iterations of the linear system function (see (6)), under the more challenging simultaneous and single-loop updating structure considered in this study. Next, we assume the bounded variance conditions on the gradients and second-order derivatives.

**Assumption 4** (Bounded Variance of Stochastic Estimates Yang et al. (2023)). *There exist $\sigma_f^2, \sigma_g^2, \sigma_{gg}^2 \geq 0$ such that for all $i$ and any $(x, y)$,*

$$\mathbb{E}\left[\|\nabla f_i(x, y) - \nabla f_i(x, y; \xi)\|^2\right] \leq \sigma_f^2, \tag{35a}$$

$$\mathbb{E}\left[\|\nabla g_i(x, y) - \nabla g_i(x, y; \zeta)\|^2\right] \leq \sigma_g^2, \tag{35b}$$

$$\mathbb{E}\left[\|\nabla^2 g_i(x, y) - \nabla^2 g_i(x, y; \zeta)\|^2\right] \leq \sigma_{gg}^2. \tag{35c}$$

**Assumption 5** (Global Heterogeneity Yang et al. (2023)). *There exist $\beta_{gh} \geq 1$ and $\sigma_{gh} \geq 0$ such that for all $(x, y)$,*

$$\sum_{i=1}^{n} p_i \|\nabla_y g_i(x, y)\|^2 \leq \beta_{gh}^2 \left\|\sum_{i=1}^{n} p_i \nabla_y g_i(x, y)\right\|^2 + \sigma_{gh}^2. \tag{36}$$

This assumption of global heterogeneity uses $\beta_{gh}$ and $\sigma_{gh}$ to measure the dissimilarity of $\nabla_y g_i(x, y)$ for all $i$.

### 5.2 Convergence and Complexity Analysis

This section sketches the key convergence and complexity results of ASFBO and its accelerated variant LA–ASFBO while preserving all relevant formulae. The detailed theoretical analysis can be found in Appendix C.

Without system heterogeneity, every client has an identical gradient–coefficient norm

$$\|a_1^{(t)}\| = \cdots = \|a_n^{(t)}\|, \tag{37}$$

which yields identical weights $w_i = p_i, \forall i$, and the upper-level objective reduces to

$$\widetilde{\Phi}(x^{(t)}) = \sum_{i=1}^{n} w_i f_i(x^{(t)}, \widetilde{y}^*(x^{(t)})). \tag{38}$$

With heterogeneity

$$\|a_i^{(t)}\| \neq \|a_j^{(t)}\| \quad (\exists i \neq j), \tag{39}$$

hence,

$$w_i = \frac{\|a_i^{(t)}\|}{\sum_j p_j \|a_j^{(t)}\|} p_i \neq p_i, \tag{40}$$

and the original form

$$\Phi(x^{(t)}) = \sum_{i=1}^{n} p_i f_i(x^{(t)}, y^*(x^{(t)})) \tag{41}$$

must be retained.

**Theorem 1** (Convergence of ASFBO/LA–ASFBO). *Under Assumptions 1–5, the following holds:*

**(1) Without system heterogeneity:**

$$\min_t \|\nabla \widetilde{\Phi}(x^{(t)})\|^2 \leq \mathcal{O}\left(M_{sync} \cdot \sqrt{\frac{1}{P \bar{\tau} T}}\right) + \mathcal{O}\left(M_{part} \cdot \frac{n - P}{n - 1} \cdot \sqrt{\frac{\bar{\tau}}{PT}}\right) + \mathcal{O}\left(M_{drift} \cdot \frac{1}{\bar{\tau} T}\right); \tag{42}$$

**(2) With system heterogeneity:**

$$\min_t \|\nabla\Phi(x^{(t)})\|^2 \leq \mathcal{O}\left(M \cdot M_{sync} \cdot \sqrt{\frac{1}{P\bar{\tau}T}}\right) + \mathcal{O}\left(M \cdot M_{part} \cdot \frac{n-P}{n-1} \cdot \sqrt{\frac{\bar{\tau}}{PT}}\right) + \mathcal{O}\left(M \cdot M_{drift} \cdot \frac{1}{\bar{\tau}T}\right),$$
(43)

*where $\bar{\tau}$ is the average local iterations per round of communications for all clients, and $P$ is the average number of active clients engaged during each round of communication. The constants $M, M_{sync}, M_{part}, M_{drift}$ are defined in the Appendix C.10.*

Based on Theorem 1, we derive the communication/computation complexity of BO/LA–ASFBO as follows.

**Corollary 1** (Complexity of ASFBO/LA–ASFBO). *For the proposed ASFBO/LA–ASFBO, we have the following results for their complexity.*

- *If almost all clients participate each round ($P \approx n$), achieving $\epsilon$ accuracy needs $T = \mathcal{O}(\epsilon^{-2})$ communication rounds and $\bar{\tau}T = \mathcal{O}(n^{-1}\epsilon^{-2})$ local steps per client. Choosing $\bar{\tau} = \mathcal{O}(T/n)$ reduces communication to $T = \mathcal{O}(\epsilon^{-1})$.*

- *If only some clients participate each round ($P < n$), the requirements become $T = \mathcal{O}(P^{-1}\epsilon^{-2})$ communications and $\bar{\tau}T = \mathcal{O}(P^{-1}\epsilon^{-2})$ local steps per client.*

## 6 Experiments

In this section, the performance of the ASFBO and LA-ASFBO algorithms will be compared with SimFBO/ShroFBO Yang et al. (2023), and FedNest/LFedNest Tarzanagh et al. (2022) on IID and Non-IID MNIST dataset settings with MLP backbones. The detailed experiment settings can be found in Appendix E. The experiments focus on hyperrepresentation tasks in federated learning, where the goal is to train models that learn representations generalizing across multiple clients. This involves optimizing two levels: Lower-level problem, optimizing client-specific parameters (e.g., output layer weights); Upper-level problem, optimizes shared parameters (e.g., hidden layer weights) for better generalization.

### 6.1 Model architecture and dataset

**Model.** For the hyper-representation experiment, we use Multilayer Perceptron (MLP) model. The 2-layer multilayer perceptron has 784 input units and 200 hidden units so that the hidden layer parameters (157,000 parameters) are optimized for solving the upper-level problem and the output layer parameters (2,010 parameters) are optimized for solving the lower-level problem.

**Dataset.** For the hyper-representation experiment, we use full MNIST datasets. In the experiment with heterogeneous local computation, we treat the first 2000 images in MNIST's default training dataset as the training data and the first 1000 images in MNIST's default test dataset as test data.

### 6.2 Comparison of existing methods

The comparison results are presented in Figure 1. Under the IID setting (Figures 1 (a)–(d)), all algorithms eventually converge, albeit with varying performance. Notably, the proposed ASFBO and its variant LA-ASFBO exhibit faster convergence than the baselines, with LA-ASFBO achieving slightly superior performance, attributed to its incorporation of the STORM variance reduction technique.

In the Non-IDD setting (Figures 1 (e)–(h)), the behavior of the algorithms diverges more significantly. LFedNest, which is primarily designed for homogeneous client distributions, fails to converge under this heterogeneous setting. In contrast, ASFBO demonstrates markedly faster convergence compared to SimFBO, ShroFBO, FedNest, and LFedNest. Moreover, LA-ASFBO further accelerates convergence over ASFBO, again benefiting from the STORM-based variance reduction. Overall, the experimental results show that the proposed methods not only attain higher training and testing accuracy but also exhibit stronger generalization capabilities in both IID and Non-IID scenarios.

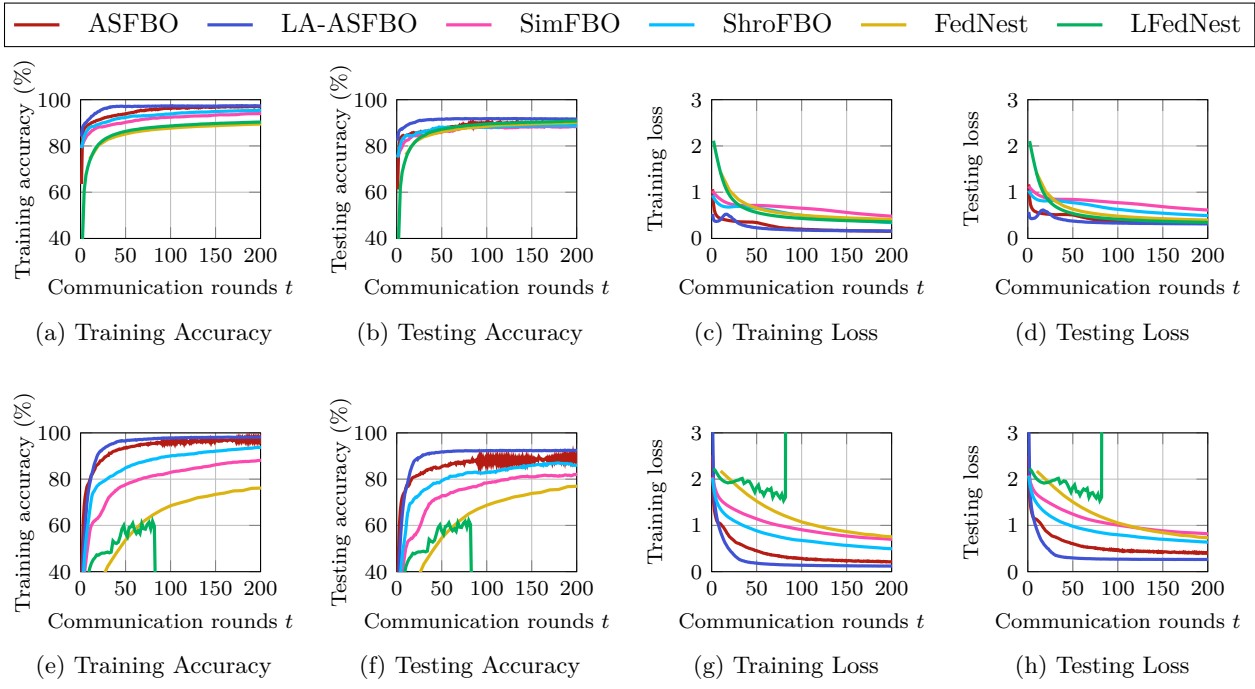

Figure 1: Accuracy and loss of each algorithm v.s. # of communication rounds on IID (subplots (a)-(d)) and Non-IID dataset settings (subplots (e)-(h))

## 7  Conclusion

In this paper, we addressed federated bilevel optimization in the presence of both data heterogeneity (non-IID client data) and system heterogeneity (partial client participation per round and varying numbers of local iterations). By unifying a communication-efficient single-loop framework with momentum, re-weighted aggregation, and AdaGrad-Norm scaling, we introduced ASFBO, which employs a surrogate hyper-gradient to sidestep costly second-order terms; its variance-reduced extension LA-ASFBO further leverages STORM for faster, more stable client updates. Our analysis shows that both algorithms reach an $\epsilon$-accurate stationary point of a non-convex–strongly-convex objective within $\mathcal{O}(\epsilon^{-1})$ communication rounds and $\mathcal{O}(n^{-1}\epsilon^{-2})$ local computations per client, while experimental results on federated hyper-representation learning confirm superior convergence across IID and Non-IID settings. These findings position ASFBO and LA-ASFBO as practical, theoretically grounded advances for efficient and robust federated learning in heterogeneous environments.

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

# Supplementary material

## A    Problem Background

Based on the definition of the federated bilevel optimization problem (1) and its main Assumptions 1–5, this section analyzes the key challenges in solving such problems and introduces relevant theoretical foundations of federated hyperparameter estimation.

### A.1    Key Challenges in Federated Bilevel Optimization

Effectively solving the federated bilevel optimization problem (1) entails several major challenges, including the computation of the federated hyper-gradient $\nabla\Phi(x)$, the presence of data and system-level heterogeneity, and the complexity introduced by the nested optimization structure. Before analyzing these challenges, we first define the mathematical form of the federated hyper-gradient.

According to the implicit function theorem Griewank & Walther (2008), under Assumptions 1—5, the federated hyper-gradient $\nabla\Phi(x)$ can be expressed as

$$\nabla\Phi(x) = \sum_{i=1}^{n} p_i \nabla_x f_i(x, y^*) - \nabla_{xy}^2 G(x, y^*)\left[\nabla_{yy}^2 G(x, y^*)\right]^{-1} \sum_{i=1}^{n} p_i \nabla_y f_i(x, y^*), \tag{44}$$

where $\nabla_x f_i(x, y^*)$ is the partial derivative of client $i$'s upper-level objective function $f_i$ with respect to $x$, and $\nabla_y f_i(x, y^*)$ is its partial derivative with respect to $y$. The term $\nabla_{xy}^2 G(x, y^*)$ denotes the mixed second-order partial derivative (Jacobian) of the global lower-level objective $G$, and $\nabla_{yy}^2 G(x, y^*)$ denotes its Hessian matrix with respect to $y$.

Specifically, the first term on the right-hand side of (44), $\sum_{i=1}^{n} p_i \nabla_x f_i(x, y^*)$, represents the direct influence of $f_i$ on $x$; the second term, $-\nabla_{xy}^2 G(x, y^*)\left[\nabla_{yy}^2 G(x, y^*)\right]^{-1} \sum_{i=1}^{n} p_i \nabla_y f_i(x, y^*)$, accounts for the indirect influence of $f_i$ on $x$ through $y^*(x)$, reflecting the dependency on the solution to the lower-level problem. This indicates that the federated hyper-gradient $\nabla\Phi(x)$ depends not only on first-order gradients but also on second-order terms (Jacobian and Hessian matrices) and complex interactions among local upper-level and lower-level objectives from all $n$ clients.

We now analyze these challenges in detail:

(1) **Implicit and complex dependency of $y^*(x)$ on $x$.** Since $\Phi(x) = F(x, y^*(x))$, the mapping $y^*(x)$ is an implicit function of $x$, defined through the solution of an optimization problem. Therefore, computing $\nabla\Phi(x)$ requires accounting for how variations in $x$ influence $y^*(x)$. This nested structure can be described using the implicit function theorem Griewank & Walther (2008) and (44). In this setting, the upper-level problem cannot be solved independently, as it is entangled with the lower-level problem: each client $i$ must solve its own local lower-level problem $g_i(x, y)$ to contribute to the global lower-level objective $G(x, y)$ and ultimately obtain $y^*(x)$, which necessitates coordination across all $n$ clients. Furthermore, the mapping $y^*(x)$ is often highly sensitive to $x$, where small changes in $x$ can lead to complex nonlinear responses in $y^*(x)$, making the gradient computation both sensitive and computationally intensive.

(2) **High computational and communication costs of second-order terms.** The hyper-gradient computation involves the $p \times q$ Jacobian $\nabla_{xy}^2 G(x, y^*)$, the $q \times q$ Hessian $\nabla_{yy}^2 G(x, y^*)$, and its inverse $\left[\nabla_{yy}^2 G(x, y^*)\right]^{-1}$. In a federated setting, computing these quantities requires collecting and aggregating local Jacobians and Hessians from all clients. When the number of clients $n$ is large or the dimensions $p$ and $q$ are high (e.g., millions of parameters in neural networks), the cost of computing, storing, and inverting these matrices becomes prohibitive (e.g., the complexity of inverting a $q \times q$ Hessian is $\mathcal{O}(q^3)$, and storing it requires $\mathcal{O}(q^2)$ memory per client), which exceeds the capabilities of typical edge devices in federated learning. Transferring these matrices from clients to the server also demands significant bandwidth (e.g., communication complexity of $\mathcal{O}(nq^2)$), which is infeasible given limited communication resources in federated settings.

(3) **Bias in global hyper-gradient aggregation (client drift).** The federated hyper-gradient $\nabla\Phi(x)$ is a global quantity. However, in federated learning, naively averaging local approximations from each client as

$$\sum_{i=1}^{n} p_i \left( \nabla_x f_i(x, y^*) - \nabla_{xy}^2 g_i(x, y^*) \left[ \nabla_{yy}^2 g_i(x, y^*) \right]^{-1} \nabla_y f_i(x, y^*) \right) \tag{45}$$

cannot yield an accurate estimate of the global hyper-gradient $\nabla\Phi(x)$. This is due to the nonlinear nature of the matrix inverse $\left[ \nabla_{yy}^2 g_i(x, y^*) \right]^{-1}$, which causes (45) to be mathematically inequivalent to $\nabla\Phi(x)$.

The correct way to aggregate second-order terms in the hyper-gradient is

$$\nabla_{xy}^2 G(x, y^*) = \sum_{i=1}^{n} p_i \nabla_{xy}^2 g_i(x, y^*), \tag{46a}$$

$$\nabla_{yy}^2 G(x, y^*) = \sum_{i=1}^{n} p_i \nabla_{yy}^2 g_i(x, y^*), \tag{46b}$$

i.e., all values must be aggregated across clients before performing matrix inversion and multiplication. Mathematically,

$$\sum_{i=1}^{n} p_i A_i \left[ B_i \right]^{-1} C_i \neq \left( \sum_{i=1}^{n} p_i A_i \right) \left( \sum_{i=1}^{n} p_i B_i \right)^{-1} \left( \sum_{i=1}^{n} p_i C_i \right), \tag{47}$$

where $A_i = \nabla_{xy}^2 g_i, B_i = \nabla_{yy}^2 g_i, C_i = \nabla_y f_i$. This nonlinearity introduces a bias known as client drift, which affects the convergence of optimization algorithms.

Moreover, due to differences in the data distributions of each client $i$, as represented by $\xi_i$ and $\zeta_i$, the local objectives $f_i$ and $g_i$ also differ. This heterogeneity implies that the local optima $y_i^*(x)$ may diverge from the global optimum $y^*(x)$, exacerbating client drift.

Client drift complicates convergence and may lead to suboptimal solutions, as the aggregated gradient fails to reflect the true descent direction of the federated hyper-gradient $\nabla\Phi(x)$.

## A.2 Federated Hypergradient Estimation

Direct computation of the federated hypergradient $\nabla\Phi(x)$ is prohibitively complex due to the implicit dependence of the lower-level solution $y^*(x)$ on $x$ and the heterogeneity inherent in federated settings. Therefore, we introduce a surrogate hypergradient:

$$\nabla\bar{F}(x, y, z) = \nabla_x F(x, y) - \nabla_{xy}^2 G(x, y) z, \tag{48}$$

where $y$ approximates the true lower-level solution $y^*(x)$ and $z \in \mathbb{R}^{d_y}$ is an auxiliary vector that approximates the term $\left[ \nabla_{yy}^2 G(x, y) \right]^{-1} \nabla_y F(x, y)$:

$$z = \left[ \nabla_{yy}^2 G(x, y) \right]^{-1} \nabla_y F(x, y). \tag{49}$$

By comparing the surrogate (48) with the true federated hypergradient (44), we see that $\nabla_x F(x, y)$ approximates $\nabla_x F(x, y^*)$ when $y \approx y^*(x)$, and $\nabla_{xy}^2 G(x, y) z$ approximates $\nabla_{xy}^2 G(x, y^*) \left[ \nabla_{yy}^2 G(x, y^*) \right]^{-1} \nabla_y F(x, y^*)$ when $y \approx y^*(x)$ and $z \approx z^*(x)$. Thus, the accuracy of $\nabla\bar{F}(x, y, z)$ hinges on how well $y$ and $z$ approximate their true values. Exact estimation of $y$ would require solving $\min_y G(x, y)$, which is infeasible globally in federated settings, so we approximate $y^*(x)$ via local client updates. Likewise, $z$ is defined by the global linear system

$$\nabla_{yy}^2 G(x, y) z = \nabla_y F(x, y), \tag{50}$$

whose solution yields the desired Hessian-inverse term. Solving this linear system is equivalent to minimising the quadratic objective

$$\min_z R(x, y, z) = \tfrac{1}{2} z^\top \nabla^2_{yy} G(x, y)\, z - z^\top \nabla_y F(x, y), \tag{51}$$

since

$$\nabla_z R(x, y, z) = \nabla^2_{yy} G(x, y)\, z - \nabla_y F(x, y), \tag{52}$$

and setting $\nabla_z R = 0$ recovers (50). In federated settings, $G$ and $F$ decompose as

$$G(x, y) = \sum_{i=1}^{n} p_i\, g_i(x, y), \tag{53a}$$

$$F(x, y) = \sum_{i=1}^{n} p_i\, f_i(x, y), \tag{53b}$$

and hence the global auxiliary objective decomposes:

$$R(x, y, z) = \sum_{i=1}^{n} p_i\, R_i(x, y, z), \tag{54}$$

where each local function is

$$R_i(x, y, z) = \tfrac{1}{2} z^\top \nabla^2_{yy} g_i(x, y)\, z - z^\top \nabla_y f_i(x, y), \tag{55}$$

with gradient

$$\nabla_z R_i(x, y, z) = \nabla^2_{yy} g_i(x, y)\, z - \nabla_y f_i(x, y). \tag{56}$$

Each client can thus compute $\nabla^2_{yy} g_i$ and $\nabla_y f_i$ locally, minimise $R_i$ to update $z$, and send $\nabla_z R_i$ to the server. The server aggregates these to refine the global $z$, driving it toward the solution of (50). Similarly, the global lower-level gradient

$$\nabla_y G(x, y) = \sum_{i=1}^{n} p_i\, \nabla_y g_i(x, y) \tag{57}$$

is obtained by aggregating local $\nabla_y g_i$. Finally, with federated estimates $y$ and $z$, the surrogate hypergradient

$$\nabla \bar{F}(x, y, z) = \sum_{i=1}^{n} p_i\, \nabla \bar{f}_i(x, y, z), \tag{58}$$

where

$$\nabla \bar{f}_i(x, y, z) = \nabla_x f_i(x, y) - \nabla^2_{xy} g_i(x, y)\, z, \tag{59}$$

enables a communication-efficient approximation of the true federated hypergradient without explicit Hessian inversion.

## B  Algorithm Details

### B.1  Communication Rounds and Client Initialization

In ASFBO and LA-ASFBO, communication rounds form the core of the algorithm operation. The algorithm assumes $n$ clients participate in optimization and conducts $T$ communication rounds indexed by $t = 0, 1, \ldots, T-1$. In each round only a subset of clients, denoted $C^{(t)}$ with $|C^{(t)}| \leq n$, is active. Without loss of generality, $C^{(t)}$ may be viewed as a uniform random sample without replacement from all clients. Not all

clients can participate in every round, reflecting practical federated learning scenarios (e.g., offline devices, high communication cost, or limited compute). Each active client $i \in C^{(t)}$ performs its local computation in parallel, and the server aggregates the results. This distributed design is a key feature of federated learning and exploits each client's local compute resources. At the start of round $t$, the server broadcasts the current global variables $\{x^{(t)}, y^{(t)}, z^{(t)}\}$ to all $i \in C^{(t)}$. Upon receiving these, each active client $i$ initializes its local copies $\{x_i^{(t,0)}, y_i^{(t,0)}, z_i^{(t,0)}\}$ by

$$x_i^{(t,0)} = x^{(t)}, \tag{60a}$$

$$y_i^{(t,0)} = y^{(t)}, \tag{60b}$$

$$z_i^{(t,0)} = z^{(t)}. \tag{60c}$$

Using the same initialization for all active clients mitigates bias from Non-IID data, ensuring consistent local updates and promoting global convergence. After variable initialization, each client initializes its momentum buffers $\{u_i^{(t,0)}, v_i^{(t,0)}, w_i^{(t,0)}\}$ via local gradients:

$$u_i^{(t,0)} = \nabla \bar{f}_i \big( x_i^{(t,0)}, y_i^{(t,0)}, z_i^{(t,0)}; \bar{\xi}_i^{(t,0)} \big), \tag{61a}$$

$$v_i^{(t,0)} = \nabla_y g_i \big( x_i^{(t,0)}, y_i^{(t,0)}; \zeta_i^{(t,0)} \big), \tag{61b}$$

$$w_i^{(t,0)} = \nabla_z R_i \big( x_i^{(t,0)}, y_i^{(t,0)}, z_i^{(t,0)}; \psi_i^{(t,0)} \big). \tag{61c}$$

Here $u_i^{(t,0)}$ is initialized by the local hypergradient surrogate $\nabla \bar{f}_i$ computed with sample $\bar{\xi}_i^{(t,0)}$; $v_i^{(t,0)}$ uses the local lower-level gradient with sample $\zeta_i^{(t,0)}$; and $w_i^{(t,0)}$ uses the local linear-system gradient with sample $\psi_i^{(t,0)}$. Momentum accelerates convergence by smoothing updates using historical gradients. The surrogate gradient at $(x_i^{(t,k)}, y_i^{(t,k)}, z_i^{(t,k)})$ is defined as

$$\nabla \bar{f}_i \big( x_i^{(t,k)}, y_i^{(t,k)}, z_i^{(t,k)}; \bar{\xi}_i \big) = \nabla_x f_i \big( x_i^{(t,k)}, y_i^{(t,k)}; \xi_i^{(t,k)} \big) - \nabla_{xy}^2 g_i \big( x_i^{(t,k)}, y_i^{(t,k)}; \zeta_i^{(t,k)} \big) z_i^{(t,k)}. \tag{62}$$

The local gradient of $R_i$ w.r.t. $z$ at the same point is

$$\nabla_z R_i \big( x_i^{(t,k)}, y_i^{(t,k)}, z_i^{(t,k)}; \psi_i^{(t,k)} \big) = \nabla_{yy}^2 g_i \big( x_i^{(t,k)}, y_i^{(t,k)}; \zeta_i^{(t,k)} \big) z_i^{(t,k)} - \nabla_y f_i \big( x_i^{(t,k)}, y_i^{(t,k)}; \xi_i^{(t,k)} \big). \tag{63}$$

By introducing the auxiliary variable $z$, the surrogate avoids direct Hessian inversion and enables efficient local estimation suitable for federated environments.

## B.2   Client Local Variable Updates

In communication round $t$, each active client $i \in C^{(t)}$ performs $\tau_i^{(t)}$ local iterations to update its local variables $\{x_i^{(t,k)}, y_i^{(t,k)}, z_i^{(t,k)}\}$. Here $x_i^{(t,k)}$ denotes the local copy of the upper-level variable (e.g., global model parameters or hyperparameters). $y_i^{(t,k)}$ denotes the local estimate of the lower-level variable, aiming to approximate the solution $y^*(x)$ of the lower-level problem. $z_i^{(t,k)}$ is the local copy of the solution to the global linear system, used to estimate the complex Hessian-inverse term in the federated hypergradient.

The goal of local updates is to progressively optimize these variables over multiple iterations so that they approach the global optimum in a distributed environment, while reducing communication overhead. Increasing $\tau_i^{(t)}$ accelerates local convergence and lowers communication frequency. However, a larger $\tau_i^{(t)}$ may induce significant client drift, i.e., local updates deviating from the global objective. Therefore, $\tau_i^{(t)}$ must be chosen to balance communication efficiency and convergence performance.

Specifically, at the $k$th local iteration, client $i$ updates all three variables simultaneously as follows:

$$x_i^{(t,k+1)} = x_i^{(t,k)} - \eta_x \, u_i^{(t,k)}, \tag{64a}$$

$$y_i^{(t,k+1)} = y_i^{(t,k)} - \eta_y \, v_i^{(t,k)}, \tag{64b}$$

$$z_i^{(t,k+1)} = z_i^{(t,k)} - \eta_z \, w_i^{(t,k)}, \tag{64c}$$

where $\{u_i^{(t,k)}, v_i^{(t,k)}, w_i^{(t,k)}\}$ are the momentum estimators guiding the update directions and magnitudes. The local learning rates $\{\eta_x, \eta_y, \eta_z\}$ control the step sizes for updating $\{x_i, y_i, z_i\}$. Each update subtracts a scaled momentum estimate rather than the raw gradient. Momentum accumulates historical gradient information, accelerating movement in flat regions and smoothing oscillations in non-convex landscapes. In federated learning, data are typically non-IID and gradient computations are noisy, so momentum enhances stability.

Unlike traditional bilevel methods (e.g., FedNest Tarzanagh et al. (2022)) that update $y$ and $x$ sequentially or in nested loops, ASFBO/LA-ASFBO updates $\{x_i, y_i, z_i\}$ concurrently within each local iteration. This design allows simultaneous progress on the upper and lower levels, eliminating nested loops and improving efficiency under limited local iteration budgets.

### B.3 Client Local Momentum Update

After each active client $i \in C^{(t)}$ completes its local variable updates, it must update its momentum estimators $\{u_i^{(t,k)}, v_i^{(t,k)}, w_i^{(t,k)}\}$. These momentum buffers guide the optimization direction of the local variables $\{x_i^{(t,k)}, y_i^{(t,k)}, z_i^{(t,k)}\}$. Their updates combine the current stochastic gradient with historical momentum to improve efficiency and stability.

In ASFBO, the **basic momentum update** is used, with the update rules given by:

$$u_i^{(t,k+1)} = \beta \nabla \bar{f}_i\big(x_i^{(t,k+1)}, y_i^{(t,k+1)}, z_i^{(t,k+1)}; \bar{\xi}_i^{(t,k+1)}\big) + (1-\beta)\, u_i^{(t,k)}, \tag{65a}$$

$$v_i^{(t,k+1)} = \beta \nabla_y g_i\big(x_i^{(t,k+1)}, y_i^{(t,k+1)}; \zeta_i^{(t,k+1)}\big) + (1-\beta)\, v_i^{(t,k)}, \tag{65b}$$

$$w_i^{(t,k+1)} = \beta \nabla_z R_i\big(x_i^{(t,k+1)}, y_i^{(t,k+1)}, z_i^{(t,k+1)}; \psi_i^{(t,k+1)}\big) + (1-\beta)\, w_i^{(t,k)}, \tag{65c}$$

where $\beta$ is the momentum coefficient in $(0,1)$ that balances the current gradient and past momentum. The terms $\{u_i^{(t,k)}, v_i^{(t,k)}, w_i^{(t,k)}\}$ are the momentum estimators from the previous iteration. The random variables $\{\xi_i^{(t,k+1)}, \zeta_i^{(t,k+1)}, \psi_i^{(t,k+1)}\}$ denote independently sampled local data points.

In the basic momentum method, each buffer is updated by a weighted average of the current stochastic gradient and historical momentum. When $\beta = 0$, the update relies solely on past momentum and ignores the current gradient. When $\beta$ approaches 1, the current gradient dominates and historical momentum has less influence. Typically, $\beta \in (0,1)$ is chosen to balance these effects.

In LA-ASFBO, a **momentum-based variance reduction (STORM)** Cutkosky & Orabona (2019) update is used to reduce the variance of gradient estimates, with rules:

$$u_i^{(t,k+1)} = \nabla \bar{f}_i\big(x_i^{(t,k+1)}, y_i^{(t,k+1)}, z_i^{(t,k+1)}; \bar{\xi}_i^{(t,k+1)}\big) + (1-\beta)\Big(u_i^{(t,k)} - \nabla \bar{f}_i\big(x_i^{(t,k)}, y_i^{(t,k)}, z_i^{(t,k)}; \bar{\xi}_i^{(t,k+1)}\big)\Big), \tag{66a}$$

$$v_i^{(t,k+1)} = \nabla_y g_i\big(x_i^{(t,k+1)}, y_i^{(t,k+1)}; \zeta_i^{(t,k+1)}\big) + (1-\beta)\Big(v_i^{(t,k)} - \nabla_y g_i\big(x_i^{(t,k)}, y_i^{(t,k)}; \zeta_i^{(t,k+1)}\big)\Big), \tag{66b}$$

$$w_i^{(t,k+1)} = \nabla_z R_i\big(x_i^{(t,k+1)}, y_i^{(t,k+1)}, z_i^{(t,k+1)}; \psi_i^{(t,k+1)}\big) + (1-\beta)\Big(w_i^{(t,k)} - \nabla_z R_i\big(x_i^{(t,k)}, y_i^{(t,k)}, z_i^{(t,k)}; \psi_i^{(t,k+1)}\big)\Big). \tag{66c}$$

Here $\beta$ controls the weight between historical momentum and the current correction term, with smaller $\beta$ giving greater influence to past momentum.

Each update consists of two parts. The first term is the current stochastic gradient at $(x_i^{(t,k+1)}, y_i^{(t,k+1)}, z_i^{(t,k+1)})$ computed using the current random samples. The second term is a correction based on the gradient difference at the previous point, using the same random samples, which estimates the local change trend and reduces variance.

Compared to the basic momentum method, where variance may accumulate across iterations, STORM leverages gradient differences to capture local changes and reduce estimate variance. Its key is using the **same random samples** at the current and previous points to compute the gradient difference, then recursively combining it with past momentum to form a new estimate. This approach theoretically reduces

stochastic gradient variance and improves stability. In federated learning with significant data heterogeneity, STORM's variance reduction effectively mitigates divergence across client gradient distributions. Moreover, STORM has been shown to converge faster in stochastic nonconvex optimization. More accurate momentum estimates better reflect the true gradient direction, speeding convergence. Additionally, STORM's use of gradient differences implies a form of adaptive step sizing that dynamically adjusts to gradient changes. In federated bilevel optimization, different variables $(x, y, z)$ may exhibit varying difficulty and gradient scales, and STORM's adaptivity helps address this challenge.

Momentum updates play a critical role in both ASFBO and LA-ASFBO. With limited communication rounds, accelerating local optimization is essential. Momentum accelerates variable updates in flat regions, reducing convergence time. It also smooths update directions, preventing oscillations near saddle points or local extrema, thus enhancing stability. By smoothing updates, momentum reduces noise from data heterogeneity and stochastic gradients, improving estimate reliability. Importantly, it enables simultaneous optimization of $\{x_i, y_i, z_i\}$ without serial or nested computations. This parallelism boosts computational efficiency and aligns with federated communication and compute constraints.

### B.4 Client Local Gradient Aggregation

In each communication round $t$, every active client $i \in C^{(t)}$ performs $\tau_i^{(t)}$ local iterations to update its local variables or momentum estimators. After completing these local updates, each client must aggregate its local computation into gradient summaries and upload them to the server. This process is called Client Local Gradient Aggregation.

Specifically, client $i \in C^{(t)}$ computes the aggregated gradients $\{q_{x,i}^{(t)}, q_{y,i}^{(t)}, q_{z,i}^{(t)}\}$ and sends them to the server for the global update. The aggregation is defined by

$$q_{x,i}^{(t)} = \sum_{k=0}^{\tau_i^{(t)}-1} u_i^{(t,k)} = \sum_{k=0}^{\tau_i^{(t)}-1} a_i^{(t,k)} \, \nabla \bar{f}_i\big(x_i^{(t,k)}, y_i^{(t,k)}, z_i^{(t,k)}; \bar{\xi}_i^{(t,k)}\big), \tag{67a}$$

$$q_{y,i}^{(t)} = \sum_{k=0}^{\tau_i^{(t)}-1} v_i^{(t,k)} = \sum_{k=0}^{\tau_i^{(t)}-1} a_i^{(t,k)} \, \nabla_y g_i\big(x_i^{(t,k)}, y_i^{(t,k)}; \zeta_i^{(t,k)}\big), \tag{67b}$$

$$q_{z,i}^{(t)} = \sum_{k=0}^{\tau_i^{(t)}-1} w_i^{(t,k)} = \sum_{k=0}^{\tau_i^{(t)}-1} a_i^{(t,k)} \, \nabla_z R_i\big(x_i^{(t,k)}, y_i^{(t,k)}, z_i^{(t,k)}; \psi_i^{(t,k)}\big), \tag{67c}$$

where the coefficients $a_i^{(t,k)}$ depend on the momentum coefficient $\beta$; see Lemma 1 for details.

Each aggregated gradient is the sum of the local momentum estimators over the $\tau_i^{(t)}$ iterations. During each local iteration, the client generates momentum buffers $\{u_i^{(t,k)}, v_i^{(t,k)}, w_i^{(t,k)}\}$ that capture the local stochastic gradient information. By summing these buffers, the client obtains a comprehensive optimization direction that better reflects the overall trend of its local data, aiding the adjustment of the global model. Gradient aggregation also avoids sending every individual gradient to the server, thereby reducing communication overhead.

Moreover, each aggregated gradient can be seen as a linear combination of the stochastic gradients across iterations. The coefficient $a_i^{(t,k)}$ determines how much each local stochastic gradient $\{\nabla \bar{f}_i, \nabla_y g_i, \nabla_z R_i\}$ contributes. Since the momentum buffers are updated via a momentum rule, $a_i^{(t,k)}$ implicitly incorporates the momentum coefficient $\beta$, smoothing the fusion of historical gradient information, reducing the noise of individual stochastic gradients, and yielding more stable gradient estimates. Furthermore, the exact form of $a_i^{(t,k)}$ can be adjusted to match the chosen momentum update (basic momentum or STORM) to meet different optimization needs. This design inherits the acceleration property of momentum—accumulating past gradients to speed up convergence—and thereby improves optimization efficiency.

## B.5 Server-Side Global Gradient Aggregation

In federated learning or distributed optimization, the server aggregates local gradient information from active clients each round to form the global gradient for updating global variables. This step is central to federated bilevel optimization, directly affecting the direction and efficacy of global model updates.

Ideally, the server receives each active client $i \in C^{(t)}$'s aggregated local gradients $\{q_{x,i}^{(t)}, q_{y,i}^{(t)}, q_{z,i}^{(t)}\}$ and simply computes a weighted average to obtain the global gradients $\{q_x^{(t)}, q_y^{(t)}, q_z^{(t)}\}$. The basic weighted aggregation is

$$q_x^{(t)} = \sum_{i \in C^{(t)}} \widetilde{p}_i \, q_{x,i}^{(t)}, \tag{68a}$$

$$q_y^{(t)} = \sum_{i \in C^{(t)}} \widetilde{p}_i \, q_{y,i}^{(t)}, \tag{68b}$$

$$q_z^{(t)} = \sum_{i \in C^{(t)}} \widetilde{p}_i \, q_{z,i}^{(t)}, \tag{68c}$$

where $\widetilde{p}_i = \frac{n}{|C^{(t)}|} p_i$ is the effective weight of client $i$. Here $p_i$ is the original client weight (e.g., proportional to data volume), satisfying $\sum_{i=1}^n p_i = 1$, and $|C^{(t)}|$ is the number of active clients in round $t$. Since $C^{(t)}$ is a uniform random sample without replacement, $\widetilde{p}_i$ ensures unbiased aggregation, i.e. $\mathbb{E}[\sum_{i \in C^{(t)}} \widetilde{p}_i] = 1$.

In practice, clients have heterogeneous computation and storage capabilities (system heterogeneity), so their local iteration counts $\tau_i^{(t)}$ may differ. Clients with more iterations generate higher-quality gradients $\{q_{x,i}^{(t)}, q_{y,i}^{(t)}, q_{z,i}^{(t)}\}$, embedding stronger directional information. Under simple weighted aggregation, these clients' contributions are amplified, especially if their $p_i$ are large. As noted in Wang et al. (2020), this bias skews the global gradient toward high-compute or high-iteration clients, causing the global model to drift away from the original objective and harming convergence and performance.

To reveal this heterogeneity-induced bias, Wang et al. (2020) rewrites the ideal aggregation (68) over all $n$ clients:

$$q_x^{(t)} = \sum_{i=1}^n p_i \, q_{x,i}^{(t)} = \rho^{(t)} \sum_{i=1}^n w_i^{(t)} h_{x,i}^{(t)} = \left( \sum_{j=1}^n p_j \|a_j^{(t)}\|_1 \right) \sum_{i=1}^n \frac{p_i \|a_i^{(t)}\|_1}{\sum_{j=1}^n p_j \|a_j^{(t)}\|_1} \frac{q_{x,i}^{(t)}}{\|a_i^{(t)}\|_1}, \tag{69a}$$

$$q_y^{(t)} = \sum_{i=1}^n p_i \, q_{y,i}^{(t)} = \rho^{(t)} \sum_{i=1}^n w_i^{(t)} h_{y,i}^{(t)} = \left( \sum_{j=1}^n p_j \|a_j^{(t)}\|_1 \right) \sum_{i=1}^n \frac{p_i \|a_i^{(t)}\|_1}{\sum_{j=1}^n p_j \|a_j^{(t)}\|_1} \frac{q_{y,i}^{(t)}}{\|a_i^{(t)}\|_1}, \tag{69b}$$

$$q_z^{(t)} = \sum_{i=1}^n p_i \, q_{z,i}^{(t)} = \rho^{(t)} \sum_{i=1}^n w_i^{(t)} h_{z,i}^{(t)} = \left( \sum_{j=1}^n p_j \|a_j^{(t)}\|_1 \right) \sum_{i=1}^n \frac{p_i \|a_i^{(t)}\|_1}{\sum_{j=1}^n p_j \|a_j^{(t)}\|_1} \frac{q_{z,i}^{(t)}}{\|a_i^{(t)}\|_1}, \tag{69c}$$

where $a_i^{(t)} = [a_i^{(t,0)}, \ldots, a_i^{(t,\tau_i^{(t)}-1)}]^\top$ is the vector of gradient coefficients for client $i$ in round $t$, relating to the momentum updates. $\|a_i^{(t)}\|_1$ is its $\ell_1$ norm, reflecting local optimization strength and directly tied to $\tau_i^{(t)}$. The scaling factor $\rho^{(t)} = \sum_{i=1}^n p_i \|a_i^{(t)}\|_1$ measures the aggregate optimization contribution. The reweighted weight $w_i^{(t)} = \frac{p_i \|a_i^{(t)}\|_1}{\rho^{(t)}}$ adjusts each client's contribution beyond its original $p_i$. The reweighted local gradient $h_{x,i}^{(t)} = \frac{q_{x,i}^{(t)}}{\|a_i^{(t)}\|_1}$ (and similarly $h_{y,i}^{(t)}, h_{z,i}^{(t)}$) normalizes by $\|a_i^{(t)}\|_1$, preventing gradient norms from scaling with $\tau_i^{(t)}$.

From the refactored formula (69), $q_x^{(t)}$ is a weighted sum of reweighted gradients $h_{x,i}^{(t)}$ with weights $w_i^{(t)}$. Since $w_i^{(t)}$ depends on both $p_i$ and $\|a_i^{(t)}\|_1$, heterogeneity in $\tau_i^{(t)}$ can shift $w_i^{(t)}$ away from $p_i$, biasing the global update toward high-iteration clients. This shift effectively changes the optimization objective to one weighted by $w_i^{(t)}$ instead of $p_i$.

To mitigate this bias, the algorithm employs reweighted global aggregation:

$$h_x^{(t)} = \sum_{i \in C^{(t)}} \widetilde{p}_i \, h_{x,i}^{(t)} = \sum_{i \in C^{(t)}} \widetilde{p}_i \, \frac{q_{x,i}^{(t)}}{\|a_i^{(t)}\|_1}, \tag{70a}$$

$$h_y^{(t)} = \sum_{i \in C^{(t)}} \widetilde{p}_i \, h_{y,i}^{(t)} = \sum_{i \in C^{(t)}} \widetilde{p}_i \, \frac{q_{y,i}^{(t)}}{\|a_i^{(t)}\|_1}, \tag{70b}$$

$$h_z^{(t)} = \sum_{i \in C^{(t)}} \widetilde{p}_i \, h_{z,i}^{(t)} = \sum_{i \in C^{(t)}} \widetilde{p}_i \, \frac{q_{z,i}^{(t)}}{\|a_i^{(t)}\|_1}, \tag{70c}$$

where $\widetilde{p}_i = \frac{n}{|C^{(t)}|} p_i$ as before. This method ensures that the server aggregates each client's "average" gradient direction rather than the cumulative sum, reducing the over-influence of high-iteration clients. Compared to the original aggregation (68), using reweighted gradients effectively corrects for system heterogeneity and yields a more balanced global update.

### B.6 Server-Side Adaptive Learning Rate Update

Adaptive learning rates dynamically adjust step sizes based on gradient information, improving convergence speed and stability. In ASFBO and LA-ASFBO, the server updates adaptive learning rates $\gamma_x^{(t)}, \gamma_y^{(t)}, \gamma_z^{(t)}$ using **AdaGrad-Norm** Ward et al. (2020) based on the global aggregated gradients $h_x^{(t)}, h_y^{(t)}, h_z^{(t)}$. AdaGrad-Norm Ward et al. (2020) is a variant of AdaGrad Duchi et al. (2011) that accumulates historical gradient norms to adjust the learning rate, suited for aggregated global gradients.

The scaling factors $\{s_x^{(t)}, s_y^{(t)}, s_z^{(t)}\}$ are updated as follows:

$$s_x^{(t+1)} = \varrho^{(t)} s_x^{(t)} + (1 - \varrho^{(t)}) \, \|h_x^{(t)}\|_2, \tag{71a}$$

$$s_y^{(t+1)} = \varrho^{(t)} s_y^{(t)} + (1 - \varrho^{(t)}) \, \|h_y^{(t)}\|_2, \tag{71b}$$

$$s_z^{(t+1)} = \varrho^{(t)} s_z^{(t)} + (1 - \varrho^{(t)}) \, \|h_z^{(t)}\|_2, \tag{71c}$$

where $\|h_x^{(t)}\|_2, \|h_y^{(t)}\|_2, \|h_z^{(t)}\|_2$ are the $\ell_2$ norms of the global aggregated gradients, measuring their magnitudes. The factors $\{s_x^{(t)}, s_y^{(t)}, s_z^{(t)}\}$ are initialized at zero and record historical gradient norm information. Their updates use an exponential moving average (EMA) style weighting to smooth changes and reduce short-term fluctuations. The coefficient $\varrho^{(t)} \in (0, 1)$ balances the weight between past information and the current gradient norm. When $\varrho^{(t)}$ is close to 1, the algorithm emphasizes historical information; when it is close to 0, it relies more on the current gradient norm.

After updating the scaling factors, the adaptive learning rates $\{\gamma_x^{(t)}, \gamma_y^{(t)}, \gamma_z^{(t)}\}$ are computed as

$$\gamma_x^{(t)} = \frac{\gamma_x}{s_x^{(t+1)} + \rho}, \tag{72a}$$

$$\gamma_y^{(t)} = \frac{\gamma_y}{s_y^{(t+1)} + \rho}, \tag{72b}$$

$$\gamma_z^{(t)} = \frac{\gamma_z}{s_z^{(t+1)} + \rho}, \tag{72c}$$

where $\{\gamma_x, \gamma_y, \gamma_z\}$ are fixed baseline learning rates and $\rho > 0$ is a small constant preventing division by zero for numerical stability. For example, if $\|h_x^{(t)}\|$ is large, then $s_x^{(t+1)}$ increases, enlarging the denominator and reducing $\gamma_x^{(t)}$, which prevents overly large updates. Conversely, when $\|h_x^{(t)}\|$ is small, $s_x^{(t+1)}$ grows slowly, keeping $\gamma_x^{(t)}$ relatively large to speed convergence. The behavior for $h_y^{(t)}$ and $h_z^{(t)}$ is analogous. This dynamic adjustment allows the algorithm to adapt to changes in gradient magnitudes during optimization.

### B.7 Server-Side Global Variable Update

In this step, the server updates the global variables $x^{(t)}, y^{(t)}, z^{(t)}$ using the aggregated gradients $h_x^{(t)}, h_y^{(t)}, h_z^{(t)}$. The updates use gradient descent combined with adaptive learning rates and a projection operation. Specifically, the update rules are:

$$x^{(t+1)} = x^{(t)} - \rho^{(t)} \gamma_x^{(t)} h_x^{(t)}, \tag{73a}$$

$$y^{(t+1)} = y^{(t)} - \rho^{(t)} \gamma_y^{(t)} h_y^{(t)}, \tag{73b}$$

$$z^{(t+1)} = \mathcal{P}_r\big(z^{(t)} - \rho^{(t)} \gamma_z^{(t)} h_z^{(t)}\big), \tag{73c}$$

where $\rho^{(t)}$ is the scaling factor defined in (69), representing the total optimization contribution of all clients. The adaptive learning rates $\{\gamma_x^{(t)}, \gamma_y^{(t)}, \gamma_z^{(t)}\}$ are generated by (72). $\mathcal{P}_r(z)$ is a projection onto the ball of radius $r$, defined as:

$$\mathcal{P}_r(z) = \begin{cases} z, & \|z\| \le r, \\ \frac{r}{\|z\|} z, & \|z\| > r, \end{cases} \tag{74}$$

where the projection radius $r = L_f/\mu_g$. This projection ensures boundedness of $z^{(t)}$, i.e. $\|z^{(t+1)}\| \le r$. The projection is crucial for the smoothness and convergence analysis of the global optimization problem.

## C  Theoretical Analysis Details

This section analyzes the convergence of the ASFBO and LA-ASFBO algorithms under Assumptions 1—5. The workflow of the convergence analysis is illustrated in Figure 2.

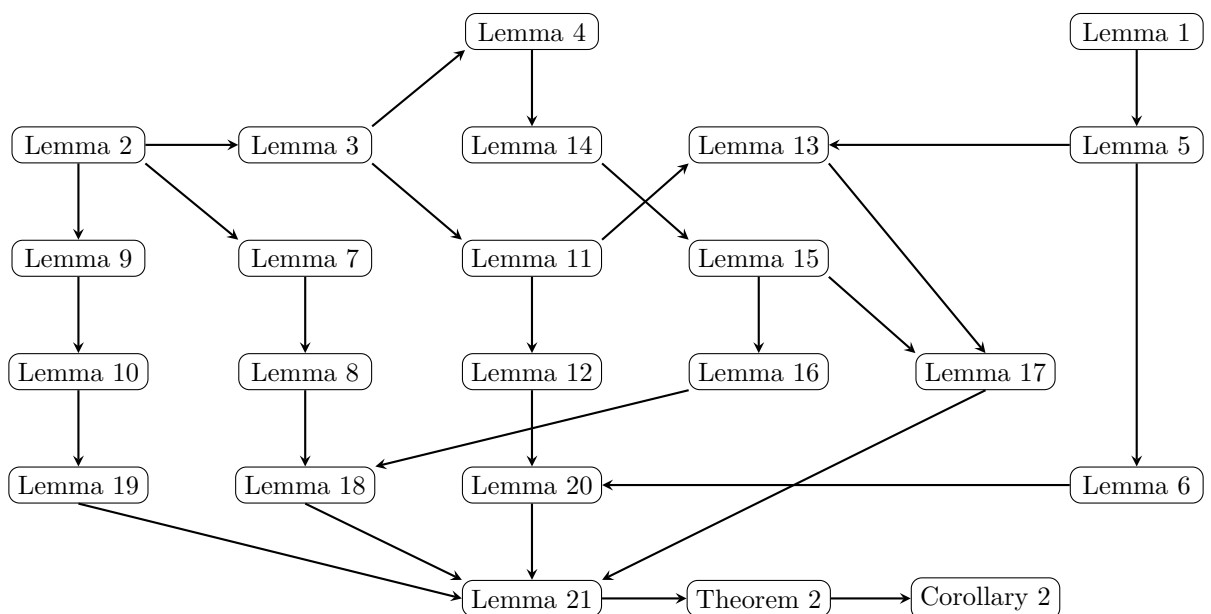

Figure 2: Flowchart of the convergence analysis for Algorithm 1

## C.1 Notation and Constraints

For convenience, we redefine the three global objective functions for solving the federated bilevel optimization problem by referring to (53) and (54).

$$\widetilde{F}(x,y) := \sum_{i=1}^{n} w_i f_i(x,y), \tag{75a}$$

$$\widetilde{G}(x,y) := \sum_{i=1}^{n} w_i g_i(x,y), \tag{75b}$$

$$\widetilde{R}(x,y,z) := \sum_{i=1}^{n} w_i R_i(x,y,z). \tag{75c}$$

Here $\widetilde{F}(x,y)$ is the global upper-level objective, $\widetilde{G}(x,y)$ is the global lower-level objective, and $\widetilde{R}(x,y,z)$ is the global auxiliary objective. The functions $f_i(x,y)$, $g_i(x,y)$, and $R_i(x,y,z)$ are the local upper-level, local lower-level, and local auxiliary objectives for client $i$. The number of clients is $n$, and $w_i$ is the weight of client $i$ satisfying $\sum_{i=1}^{n} w_i = 1$, indicating its contribution to the global objectives.

Then, the federated bilevel optimization problem (1) can be restated as:

$$\min_{x \in \mathbb{R}^p} \widetilde{\Phi}(x) = \widetilde{F}(x, \widetilde{y}^*(x)) := \sum_{i=1}^{n} w_i f_i(x, \widetilde{y}^*(x)) = \sum_{i=1}^{n} w_i \mathbb{E}_\xi \left[ f_i(x, \widetilde{y}^*(x); \xi) \right], \tag{76a}$$

$$\text{s.t.} \quad \widetilde{y}^*(x) = \arg\min_{y \in \mathbb{R}^q} \widetilde{G}(x,y) := \sum_{i=1}^{n} w_i g_i(x,y) = \sum_{i=1}^{n} w_i \mathbb{E}_\zeta \left[ g_i(x,y; \zeta) \right]. \tag{76b}$$

Here $\widetilde{y}^*(x)$ is the solution of the lower-level problem given $x$, and $\widetilde{\Phi}(x)$ is the global upper-level objective depending on $x$ and $\widetilde{y}^*(x)$.

The upper-level goal is to minimize $\widetilde{\Phi}(x)$ by optimizing $x$. The lower-level goal is, for each $x$, to solve the strongly convex problem $\widetilde{y}^*(x)$, whose uniqueness is guaranteed by the strong convexity of $g_i(x,y)$ (Assumption 2).

Similarly, referring to the federated hyper-gradient surrogate in (48), we define the global hyper-gradient surrogate:

$$\nabla \widetilde{\Phi}(x) := \sum_{i=1}^{n} w_i \nabla \bar{f}_i(x, \widetilde{y}^*, \widetilde{z}^*), \tag{77}$$

where $\widetilde{y}^* = \arg\min_y \widetilde{G}(x,y)$ and $\widetilde{z}^* = \arg\min_z \widetilde{R}(x, \widetilde{y}^*, z)$. The surrogate $\nabla \bar{f}_i$ is defined in (62).

To simplify notation, assume each client $i$ performs $\tau_i$ local iterations per global iteration. Then, combining (67) and (70), the local gradient aggregation for client $i$ can be restated as

$$h_{x,i}^{(t)} = \frac{1}{\|a_i^{(t)}\|_1} \sum_{k=0}^{\tau_i - 1} a_i^{(t,k)} \nabla \bar{f}_i(x_i^{(t,k)}, y_i^{(t,k)}, z_i^{(t,k)}; \bar{\xi}_i^{(t,k)}), \tag{78a}$$

$$h_{y,i}^{(t)} = \frac{1}{\|a_i^{(t)}\|_1} \sum_{k=0}^{\tau_i - 1} a_i^{(t,k)} \nabla_y g_i(x_i^{(t,k)}, y_i^{(t,k)}; \zeta_i^{(t,k)}), \tag{78b}$$

$$h_{z,i}^{(t)} = \frac{1}{\|a_i^{(t)}\|_1} \sum_{k=0}^{\tau_i - 1} a_i^{(t,k)} \nabla_z R_i(x_i^{(t,k)}, y_i^{(t,k)}, z_i^{(t,k)}; \psi_i^{(t,k)}), \tag{78c}$$

where $\|a_i^{(t)}\|_1 = \sum_{k=0}^{\tau_i - 1} a_i^{(t,k)}$ ensures aggregation is not biased by differing $\tau_i$, mitigating client heterogeneity.

The expected local aggregated gradients are

$$\widetilde{h}_{x,i}^{(t)} = \mathbb{E}\left[h_{x,i}^{(t)}\right] = \frac{1}{\|a_i^{(t)}\|_1} \sum_{k=0}^{\tau_i-1} a_i^{(t,k)} \nabla \bar{f}_i(x_i^{(t,k)}, y_i^{(t,k)}, z_i^{(t,k)}), \tag{79a}$$

$$\widetilde{h}_{y,i}^{(t)} = \mathbb{E}\left[h_{y,i}^{(t)}\right] = \frac{1}{\|a_i^{(t)}\|_1} \sum_{k=0}^{\tau_i-1} a_i^{(t,k)} \nabla_y g_i(x_i^{(t,k)}, y_i^{(t,k)}), \tag{79b}$$

$$\widetilde{h}_{z,i}^{(t)} = \mathbb{E}\left[h_{z,i}^{(t)}\right] = \frac{1}{\|a_i^{(t)}\|_1} \sum_{k=0}^{\tau_i-1} a_i^{(t,k)} \nabla_z R_i(x_i^{(t,k)}, y_i^{(t,k)}, z_i^{(t,k)}). \tag{79c}$$

Since the global objectives in (75) use weights $w_i$, the server aggregates global gradients by weighting the local aggregates in (78) as

$$h_x^{(t)} = \sum_{i \in C^{(t)}} \widetilde{w}_i h_{x,i}^{(t)}, \tag{80a}$$

$$h_y^{(t)} = \sum_{i \in C^{(t)}} \widetilde{w}_i h_{y,i}^{(t)}, \tag{80b}$$

$$h_z^{(t)} = \sum_{i \in C^{(t)}} \widetilde{w}_i h_{z,i}^{(t)}, \tag{80c}$$

where $\widetilde{w}_i = \frac{n}{|C^{(t)}|} w_i$ and $|C^{(t)}| = P$.

The expected global aggregated gradients are

$$\widetilde{h}_x^{(t)} = \mathbb{E}_{C^{(t)}}\left[h_x^{(t)}\right] = \sum_{i=1}^{n} w_i h_{x,i}^{(t)}, \tag{81a}$$

$$\widetilde{h}_y^{(t)} = \mathbb{E}_{C^{(t)}}\left[h_y^{(t)}\right] = \sum_{i=1}^{n} w_i h_{y,i}^{(t)}, \tag{81b}$$

$$\widetilde{h}_z^{(t)} = \mathbb{E}_{C^{(t)}}\left[h_z^{(t)}\right] = \sum_{i=1}^{n} w_i h_{z,i}^{(t)}, \tag{81c}$$

where $\mathbb{I}_{C^{(t)}}(i)$ is the indicator function (eigenfunction), the value of which is satisfied:

$$\mathbb{I}_{C^{(t)}}(i) = \begin{cases} 1, & \text{if} \quad i \in C^{(t)}, \\ 0, & \text{else.} \end{cases} \tag{82}$$

Combining (71) and (72), the server updates adaptive learning rates:

$$s_x^{(t+1)} = \varrho^{(t)} s_x^{(t)} + (1 - \varrho^{(t)}) \|h_x^{(t)}\|_2, \quad \gamma_x^{(t)} = \frac{\gamma_x}{s_x^{(t+1)} + \rho}; \tag{83a}$$

$$s_y^{(t+1)} = \varrho^{(t)} s_y^{(t)} + (1 - \varrho^{(t)}) \|h_y^{(t)}\|_2, \quad \gamma_y^{(t)} = \frac{\gamma_y}{s_y^{(t+1)} + \rho}; \tag{83b}$$

$$s_z^{(t+1)} = \varrho^{(t)} s_z^{(t)} + (1 - \varrho^{(t)}) \|h_z^{(t)}\|_2, \quad \gamma_z^{(t)} = \frac{\gamma_z}{s_z^{(t+1)} + \rho}. \tag{83c}$$

Then, using the scaling factor $\rho^{(t)}$ from (69) and adaptive learning rates, the server updates global variables:

$$x^{(t+1)} = x^{(t)} - \rho^{(t)} \gamma_x^{(t)} h_x^{(t)}, \tag{84a}$$

$$y^{(t+1)} = y^{(t)} - \rho^{(t)} \gamma_y^{(t)} h_y^{(t)}, \tag{84b}$$

$$z^{(t+1)} = \mathcal{P}_r\left(z^{(t)} - \rho^{(t)} \gamma_z^{(t)} h_z^{(t)}\right), \tag{84c}$$

where $\mathcal{P}_r(z) = \min\{1, r/\|z\|\}z$ with $r = L_f/\mu_g$.

To ensure the stability of algorithm convergence, we impose the following auxiliary constraints for convergence analysis.

**Assumption 6** (**Gradient Coefficient Bounds**). *For each global iteration $t \in \{0, 1, \ldots, T-1\}$, each client $i \in \{1, 2, \ldots, n\}$, and each local iteration $k \in \{0, 1, \ldots, \tau_i - 1\}$, the aggregation coefficient $a_i^{(t,k)}$ satisfies*

$$0 \leq \alpha_{\min} \leq a_i^{(t,k)} \leq \alpha_{\max}. \tag{85}$$

This bound on local gradient coefficients restricts the degree of system heterogeneity, and in the ASFBO framework this coefficient is determined by the momentum parameter $\beta$, so a reasonable choice of $\beta$ ensures this assumption holds.

**Assumption 7** (**Gradient Coefficient Vector Norm Bounds**). *For each global iteration $t \in \{0, 1, \ldots, T-1\}$ and each client $i \in \{1, 2, \ldots, n\}$, the $\ell_1$-norm of the coefficient vector $\|a_i^{(t)}\|_1$ satisfies*

$$c_a' \bar{\tau} \alpha_{\min} \leq \|a_i^{(t)}\|_1 \leq c_a \bar{\tau} \alpha_{\max}, \tag{86}$$

*where $\bar{\tau} = \frac{1}{n} \sum_{i=1}^{n} \tau_i$ is the average number of local iterations across all clients, and $c_a'$ and $c_a$ are positive constants.*

Bounding the $\ell_1$-norm of the coefficient vector ensures stability of the aggregated gradient.

**Assumption 8** (**Client Weight Coefficient Bounds**). *For all clients $i \in \{1, 2, \ldots, n\}$, the adjusted weight $w_i$ and the original weight $p_i$ satisfy*

$$\frac{\beta_{\min}}{n} \leq w_i \leq \frac{\beta_{\max}}{n}, \tag{87a}$$

$$\frac{\beta_{\min}'}{n} \leq p_i \leq \frac{\beta_{\max}'}{n}. \tag{87b}$$

Bounding client weight coefficients limits each client's contribution to the global gradient, maintaining balance in global optimization and characterizing the impact of system heterogeneity on convergence.

**Assumption 9** (**Adaptive Learning Rate Bounds**). *For each global iteration $t \in \{0, 1, \ldots, T-1\}$, the adaptive learning rates satisfy*

$$\gamma_{x,\min} \leq \gamma_x^{(t)} \leq \gamma_{x,\max}, \tag{88a}$$

$$\gamma_{y,\min} \leq \gamma_y^{(t)} \leq \gamma_{y,\max}, \tag{88b}$$

$$\gamma_{z,\min} \leq \gamma_z^{(t)} \leq \gamma_{z,\max}. \tag{88c}$$

Bounding adaptive learning rates mitigates slow convergence due to overly small rates in early optimization and oscillations caused by excessively large rates in later stages.

### C.2 Local Gradient Aggregation Coefficients

This subsection presents the explicit form of the local aggregation coefficients $a_i^{(t,k)}$ and their relationship with the momentum coefficient $\beta$.

**Lemma 1** (**Relation between $a_i^{(t,k)}$ and $\beta$**). *For each global iteration $t = 0, 1, \ldots, T-1$, each client $i$, and each local iteration $k = 0, 1, \ldots, \tau_i - 1$, the local aggregation coefficient $a_i^{(t,k)}$ satisfies:*

$$a_i^{(t,k)} = \begin{cases} \frac{1-(1-\beta)^{\tau_i}}{\beta}, & k = 0, \\ 1 - (1-\beta)^{\tau_i - k}, & 1 \leq k \leq \tau_i - 1, \end{cases} \tag{89}$$

*where $\beta$ is the momentum coefficient with $0 < \beta < 1$.*

*Proof.* Combining the local update for $z_i$ (see (64c)) and the gradient aggregation in (67c), we obtain

$$
\begin{aligned}
\sum_{k=0}^{\tau_i-1} w_i^{(t,k)} &= \frac{z_i^{(t,0)} - z_i^{(t,\tau_i)}}{\eta_z} \\
&= \sum_{k=0}^{\tau_i-1} a_i^{(t,k)} \nabla_z R_i\big(x_i^{(t,k)}, y_i^{(t,k)}, z_i^{(t,k)}; \psi_i^{(t,k)}\big).
\end{aligned}
\tag{90}
$$

From the initialization in (61c) and the basic momentum update in (65c), the momentum term $w_i^{(t,k)}$ can be written as

$$
\begin{aligned}
w_i^{(t,k)} &= \sum_{j=0}^{k} c_{j,k} \nabla_z R_i(x_i^{(t,j)}, y_i^{(t,j)}, z_i^{(t,j)}; \psi_i^{(t,j)}) \\
&= (1-\beta)^k \nabla_z R_i(x_i^{(t,0)}, y_i^{(t,0)}, z_i^{(t,0)}; \psi_i^{(t,0)}) \\
&\quad + \sum_{j=1}^{k} \beta(1-\beta)^{k-j} \nabla_z R_i(x_i^{(t,j)}, y_i^{(t,j)}, z_i^{(t,j)}; \psi_i^{(t,j)}),
\end{aligned}
\tag{91}
$$

where $c_{0,k} = (1-\beta)^k$ and $c_{j,k} = \beta(1-\beta)^{k-j}$ for $1 \le j \le k$.

Substituting (91) into (90) yields

$$
\begin{aligned}
\sum_{k=0}^{\tau_i-1} w_i^{(t,k)} &= \sum_{k=0}^{\tau_i-1} \Big(\sum_{j=0}^{k} c_{j,k}\Big) \nabla_z R_i(x_i^{(t,k)}, y_i^{(t,k)}, z_i^{(t,k)}; \psi_i^{(t,k)}) \\
&= \sum_{k=0}^{\tau_i-1} a_i^{(t,k)} \nabla_z R_i\big(x_i^{(t,k)}, y_i^{(t,k)}, z_i^{(t,k)}; \psi_i^{(t,k)}\big),
\end{aligned}
\tag{92}
$$

with $a_i^{(t,k)} = \sum_{j=k}^{\tau_i-1} c_{k,j}$.

For $k = 0$, we have

$$
a_i^{(t,0)} = \sum_{j=0}^{\tau_i-1} (1-\beta)^j = \frac{1 - (1-\beta)^{\tau_i}}{\beta}.
\tag{93}
$$

For $1 \le k \le \tau_i - 1$, letting $m = j - k$ gives

$$
a_i^{(t,k)} = \beta \sum_{m=0}^{\tau_i-1-k} (1-\beta)^m = 1 - (1-\beta)^{\tau_i-k}.
\tag{94}
$$

Therefore, $a_i^{(t,k)}$ has the form stated in (89). $\qquad\square$

### C.3 Properties of the Optimization Objectives

This subsection proves the $\mu_g$-strong convexity of the global lower-level objective $G(x,y)$ in $y$ and the local linear-system objective $R_i(x, y, z)$ in $z$.

**Lemma 2** ($G(x,y)$ **is $\mu_g$-strongly convex in** $y$). *Under Assumption 2, for any $x \in \mathbb{R}^{d_x}$ and $y \in \mathbb{R}^{d_y}$, the function $G(x,y)$ is $\mu_g$-strongly convex in $y$. Its Hessian $\nabla_{yy}^2 G(x,y)$ is positive definite and satisfies*

$$
\nabla_{yy}^2 G(x,y) \succeq \mu_g I.
\tag{95}
$$

*Proof.* By Assumption 2, each local $g_i(x,y)$ is $\mu_g$-strongly convex in $y$. From the definition $G(x,y) = \sum_{i=1}^{n} w_i g_i(x,y)$ we have

$$\nabla_{yy}^2 G(x,y) = \sum_{i=1}^{n} w_i \nabla_{yy}^2 g_i(x,y) \succeq \mu_g I, \tag{96}$$

since $\sum_{i=1}^{n} w_i = 1$. Hence the minimum eigenvalue of $\nabla_{yy}^2 G(x,y)$ is at least $\mu_g$, which by the second derivative condition of the strong convex function implies $G(x,y)$ is $\mu_g$-strongly convex in $y$. □

The $\mu_g$-strong convexity of $G(x,y)$ in $y$ guarantees a unique minimizer for each $x$ and fast convergence of gradient-based methods, ensuring stability of the global lower-level update $y^{(t)}$.

**Lemma 3** ($R_i(x,y,z)$ **is** $\mu_g$**-strongly convex in** $z$). *Under Assumptions 1 and 2, for any $x \in \mathbb{R}^{d_x}$, $y \in \mathbb{R}^{d_y}$, $z \in \mathbb{R}^{d_z}$, and client $i$, the function $R_i(x,y,z)$ is $\mu_g$-strongly convex in $z$. Its Hessian $\nabla_{zz}^2 R_i(x,y,z)$ is positive definite and satisfies*

$$\nabla_{zz}^2 R_i(x,y,z) \succeq \mu_g I. \tag{97}$$

*Proof.* From the local gradient in (63) we get

$$\nabla_{zz}^2 R_i(x,y,z) = \nabla_{yy}^2 G(x,y) \succeq \mu_g I, \tag{98}$$

and by Lemma 2, $\nabla_{yy}^2 G(x,y)$ is positive definite with minimal eigenvalue $\geq \mu_g$. Therefore $R_i(x,y,z)$ is $\mu_g$-strongly convex in $z$. □

The $\mu_g$-strong convexity of $R_i(x,y,z)$ in $z$ ensures a unique minimizer for each $(x,y)$ and fast convergence of gradient methods, guaranteeing stability of the local auxiliary update $z_i^{(t)}$.

## C.4 Properties of the Auxiliary Variable

This subsection proves the boundedness of the linear system optimization variable $z$, including the global optimal solution $z^*$ and the local auxiliary variable $z_i$.

**Lemma 4** (**Boundedness of** $z^*$). *Under Assumptions 1 and 2, for the optimal solution $z^*$ defined in the global linear system problem* (51), *the following holds:*

$$\|z^*\|^2 \leq \frac{L_f^2}{\mu_g^2} = r^2, \tag{99}$$

*where $z^* = \arg\min_z R(x,y^*,z)$ is the minimizer of the auxiliary objective $R(x,y^*,z)$ with respect to $z$.*

*Proof.* From Lemma 3, the auxiliary function $R(x,y^*,z)$ is $\mu_g$-strongly convex in $z$. Therefore, by the definition $z^* = \arg\min_z R(x,y^*,z)$, the point $z^*$ satisfies the first-order optimality condition:

$$\nabla_z R(x,y^*,z^*) = 0. \tag{100}$$

Substituting the optimality condition into the gradient of $R(x,y^*,z)$ at $(y = y^*, z = z^*)$, given by (52), we obtain:

$$\nabla_z R(x,y^*,z^*) = \nabla_{yy}^2 G(x,y^*)z^* - \nabla_y F(x,y^*) = 0. \tag{101}$$

By Lemma 2, the matrix $\nabla_{yy}^2 G(x,y^*)$ is positive definite, which implies:

$$z^* = \left[\nabla_{yy}^2 G(x,y^*)\right]^{-1} \nabla_y F(x,y^*). \tag{102}$$

Using the norm inequality $\|Ab\| \leq \|A\| \cdot \|b\|$ for any matrix $A$ and vector $b$, we get:

$$\|z^*\| = \left\| \left[\nabla_{yy}^2 G(x, y^*)\right]^{-1} \nabla_y F(x, y^*) \right\|$$
$$\leq \left\| \left[\nabla_{yy}^2 G(x, y^*)\right]^{-1} \right\| \cdot \|\nabla_y F(x, y^*)\|. \tag{103}$$

Squaring both sides:

$$\|z^*\|^2 \leq \left\| \left[\nabla_{yy}^2 G(x, y^*)\right]^{-1} \right\|^2 \cdot \|\nabla_y F(x, y^*)\|^2$$
$$\overset{(a)}{\leq} \frac{L_f^2}{\mu_g^2}$$
$$\overset{(b)}{=} r^2. \tag{104}$$

For step $(a)$, note that the smallest eigenvalue of $\nabla_{yy}^2 G(x, y^*)$ satisfies $\lambda_{\min} \geq \mu_g$, so:

$$\left\| \left[\nabla_{yy}^2 G(x, y^*)\right]^{-1} \right\|^2 \leq \frac{1}{\mu_g^2}. \tag{105}$$

For the gradient norm:

$$\|\nabla_y F(x, y^*)\|^2 \leq L_f^2, \tag{106}$$

which follows from the Lipschitz continuity of $\nabla_y f_i(x, y)$ and the convex combination structure in the definition of $F(x, y)$ in (75a).

In step $(b)$, the projection radius in (84) is defined as $r = L_f/\mu_g$.

This completes the proof of Lemma 4. □

The boundedness of $z^*$ guarantees the stability of the global auxiliary variable throughout the optimization process, preventing divergence due to unbounded growth. This property plays a crucial role in the convergence analysis, where it helps control error terms.

**Lemma 5** (**Boundedness of** $z_i$). *Under Assumptions 1 and 2, for any global iteration $t \in \{0, 1, \cdots, T-1\}$, client $i \in \{1, 2, \cdots, n\}$, and local iteration $k = \{0, 1, \cdots, \tau_i - 1\}$, the local auxiliary variable $z_i^{(t,k)}$ satisfies:*

$$r_i := \|z_i^{(t,k)}\| \leq \left(1 + \frac{\alpha_{\max}}{\alpha_{\min}}\right) r =: r_{\max}, \tag{107}$$

*where $r = L_f/\mu_g$ is the projection radius.*

*Proof.* Using the expression for $a_i^{(t,k)}$ in (89) and combining with (90), the update rule (64c) can be rewritten as:

$$z_i^{(t,k+1)} = z_i^{(t,k)} - \eta_z a_i^{(t,k)} \nabla_z R_i \left(x_i^{(t,k)}, y_i^{(t,k)}, z_i^{(t,k)}; \psi_i^{(t,k)}\right). \tag{108}$$

Substituting the definition of $\nabla_z R_i(x, y, z)$ from (63), we obtain:

$$z_i^{(t,k)} = \left(I - \eta_z a_i^{(t,k-1)} \nabla_{yy}^2 g_i(x_i^{(t,k-1)}, y_i^{(t,k-1)}; \psi_i^{(t,k-1)})\right) z_i^{(t,k-1)}$$
$$+ \eta_z a_i^{(t,k-1)} \nabla_y f_i(x_i^{(t,k-1)}, y_i^{(t,k-1)}; \psi_i^{(t,k-1)}). \tag{109}$$

Taking the $\ell_2$ norm of both sides:

$$\|z_i^{(t,k)}\| \overset{(a)}{\leq} \left\| I - \eta_z a_i^{(t,k-1)} \nabla_{yy}^2 g_i \right\| \cdot \|z_i^{(t,k-1)}\| + \eta_z a_i^{(t,k-1)} \|\nabla_y f_i\|$$

$$\overset{(b)}{\leq} (1 - \eta_z a_i^{(t,k-1)} \mu_g) \| z_i^{(t,k-1)} \| + \eta_z a_i^{(t,k-1)} L_f$$

$$\overset{(c)}{\leq} (1 - \eta_z \alpha_{\min} \mu_g) \| z_i^{(t,k-1)} \| + \eta_z \alpha_{\max} L_f$$

$$\overset{(d)}{\leq} (1 - \eta_z \alpha_{\min} \mu_g)^k \| z_i^{(t,0)} \| + \sum_{j=0}^{k-1} (1 - \eta_z \alpha_{\min} \mu_g)^j \eta_z \alpha_{\max} L_f$$

$$\overset{(e)}{\leq} (1 - \eta_z \alpha_{\min} \mu_g)^k \cdot r + \sum_{j=0}^{k-1} (1 - \eta_z \alpha_{\min} \mu_g)^j \eta_z \alpha_{\max} L_f$$

$$\overset{(f)}{\leq} r + \sum_{j=0}^{k-1} (1 - \eta_z \alpha_{\min} \mu_g)^j \eta_z \alpha_{\max} L_f$$

$$\overset{(g)}{\leq} r + \frac{\alpha_{\max} L_f}{\alpha_{\min} \mu_g}$$

$$= \left( 1 + \frac{\alpha_{\max}}{\alpha_{\min}} \right) r. \tag{110}$$

Step $(a)$ uses the triangle inequality and properties of matrix norms. Step $(b)$ uses the assumption that the Hessian $\nabla_{yy}^2 g_i$ is positive definite with minimal eigenvalue $\mu_g$. Step $(c)$ applies bounds $\alpha_{\min} \leq a_i^{(t,k)} \leq \alpha_{\max}$. Step $(d)$ unfolds the recurrence. Step $(e)$ uses the initialization condition $\| z_i^{(t,0)} \| = \| z^{(t)} \| \leq r$ from (60) and (73). Step $(f)$ holds because $(1 - \eta_z \alpha_{\min} \mu_g)^k \leq 1$. Step $(g)$ applies the geometric series upper bound.

This completes the proof of Lemma 5. $\qquad \square$

The boundedness of the local auxiliary variable $z_i$ ensures that $\| z_i^{(t,k)} \|$ remains within a fixed range over local iterations, preventing divergence due to heterogeneity. Since the server projects the global variable $z^{(t)}$ onto a ball of radius $r = L_f / \mu_g$, and $r_{\max}$ is a linear function of $r$, local variables inherit boundedness without extra projection. This justifies parameter choices and enables performance tuning by adjusting $\alpha_{\max}$ and $\alpha_{\min}$.

## C.5 Properties of the Gradients of the Optimization Objective

This subsection introduces and analyzes the Lipschitz continuity of the optimization objective and its gradients. The proofs are given in (Yang et al., 2023; Ghadimi & Wang, 2018; Chen et al., 2021). The results in this subsection will be used in Section C.8 to analyze the behavior of the global objective function during global iterations.

**Lemma 6** ($\nabla_z R_i(x, y, z)$ **is** $L_R$**-Lipschitz continuous w.r.t.** $(x, y)$ (Yang et al., 2023)). *Under Assumptions 1–3, for any $x \in \mathbb{R}^{d_x}$, $y \in \mathbb{R}^{d_y}$, and $i \in \{1, 2, \cdots, n\}$, the local auxiliary objective function $R_i(x, y, z)$ is $L_R$-Lipschitz continuous. That is, for any input pairs $(x_1, y_1), (x_2, y_2) \in \mathbb{R}^{d_x} \times \mathbb{R}^{d_y}$, it holds that*

$$\| \nabla_z R_i(x_1, y_1, z) - \nabla_z R_i(x_2, y_2, z) \|^2 \leq L_R^2 (\| x_1 - x_2 \|^2 + \| y_1 - y_2 \|^2), \tag{111}$$

*where the Lipschitz constant $L_R$ is defined as*

$$L_R^2 := 2(L_2^2 r_{\max}^2 + L_1^2). \tag{112}$$

The $L_R$-Lipschitz continuity of $\nabla_z R_i(x, y, z)$ with respect to $(x, y)$ limits the rate of gradient variation with changes in $x$ and $y$, which facilitates subsequent analysis on the stability of local updates and global aggregation.

**Lemma 7** (**Boundedness of Hypergradient Estimation Error** (Ghadimi & Wang, 2018)). *Under Assumptions 1–4, for any $x \in \mathbb{R}^{d_x}$ and $y \in \mathbb{R}^{d_y}$, it holds that*

$$\| \nabla f(x, y) - \nabla \Phi(x) \|^2 \leq \widetilde{L}^2 \| y - y^*(x) \|^2, \tag{113}$$

where $\nabla\Phi(x)$ is the hypergradient, $\nabla f(x,y)$ is its estimator at point $(x,y)$, and $y^*(x)$ is the optimal solution to the lower-level problem. The constant $\widetilde{L}$ is defined as

$$\widetilde{L} = L_1 + \frac{L_1^2}{\mu_g} + L_f\left(\frac{L_2}{\mu_g} + \frac{L_1 L_2}{\mu_g^2}\right). \tag{114}$$

Inequality (113) characterizes the deviation between the estimated hypergradient $\nabla f(x,y)$ and the true hypergradient $\nabla\Phi(x)$, which is controlled by the distance between $y$ and $y^*(x)$. The bound $\widetilde{L}$ reflects the magnitude of this gradient discrepancy.

**Lemma 8** ($\nabla\Phi(x)$ **is** $L_\Phi$**-Lipschitz continuous w.r.t.** $x$ (Ghadimi & Wang, 2018)). *Under Assumptions 1–4, the hypergradient $\nabla\Phi(x)$ is $L_\Phi$-Lipschitz continuous with respect to $x$. That is, for any $x_1, x_2 \in \mathbb{R}^{d_x}$, it holds that*

$$\|\nabla\Phi(x_1) - \nabla\Phi(x_2)\|^2 \le L_\Phi^2 \|x_1 - x_2\|^2, \tag{115}$$

*where the Lipschitz constant $L_\Phi$ is given by*

$$L_\Phi = \widetilde{L} + \frac{\widetilde{L}L_1}{\mu_g}. \tag{116}$$

The $L_\Phi$-Lipschitz continuity of the hypergradient $\nabla\Phi(x)$ constrains the rate of change with respect to $x$, ensuring the smoothness of the hypergradient. The constant $L_\Phi$ depends on the upper bound $\widetilde{L}$ of the hypergradient estimation error.

**Lemma 9** ($y^*(x)$ **is** $L_y$**-Lipschitz continuous w.r.t.** $x$ (Ghadimi & Wang, 2018)). *Under Assumptions 1–4, the optimal solution $y^*(x)$ to the lower-level optimization problem is $L_y$-Lipschitz continuous with respect to $x$. That is, for any $x_1, x_2 \in \mathbb{R}^{d_x}$, it holds that*

$$\|y^*(x_1) - y^*(x_2)\|^2 \le L_y^2 \|x_1 - x_2\|^2, \tag{117}$$

*where*

$$L_y = \frac{L_1}{\mu_g}. \tag{118}$$

The $L_y$-Lipschitz continuity of $y^*(x)$ ensures the smoothness of the lower-level solution with respect to $x$. The constant $L_y$ arises from the implicit function theorem and the strong convexity of $g(x,y)$.

**Lemma 10** ($\nabla y^*(x)$ **is** $L_{yx}$**-Lipschitz continuous w.r.t.** $x$ (Chen et al., 2021)). *Under Assumptions 1–4, the gradient of the lower-level optimal solution $\nabla y^*(x)$ is $L_{yx}$-Lipschitz continuous with respect to $x$. That is, for any $x_1, x_2 \in \mathbb{R}^{d_x}$, it holds that*

$$\|\nabla y^*(x_1) - \nabla y^*(x_2)\|^2 \le L_{yx}^2 \|x_1 - x_2\|^2, \tag{119}$$

*where*

$$L_{yx} = \frac{L_2 + L_2 L_y}{\mu_g} + \frac{L_1(L_2 + L_2 L_y)}{\mu_g^2}. \tag{120}$$

The $L_{yx}$-Lipschitz continuity of $\nabla y^*(x)$ controls the smoothness of the gradient of the lower-level solution with respect to $x$. The constant $L_{yx}$ depends on the smoothness and strong convexity of $y^*(x)$, as well as the effect of the second-order derivative $\nabla^2 G(x,y)$.

**Lemma 11** ($z^*(x)$ **is** $L_z$**-Lipschitz continuous w.r.t.** $x$ (Yang et al., 2023)). *Under Assumptions 1–4, the optimal solution $z^*(x)$ to the global linear system is $L_z$-Lipschitz continuous with respect to $x$. That is, for any $x_1, x_2 \in \mathbb{R}^{d_x}$, it holds that*

$$\|z^*(x_1) - z^*(x_2)\|^2 \le L_z^2 \|x_1 - x_2\|^2, \tag{121}$$

*where*

$$L_z = \left( \frac{2L_1^2}{\mu_g^2} + \frac{2L_f^2 L_2^2}{\mu_g^4} \right)^{\frac{1}{2}} \left( 1 + L_y^2 \right)^{\frac{1}{2}}. \tag{122}$$

The $L_z$-Lipschitz continuity of $z^*(x)$ guarantees the smoothness of the auxiliary variable with respect to $x$. The constant $L_z$ also depends on the smoothness and strong convexity of $y^*(x)$, as well as the second-order derivative $\nabla^2 G(x,y)$.

**Lemma 12** ($\nabla z^*(x)$ **is $L_{zx}$-Lipschitz continuous w.r.t.** $x$ (Yang et al., 2023)). *Under Assumptions 1–4, the gradient $\nabla z^*(x)$ of the optimal solution to the global linear system is $L_{zx}$-Lipschitz continuous with respect to $x$. That is, for any $x_1, x_2 \in \mathbb{R}^{d_x}$, it holds that*

$$\|\nabla z^*(x_1) - \nabla z^*(x_2)\|^2 \le L_{zx}^2 \|x_1 - x_2\|^2, \tag{123}$$

*where*

$$L_{zx} = \frac{2}{\mu_g} \left( (L_2^2 + r^2 L_3^2 + L_2^2 L_z^2)(1 + L_y^2) + L_2^2 L_z^2 \right)^{\frac{1}{2}}. \tag{124}$$

The $L_{zx}$-Lipschitz continuity of $\nabla z^*(x)$ ensures the smoothness of the gradient of the auxiliary optimal solution with respect to $x$. The constant $L_{zx}$ is the most complex and involves the smoothness of $z^*(x)$, as well as the influence of the third-order derivative $\nabla^3 G(x,y)$.

### C.6 Properties of Global Heterogeneity

Building upon the work in (Yang et al., 2023), this subsection investigates the impact of data heterogeneity-induced client drift on global aggregated gradients and establishes the boundedness of such drift. The results herein will be used in Section C.9 to analyze the behavior of the Lyapunov function of Algorithm 1 during the global iterative process.

**Lemma 13** (**Boundedness of Gradient Global Heterogeneity** (Yang et al., 2023)). *Under Assumptions 3 and 5, for each client $i \in \{1, 2, \cdots, n\}$, the gradients $\nabla \bar{f}_i(x,y,z), \nabla R_i(x,y,z), \nabla g_i(x,y)$ admit bounded global heterogeneity:*

$$\sum_{i=1}^{n} w_i \|\nabla \bar{f}_i(x,y,z)\|^2 \le 2r_{\max}^2 L_1^2 + 2L_f^2, \tag{125a}$$

$$\sum_{i=1}^{n} w_i \|\nabla_z R_i(x,y,z)\|^2 \le 2r_{\max}^2 L_1^2 + 2L_f^2, \tag{125b}$$

$$\sum_{i=1}^{n} w_i \|\nabla_y g_i(x,y)\|^2 \le \beta_{gh}^2 L_1^2 \|y - y^*(x)\|^2 + \sigma_{gh}^2, \tag{125c}$$

*where the weights $\{w_i\}_{i=1}^n$ are non-negative and satisfy $\sum_{i=1}^n w_i = 1$.*

Inequalities (125a) and (125b) provide constant upper bounds for the heterogeneity of the upper-level and auxiliary objective gradients, indicating that their variations are controlled when $z$ is bounded. In contrast, inequality (125c) shows that the heterogeneity of the lower-level gradient $\nabla_y g_i(x,y)$ is proportional to the distance between $y$ and the lower-level optimal solution $y^*(x)$, with an inherent heterogeneity term $\sigma_{gh}^2$. These bounds reflect the influence of client-wise data distribution differences and serve to quantify the degree of client drift.

**Lemma 14** (**Boundedness of Gradient Drift Across Clients** (Yang et al., 2023)). *Under Assumptions 3 and 5, for each global iteration $t \in \{0, 1, \cdots, T-1\}$, client $i \in \{1, 2, \cdots, n\}$, and local iteration $k \in$*

$\{0, 1, \cdots, \tau_i - 1\}$, *the drift in gradients* $\nabla \bar{f}_i(x, y, z), \nabla R_i(x, y, z), \nabla g_i(x, y)$ *is bounded by*

$$\left\| \nabla \bar{f}_i(x^{(t)}, y^{(t)}, z^{(t)}) - \nabla \bar{f}_i(x_i^{(t,k)}, y_i^{(t,k)}, z_i^{(t,k)}) \right\|^2 \le \Delta_{f,i}^{(t,k)}, \tag{126a}$$

$$\left\| \nabla_z R_i(x^{(t)}, y^{(t)}, z^{(t)}) - \nabla_z R_i(x_i^{(t,k)}, y_i^{(t,k)}, z_i^{(t,k)}) \right\|^2 \le \Delta_{R,i}^{(t,k)}, \tag{126b}$$

$$\left\| \nabla g_i(x^{(t)}, y^{(t)}) - \nabla g_i(x_i^{(t,k)}, y_i^{(t,k)}) \right\|^2 \le \Delta_{g,i}^{(t,k)}, \tag{126c}$$

*where the gradient drift upper bounds are defined as*

$$\Delta_{f,i}^{(t,k)} = \Delta_{R,i}^{(t,k)} := 3(L_1^2 + r^2 L_2^2) \left[ \|x_i^{(t,k)} - x^{(t)}\|^2 + \|y_i^{(t,k)} - y^{(t)}\|^2 \right] + 3L_1^2 \|z_i^{(t,k)} - z^{(t)}\|^2, \tag{127a}$$

$$\Delta_{g,i}^{(t,k)} := L_1^2 \left[ \|x_i^{(t,k)} - x^{(t)}\|^2 + \|y_i^{(t,k)} - y^{(t)}\|^2 \right]. \tag{127b}$$

The divergence in local sample distributions leads to gradient inconsistencies across clients. This lemma connects the gradient drift to local variable deviations via $\Delta_{f,i}^{(t,k)}, \Delta_{R,i}^{(t,k)}, \Delta_{g,i}^{(t,k)}$, enabling quantification of gradient estimation errors caused by client drift. In federated bilevel optimization, client drift plays a crucial role in convergence analysis and affects global objective updates. Hence, this lemma serves as a foundation for bounding convergence rate and error.

**Lemma 15** (**Boundedness of Client Drift** (Yang et al., 2023)). *Under Assumptions 1–7, for each global iteration* $t \in \{0, 1, \cdots, T - 1\}$, *client* $i \in \{1, 2, \cdots, n\}$, *and local iteration* $k \in \{0, 1, \cdots, \tau_i - 1\}$, *the drift of local variables* $x_i^{(t,k)}, y_i^{(t,k)}, z_i^{(t,k)}$ *is bounded by:*

$$\sum_{i=1}^{n} \frac{w_i}{\|a_i^{(t)}\|_1} \sum_{k=0}^{\tau_i - 1} a_i^{(t,k)} \mathbb{E} \left[ \left\| x^{(t)} - x_i^{(t,k)} \right\|^2 \right] \le \eta_x^2 \bar{\tau} \sigma_{M1}^2, \tag{128a}$$

$$\sum_{i=1}^{n} \frac{w_i}{\|a_i^{(t)}\|_1} \sum_{k=0}^{\tau_i - 1} a_i^{(t,k)} \mathbb{E} \left[ \left\| z^{(t)} - z_i^{(t,k)} \right\|^2 \right] \le \eta_z^2 \bar{\tau} \sigma_{M1}^2, \tag{128b}$$

$$
\begin{aligned}
\sum_{i=1}^{n} \frac{w_i}{\|a_i^{(t)}\|_1} \sum_{k=0}^{\tau_i - 1} a_i^{(t,k)} \mathbb{E} \left[ \left\| y^{(t)} - y_i^{(t,k)} \right\|^2 \right] \le \; & \frac{\eta_y^2 \bar{\tau}}{1 - 2\eta_y^2 c_a \bar{\tau} \alpha_{\max} L_1^2} \Big[ \alpha_{\max}^2 \sigma_g^2 \\
& + 2 c_a \alpha_{\max} L_1^2 \eta_x^2 \bar{\tau} \sigma_{M1}^2 \\
& + 2 c_a \alpha_{\max} L_1^2 \mathbb{E} \left[ \left\| y^{(t)} - \widetilde{y}^*(x^{(t)}) \right\|^2 \right] \\
& + 2 c_a \alpha_{\max} \sigma_{gh}^2 \Big],
\end{aligned}
\tag{128c}
$$

*where*

$$\sigma_{M1}^2 = \alpha_{\max}^2 (\sigma_f^2 + r_{\max}^2 \sigma_{gg}^2) + \alpha_{\max}(L_f^2 + r_{\max}^2 L_1^2). \tag{129}$$

*When the following conditions hold:*

$$2\eta_x^2 \bar{\tau} c_a \alpha_{\max} L_1^2 \le 1, \tag{130}$$

$$4\eta_y^2 c_a \bar{\tau} \alpha_{\max} L_1^2 \le 1, \tag{131}$$

*the upper bound (128c) on the drift of* $y_i^{(t,k)}$ *can be simplified as:*

$$\sum_{i=1}^{n} \frac{w_i}{\|a_i^{(t)}\|_1} \sum_{k=0}^{\tau_i - 1} a_i^{(t,k)} \mathbb{E} \left[ \left\| y_i^{(t,k)} - y^{(t)} \right\|^2 \right] \le 2\eta_y^2 \bar{\tau} \sigma_{M2}^2 + 4\eta_y^2 \bar{\tau} c_a \alpha_{\max} L_1^2 \mathbb{E} \left[ \left\| y^{(t)} - \widetilde{y}^*(x^{(t)}) \right\|^2 \right], \tag{132}$$

*where*

$$\sigma_{M2}^2 = \alpha_{\max}^2 \sigma_g^2 + \alpha_{\max}^2 (\sigma_f^2 + r_{\max}^2 \sigma_{gg}^2) + \alpha_{\max}(L_f^2 + r_{\max}^2 L_1^2) + 2 c_a \alpha_{\max} \sigma_{gh}^2. \tag{133}$$

This lemma reveals how client drift is influenced by local step sizes $\eta_x, \eta_y, \eta_z$, the number of local updates $\bar{\tau}$, gradient variances $\sigma_{M1}^2, \sigma_g^2$, and intrinsic system heterogeneity $\sigma_{gh}^2$. It clarifies the cumulative effect of drift and provides practical guidance on selecting appropriate step sizes and local iteration counts to balance computational efficiency and drift control.

**Lemma 16** (**Boundedness of Aggregated Gradient Drift**). *Under Assumptions 1–7, for each global iteration $t \in \{0, 1, \cdots, T-1\}$ and client $i \in \{1, 2, \cdots, n\}$, the upper bounds of the aggregated gradient drift $\Delta_f^{(t)}, \Delta_R^{(t)}, \Delta_g^{(t)}$ satisfy:*

$$
\begin{aligned}
\Delta_f^{(t)} = \Delta_R^{(t)} &:= \sum_{i=1}^n \frac{w_i}{\|a_i^{(t)}\|_1} \sum_{k=0}^{\tau_i-1} a_i^{(t,k)} \Delta_{f,i}^{(t,k)} \\
&= \sum_{i=1}^n \frac{w_i}{\|a_i^{(t)}\|_1} \sum_{k=0}^{\tau_i-1} a_i^{(t,k)} \Delta_{R,i}^{(t,k)} \\
&\le 3\eta_x^2 \bar{\tau}(L_1^2 + r^2 L_2^2)\sigma_{M1}^2 + 3\eta_z^2 \bar{\tau} L_1^2 \sigma_{M1}^2 + 6\eta_y^2 \bar{\tau}(L_1^2 + r^2 L_2^2)\sigma_{M2}^2 \\
&\quad + 12\eta_y^2 \bar{\tau}(L_1^2 + r^2 L_2^2)c_a \alpha_{\max} L_1^2 \mathbb{E}\left[\left\|y^{(t)} - \widetilde{y}^*(x^{(t)})\right\|^2\right],
\end{aligned}
\tag{134a}
$$

$$
\begin{aligned}
\Delta_g^{(t)} &:= \sum_{i=1}^n \frac{w_i}{\|a_i^{(t)}\|_1} \sum_{k=0}^{\tau_i-1} a_i^{(t,k)} \Delta_{g_i}^{(t,k)} \\
&\le \eta_x^2 \bar{\tau} L_1^2 \sigma_{M1}^2 + 2\eta_y^2 \bar{\tau} L_1^2 \sigma_{M2}^2 + 4\eta_y^2 \bar{\tau} c_a \alpha_{\max} L_1^4 \mathbb{E}\left[\left\|y^{(t)} - \widetilde{y}^*(x^{(t)})\right\|^2\right].
\end{aligned}
\tag{134b}
$$

*Proof.* By substituting the definition of $\Delta_{f,i}^{(t,k)}$ from (127a) into the weighted sum in $\Delta_f^{(t)}$, we obtain

$$
\begin{aligned}
\Delta_f^{(t)} &:= \sum_{i=1}^n \frac{w_i}{\|a_i^{(t)}\|_1} \sum_{k=0}^{\tau_i-1} a_i^{(t,k)} \Delta_{f,i}^{(t,k)} \\
&= \sum_{i=1}^n \frac{w_i}{\|a_i^{(t)}\|_1} \sum_{k=0}^{\tau_i-1} a_i^{(t,k)} \left(3(L_1^2 + r^2 L_2^2)[\|x_i^{(t,k)} - x^{(t)}\|^2 + \|y_i^{(t,k)} - y^{(t)}\|^2] + 3L_1^2\|z_i^{(t,k)} - z^{(t)}\|^2\right) \\
&= 3(L_1^2 + r^2 L_2^2) \sum_{i=1}^n \frac{w_i}{\|a_i^{(t)}\|_1} \sum_{k=0}^{\tau_i-1} a_i^{(t,k)}[\|x_i^{(t,k)} - x^{(t)}\|^2 + \|y_i^{(t,k)} - y^{(t)}\|^2] \\
&\quad + 3L_1^2 \sum_{i=1}^n \frac{w_i}{\|a_i^{(t)}\|_1} \sum_{k=0}^{\tau_i-1} a_i^{(t,k)}\|z_i^{(t,k)} - z^{(t)}\|^2 \\
&\overset{(a)}{\le} 3\eta_x^2 \bar{\tau}(L_1^2 + r^2 L_2^2)\sigma_{M1}^2 + 3\eta_z^2 \bar{\tau} L_1^2 \sigma_{M1}^2 + 6\eta_y^2 \bar{\tau}(L_1^2 + r^2 L_2^2)\sigma_{M2}^2 \\
&\quad + 12\eta_y^2 \bar{\tau}(L_1^2 + r^2 L_2^2)c_a \alpha_{\max} L_1^2 \mathbb{E}\left[\left\|y^{(t)} - \widetilde{y}^*(x^{(t)})\right\|^2\right],
\end{aligned}
\tag{135}
$$

where step $(a)$ uses inequalities (128a), (132), and (128b) from Lemma 15.

Similarly, substituting $\Delta_{g,i}^{(t,k)}$ from (127b) yields

$$
\begin{aligned}
\Delta_g^{(t)} &:= \sum_{i=1}^n \frac{w_i}{\|a_i^{(t)}\|_1} \sum_{k=0}^{\tau_i-1} a_i^{(t,k)} \Delta_{g_i}^{(t,k)} \\
&= L_1^2 \sum_{i=1}^n \frac{w_i}{\|a_i^{(t)}\|_1} \sum_{k=0}^{\tau_i-1} a_i^{(t,k)}[\|x_i^{(t,k)} - x^{(t)}\|^2 + \|y_i^{(t,k)} - y^{(t)}\|^2] \\
&\overset{(b)}{\le} \eta_x^2 \bar{\tau} L_1^2 \sigma_{M1}^2 + 2\eta_y^2 \bar{\tau} L_1^2 \sigma_{M2}^2 + 4\eta_y^2 \bar{\tau} c_a \alpha_{\max} L_1^4 \mathbb{E}\left[\left\|y^{(t)} - \widetilde{y}^*(x^{(t)})\right\|^2\right],
\end{aligned}
\tag{136}
$$

where step $(b)$ again uses inequalities (128a) and (132). $\qquad\square$

This lemma quantifies the influence of client drift on global gradient estimation. Notably, the lower-level optimization error term $\mathbb{E}\left[\left\|y^{(t)} - \widetilde{y}^*(x^{(t)})\right\|^2\right]$ affects all components of the aggregated gradients for variables $x, y, z$.

## C.7 Properties of Aggregated Gradients

This subsection introduces and analyzes the boundedness of the expected global aggregated gradients. The proof is given in (Yang et al., 2023). The results in this subsection will be used in Section C.9 to analyze the behavior of the Lyapunov function of Algorithm 1 during the global iteration process.

**Lemma 17** (**Boundedness of Global Aggregated Gradient Estimates** (Yang et al., 2023)). *Under Assumptions 1–8, for each global round $t \in \{0, 1, \cdots, T-1\}$, when the server randomly selects $P = |C^{(t)}|$ clients from $n$ total clients, the estimates of the global aggregated gradients for variables $x^{(t)}, y^{(t)}, z^{(t)}$ satisfy*

$$\mathbb{E}\left[\left\|\sum_{i \in C^{(t)}} \widetilde{w}_i h_{x,i}^{(t)}\right\|^2\right] \leq \frac{2n}{P} \sum_{i=1}^{n} \frac{w_i^2}{\|a_i^{(t)}\|_1^2} \sum_{k=0}^{\tau_i - 1} \left(a_i^{(t,k)}\right)^2 (\sigma_f^2 + r_i^2 \sigma_{gg}^2)$$

$$+ \frac{n(P-1)}{P(n-1)} \mathbb{E}\left[\left\|\sum_{i=1}^{n} w_i \widetilde{h}_{x,i}^{(t)}\right\|^2\right]$$

$$+ \frac{2n(n-P)}{P(n-1)} \sum_{i=1}^{n} \frac{w_i^2}{\|a_i^{(t)}\|_1} \sum_{k=0}^{\tau_i - 1} a_i^{(t,k)} \Delta_{f,i}^{(t,k)}$$

$$+ \frac{4(n-P)\beta_{\max}}{P(n-1)} (r_{\max}^2 L_1^2 + L_f^2), \tag{137a}$$

$$\mathbb{E}\left[\left\|\sum_{i \in C^{(t)}} \widetilde{w}_i h_{y,i}^{(t)}\right\|^2\right] \leq \frac{n}{P} \sum_{i=1}^{n} \frac{w_i^2}{\|a_i^{(t)}\|_1^2} \sum_{k=0}^{\tau_i - 1} \left(a_i^{(t,k)}\right)^2 \sigma_g^2 + \frac{2(n-P)\beta_{\max}}{P(n-1)} \sigma_{gh}^2$$

$$+ \left(\frac{2n(n-P)}{P(n-1)} \sum_{i=1}^{n} \frac{w_i^2}{\|a_i^{(t)}\|_1} \sum_{k=0}^{\tau_i - 1} a_i^{(t,k)} + 3 \sum_{i=1}^{n} \frac{w_i}{\|a_i^{(t)}\|_1} \sum_{k=0}^{\tau_i - 1} a_i^{(t,k)}\right) \Delta_{g,i}^{(t,k)}$$

$$+ \left(\frac{2(n-P)\beta_{\max}\beta_{gh}^2}{P(n-1)} + 3L_1^2\right) \mathbb{E}\left[\left\|y^{(t)} - \widetilde{y}^*(x^{(t)})\right\|^2\right], \tag{137b}$$

$$\mathbb{E}\left[\left\|\sum_{i \in C^{(t)}} \widetilde{w}_i h_{z,i}^{(t)}\right\|^2\right] \leq \frac{2n}{P} \sum_{i=1}^{n} \frac{w_i^2}{\|a_i^{(t)}\|_1^2} \sum_{k=0}^{\tau_i - 1} \left(a_i^{(t,k)}\right)^2 (\sigma_f^2 + r_i^2 \sigma_{gg}^2)$$

$$+ \frac{4(n-P)\beta_{\max}}{P(n-1)} (r_{\max}^2 L_1^2 + L_f^2)$$

$$+ \left(\frac{2n(n-P)}{P(n-1)} \sum_{i=1}^{n} w_i^2 \sum_{k=0}^{\tau_i - 1} \frac{a_i^{(t,k)}}{\|a_i^{(t)}\|_1} + 3 \sum_{i=1}^{n} w_i \sum_{k=0}^{\tau_i - 1} \frac{a_i^{(t,k)}}{\|a_i^{(t)}\|_1}\right) \Delta_{R,i}^{(t,k)}$$

$$+ 3L_1^2 \mathbb{E}\left[\left\|z^{(t)} - z^*(x^{(t)})\right\|^2\right]. \tag{137c}$$

This lemma shows that the global aggregated gradient estimate for variable $x^{(t)}$ depends on the gradient drift $\Delta_{f,i}^{(t,k)}$, the variance of $f_i$'s gradient $\sigma_f^2$, and the second-order derivative variance of $g_i$, $\sigma_{gg}^2$, and is also affected by the expected norm of the global gradient estimate. The estimate for variable $y^{(t)}$ depends on the gradient drift $\Delta_{g,i}^{(t,k)}$ and gradient variance $\sigma_g^2$, as well as the deviation between the current iterate $y^{(t)}$ and the optimal value $y^*(x^{(t)})$, measured by $\mathbb{E}\left[\left\|y^{(t)} - \widetilde{y}^*(x^{(t)})\right\|^2\right]$. Similarly, the estimate for variable $z^{(t)}$ depends on the gradient drift $\Delta_{R,i}^{(t,k)}$, gradient variance $\sigma_f^2$, and second-order variance $\sigma_{gg}^2$, as well as the deviation between $z^{(t)}$ and the optimal value $z^*(x^{(t)})$, measured by $\mathbb{E}\left[\left\|z^{(t)} - z^*(x^{(t)})\right\|^2\right]$.

## C.8 Descent in the Objective Function

Building upon the work in (Yang et al., 2023), this subsection analyzes the expected descent behavior of the global optimization objective across global iterations and explores the factors that influence the descent of the objective function. The results in this subsection will be used in Section C.9 to analyze the behavior of the Lyapunov function of Algorithm 1 during global iterations.

**Lemma 18** (**Descent of the Global Upper-Level Objective Function** (Yang et al., 2023)). *Under Assumptions 1–8, for each global iteration $t \in \{0, 1, \cdots, T-1\}$, the expectation difference between consecutive iterations of the smooth and non-convex global upper-level objective function $\widetilde{\Phi}(x)$ satisfies*

$$
\mathbb{E}\left[\widetilde{\Phi}(x^{(t+1)})\right] - \mathbb{E}\left[\widetilde{\Phi}(x^{(t)})\right]
$$

$$
\leq -\frac{\rho^{(t)}\gamma_x^{(t)}}{2}\left(\mathbb{E}\left[\left\|\nabla\widetilde{\Phi}(x^{(t)})\right\|^2\right] + \mathbb{E}\left[\left\|\sum_{i=1}^{n} w_i \widetilde{h}_{x,i}^{(t)}\right\|^2\right]\right)
$$

$$
+ \frac{\rho^{(t)}\gamma_x^{(t)}}{2}\left(6(L_1^2 + r^2 L_2^2)\mathbb{E}\left[\left\|y^{(t)} - \widetilde{y}^*(x^{(t)})\right\|^2\right] + 3L_1^2\mathbb{E}\left[\left\|z^{(t)} - \widetilde{z}^*(x^{(t)})\right\|^2\right]\right)
$$

$$
+ \frac{3\rho^{(t)}\gamma_x^{(t)}}{2}\Delta_f^{(t)}
$$

$$
+ \frac{L_{\widetilde{\Phi}}^2(\rho^{(t)}\gamma_x^{(t)})^2}{2}\mathbb{E}\left[\left\|\sum_{i\in C^{(t)}} \widetilde{w}_i h_{x,i}^{(t)}\right\|^2\right]. \tag{138}
$$

The first term on the right-hand side of inequality (138) (the **descent term**),

$$
-\frac{\rho^{(t)}\gamma_x^{(t)}}{2}\left(\mathbb{E}\left[\left\|\nabla\widetilde{\Phi}(x^{(t)})\right\|^2\right] + \mathbb{E}\left[\left\|\sum_{i=1}^{n} w_i \widetilde{h}_{x,i}^{(t)}\right\|^2\right]\right), \tag{139}
$$

characterizes the decreasing tendency of the objective function and is therefore negative. Here, the expected squared norm $\mathbb{E}\left[\|\nabla\widetilde{\Phi}(x^{(t)})\|^2\right]$ reflects the strength of the optimization direction. The term $\mathbb{E}\left[\left\|\sum_{i=1}^{n} w_i \widetilde{h}_{x,i}^{(t)}\right\|^2\right]$ captures the contribution of the aggregated gradient across all clients. The product $\rho^{(t)}\gamma_x^{(t)}$ controls the magnitude of the descent.

The second term on the right-hand side (the **variable deviation term**),

$$
\frac{\rho^{(t)}\gamma_x^{(t)}}{2}\left(6(L_1^2 + r^2 L_2^2)\mathbb{E}\left[\left\|y^{(t)} - \widetilde{y}^*(x^{(t)})\right\|^2\right] + 3L_1^2\mathbb{E}\left[\left\|z^{(t)} - \widetilde{z}^*(x^{(t)})\right\|^2\right]\right), \tag{140}
$$

quantifies the interference from the deviation of the lower-level variable $y^{(t)}$ and the auxiliary variable $z^{(t)}$ from their respective optimal values $\widetilde{y}^*(x^{(t)})$ and $\widetilde{z}^*(x^{(t)})$. This term suggests that larger deviations weaken the descent of the objective.

The third term (the **client drift term**),

$$
\frac{3\rho^{(t)}\gamma_x^{(t)}}{2}\Delta_f^{(t)} = \frac{3\rho^{(t)}\gamma_x^{(t)}}{2}\sum_{i=1}^{n}\frac{w_i}{\|a_i^{(t)}\|_1}\sum_{k=0}^{\tau_i-1} a_i^{(t,k)}\Delta_{f,i}^{(t,k)}
$$

$$
\leq \frac{3\rho^{(t)}\gamma_x^{(t)}}{2}\left(3\eta_x^2\bar{\tau}(L_1^2 + r^2 L_2^2)\sigma_{M1}^2 + 3\eta_z^2\bar{\tau}L_1^2\sigma_{M1}^2\right.
$$

$$
\left. + 6\eta_y^2\bar{\tau}(L_1^2 + r^2 L_2^2)\sigma_{M2}^2 + 12\eta_y^2\bar{\tau}(L_1^2 + r^2 L_2^2)c_a\alpha_{\max}L_1^2\mathbb{E}\left[\left\|y^{(t)} - \widetilde{y}^*(x^{(t)})\right\|^2\right]\right), \tag{141}
$$

quantifies the effect of client drift, i.e., the deviation between local updates and the global target. This term reflects that local optimization may lead to global deviation, especially under high data heterogeneity.

The fourth term (the **aggregation error term**),

$$\frac{L_\Phi^2(\rho^{(t)}\gamma_x^{(t)})^2}{2}\mathbb{E}\left[\left\|\sum_{i\in C^{(t)}}\widetilde{w}_i h_{x,i}^{(t)}\right\|^2\right], \tag{142}$$

captures the influence of the norm of the aggregated gradient estimator and the smoothness of the objective function (represented by $L_\Phi$). This term grows quadratically with the step size, indicating that excessively large steps can amplify the error.

**Lemma 19 (Descent of the Global Lower-Level Objective Function** (Yang et al., 2023)**).** *Under Assumptions 1–8, the mean squared error between the global lower-level solution $y^{(t)}$ and its optimal value $\widetilde{y}^*(x^{(t)})$ satisfies the following inequality at each global iteration $t \in \{0, 1, \cdots, T-1\}$*

$$\mathbb{E}\left[\left\|y^{(t+1)}-\widetilde{y}^*(x^{(t+1)})\right\|^2\right]-\mathbb{E}\left[\left\|y^{(t)}-\widetilde{y}^*(x^{(t)})\right\|^2\right]$$

$$\leq (\delta_y^{(t)}-\rho^{(t)}\gamma_y^{(t)}\mu_g-\delta_y^{(t)}\rho^{(t)}\gamma_y^{(t)}\mu_g)\mathbb{E}\left[\left\|y^{(t)}-\widetilde{y}^*(x^{(t)})\right\|^2\right]$$

$$+ (1+\delta_y^{(t)})(\rho^{(t)}\gamma_y^{(t)})^2\mathbb{E}\left[\left\|\sum_{i\in C^{(t)}}\widetilde{w}_i h_{y,i}^{(t)}\right\|^2\right]$$

$$+ (1+\delta_y^{(t)})\rho^{(t)}\gamma_y^{(t)}\frac{2L_1^2}{\mu_g}\sum_{i=1}^n\frac{w_i}{\|a_i^{(t)}\|_1}\sum_{k=0}^{\tau_i-1}a_i^{(t,k)}\mathbb{E}\left[\left\|x^{(t)}-x_i^{(t,k)}\right\|^2+\left\|y^{(t)}-y_i^{(t,k)}\right\|^2\right]$$

$$+ \left(\rho^{(t)}\gamma_x^{(t)}\right)^2\left(L_y^2+\frac{L_{yx}}{2}\right)\mathbb{E}\left[\left\|\sum_{i\in C^{(t)}}\widetilde{w}_i h_{x,i}^{(t)}\right\|^2\right]$$

$$+ (\rho^{(t)}\gamma_x^{(t)})^2\frac{L_y^2}{\delta_{y,1}^{(t)}}\mathbb{E}\left[\left\|\sum_{i=1}^n w_i\widetilde{h}_{x,i}^{(t)}\right\|^2\right], \tag{143}$$

*where $\delta_{y,1}^{(t)}>0$ is the coefficient from Young's inequality:*

$$2\mathbb{E}\left[\left\|y^{(t+1)}-\widetilde{y}^*(x^{(t)})\right\|^2\right]\cdot(\rho^{(t)}\gamma_x^{(t)}L_y)\mathbb{E}\left[\left\|\sum_{i=1}^n w_i\widetilde{h}_{x,i}^{(t)}\right\|^2\right]$$

$$\leq \delta_{y,1}^{(t)}\mathbb{E}\left[\left\|y^{(t+1)}-\widetilde{y}^*(x^{(t)})\right\|^2\right]+(\rho^{(t)}\gamma_x^{(t)})^2\frac{L_y^2}{\delta_{y,1}^{(t)}}\mathbb{E}\left[\left\|\sum_{i=1}^n w_i\widetilde{h}_{x,i}^{(t)}\right\|^2\right], \tag{144}$$

*and $\delta_y^{(t)}$ is defined as*

$$\delta_y^{(t)} := \delta_{y,1}^{(t)}+\frac{L_{yx}(r_{\max}^2 L_1^2+L_f^2)(\rho^{(t)}\gamma_x^{(t)})^2}{2}. \tag{145}$$

The first term on the right-hand side of inequality (143) (the **contraction term**),

$$(\delta_y^{(t)}-\rho^{(t)}\gamma_y^{(t)}\mu_g-\delta_y^{(t)}\rho^{(t)}\gamma_y^{(t)}\mu_g)\mathbb{E}\left[\left\|y^{(t)}-\widetilde{y}^*(x^{(t)})\right\|^2\right], \tag{146}$$

captures the decreasing trend of the objective function. This term becomes negative when $\delta_y^{(t)} < \rho^{(t)}\gamma_y^{(t)}\mu_g/(1 + \rho^{(t)}\gamma_y^{(t)}\mu_g)$, indicating convergence of the error.

The second term (the **aggregated gradient error term**),

$$(1 + \delta_y^{(t)})(\rho^{(t)}\gamma_y^{(t)})^2 \mathbb{E}\left[\left\|\sum_{i \in C^{(t)}} \widetilde{w}_i h_{y,i}^{(t)}\right\|^2\right], \tag{147}$$

reflects the gradient estimation error from the randomly selected clients. An excessively large step size $\rho^{(t)}\gamma_y^{(t)}$ can amplify this error.

The third term (the **client drift term**),

$$(1 + \delta_y^{(t)})\rho^{(t)}\gamma_y^{(t)}\frac{2L_1^2}{\mu_g}\sum_{i=1}^{n}\frac{w_i}{\|a_i^{(t)}\|_1}\sum_{k=0}^{\tau_i-1}a_i^{(t,k)}\mathbb{E}\left[\left\|x^{(t)} - x_i^{(t,k)}\right\|^2 + \left\|y^{(t)} - y_i^{(t,k)}\right\|^2\right]$$

$$\overset{(a)}{\leq} (1 + \delta_y^{(t)})\rho^{(t)}\gamma_y^{(t)}\frac{2L_1^2}{\mu_g}\left(\eta_x^2\bar{\tau}\sigma_{M1}^2 + 2\eta_y^2\bar{\tau}\sigma_{M2}^2 + 4\eta_y^2\bar{\tau}c_a\alpha_{\max}L_1^2\mathbb{E}\left[\left\|y^{(t)} - \widetilde{y}^*(x^{(t)})\right\|^2\right]\right), \tag{148}$$

quantifies the deviation between local and global variables. Here, $\tau_i$ denotes the number of local iterations on client $i$, and the deviation accumulates as local updates progress. A large update step size $\rho^{(t)}\gamma_y^{(t)}$ increases the error. Inequality $(a)$ follows from (128a) and (132).

The fourth term (the **influence of upper-level updates**),

$$\left(\rho^{(t)}\gamma_x^{(t)}\right)^2\left(L_y^2 + \frac{L_{yx}}{2}\right)\mathbb{E}\left[\left\|\sum_{i \in C^{(t)}}\widetilde{w}_i h_{x,i}^{(t)}\right\|^2\right], \tag{149}$$

represents how updates to the upper-level variable $x^{(t)}$ affect the convergence of the lower level. Again, large update steps amplify this error.

The fifth term (the **global hypergradient estimation error term**),

$$\left(\rho^{(t)}\gamma_x^{(t)}\right)^2\frac{L_y^2}{\delta_{y,1}^{(t)}}\mathbb{E}\left[\left\|\sum_{i=1}^{n}w_i\widetilde{h}_{x,i}^{(t)}\right\|^2\right], \tag{150}$$

quantifies the influence of global gradient estimation errors with respect to the upper-level variable $x$ on the lower-level variable $z^{(t)}$. A larger upper-level step size $\rho^{(t)}\gamma_y^{(t)}$ also enlarges this term.

**Lemma 20 (Descent of the Global Linear System Objective Function** (Yang et al., 2023)**).** *Under Assumptions 1–8, the mean squared error between the solution $z^{(t)}$ of the global linear system and its optimal solution $\widetilde{z}^*(x^{(t)})$ satisfies the following relation in each global iteration round $t \in \{0, 1, \cdots, T-1\}$*

$$\mathbb{E}\left[\left\|z^{(t+1)} - \widetilde{z}^*(x^{(t+1)})\right\|^2\right] - \mathbb{E}\left[\left\|z^{(t)} - \widetilde{z}^*(x^{(t)})\right\|^2\right]$$

$$\leq (\delta_z^{(t)} - \rho^{(t)}\gamma_z^{(t)}\mu_g - \delta_z^{(t)}\rho^{(t)}\gamma_z^{(t)}\mu_g)\mathbb{E}\left[\left\|z^{(t)} - \widetilde{z}^*(x^{(t)})\right\|^2\right]$$

$$+ (1 + \delta_z^{(t)})(\rho^{(t)}\gamma_z^{(t)})^2\mathbb{E}\left[\left\|\sum_{i \in C^{(t)}}\widetilde{w}_i h_{z,i}^{(t)}\right\|^2\right]$$

$$+ (1 + \delta_z^{(t)})\rho^{(t)}\gamma_z^{(t)}\frac{4L_R^2}{\mu_g}\sum_{i=1}^{n}\frac{w_i}{\|a_i^{(t)}\|_1}\sum_{k=0}^{\tau_i-1}a_i^{(t,k)}\mathbb{E}\left[\left\|x^{(t)} - x_i^{(t,k)}\right\|^2 + \left\|y^{(t)} - y_i^{(t,k)}\right\|^2\right]$$

$$
\begin{aligned}
&+ \left\| z^{(t)} - z_i^{(t,k)} \right\|^2 \Big] \\
&+ (1 + \delta_z^{(t)}) \rho^{(t)} \gamma_z^{(t)} \frac{4L_R^2}{\mu_g} \mathbb{E} \left[ \left\| y^{(t)} - \widetilde{y}^*(x^{(t)}) \right\|^2 \right] \\
&+ (\rho^{(t)} \gamma_x^{(t)})^2 \left( L_z^2 + \frac{L_{zx}}{2} \right) \mathbb{E} \left[ \left\| \sum_{i \in C^{(t)}} \widetilde{w}_i h_{x,i}^{(t)} \right\|^2 \right] \\
&+ (\rho^{(t)} \gamma_x^{(t)})^2 \frac{L_z^2}{\delta_{z,1}^{(t)}} \mathbb{E} \left[ \left\| \sum_{i=1}^n w_i \widetilde{h}_{x,i}^{(t)} \right\|^2 \right],
\end{aligned}
\tag{151}
$$

where $\delta_{z,1}^{(t)} > 0$ is the coefficient in the Young's inequality

$$
2\mathbb{E} \left[ \left\| z^{(t+1)} - \widetilde{z}^*(x^{(t)}) \right\| \right] \cdot (\rho^{(t)} \gamma_x^{(t)} L_z) \mathbb{E} \left[ \left\| \sum_{i=1}^n w_i \widetilde{h}_{x,i}^{(t)} \right\| \right]
$$
$$
\leq \delta_{z,1}^{(t)} \mathbb{E} \left[ \left\| z^{(t+1)} - \widetilde{z}^*(x^{(t)}) \right\| \right] + (\rho^{(t)} \gamma_x^{(t)})^2 \frac{L_z^2}{\delta_{z,1}^{(t)}} \mathbb{E} \left[ \left\| \sum_{i=1}^n w_i \widetilde{h}_{x,i}^{(t)} \right\|^2 \right],
\tag{152}
$$

and $\delta_z^{(t)}$ is defined as

$$
\delta_z^{(t)} := \delta_{z,1}^{(t)} + \frac{L_{zx}(r_{\max}^2 L_1^2 + L_f^2)(\rho^{(t)} \gamma_x^{(t)})^2}{2}.
\tag{153}
$$

The first term on the right-hand side of inequality (151) (**Contraction Term**):

$$
(\delta_z^{(t)} - \rho^{(t)} \gamma_z^{(t)} \mu_g - \delta_z^{(t)} \rho^{(t)} \gamma_z^{(t)} \mu_g) \mathbb{E} \left[ \left\| z^{(t)} - \widetilde{z}^*(x^{(t)}) \right\|^2 \right],
\tag{154}
$$

describes the variation trend of the current error $\mathbb{E}\big[\left\| z^{(t)} - \widetilde{z}^*(x^{(t)}) \right\|^2\big]$. When $\delta_z^{(t)} < \frac{\rho^{(t)} \gamma_z^{(t)} \mu_g}{1 + \rho^{(t)} \gamma_z^{(t)} \mu_g}$, the coefficient is negative, indicating that the error decreases with iterations, thereby driving $z^{(t)}$ toward $\widetilde{z}^*(x^{(t)})$.

The second term on the right-hand side of inequality (151) (**Gradient Aggregation Error Term**):

$$
(1 + \delta_z^{(t)})(\rho^{(t)} \gamma_z^{(t)})^2 \mathbb{E} \left[ \left\| \sum_{i \in C^{(t)}} \widetilde{w}_i h_{z,i}^{(t)} \right\|^2 \right],
\tag{155}
$$

quantifies the gradient estimation error of the auxiliary variable $z$ caused by the randomly selected client subset $C^{(t)}$. An excessively large update step $\rho^{(t)} \gamma_z^{(t)}$ for $z^{(t)}$ will amplify this error.

The third term on the right-hand side of inequality (151) (**Client Drift Term**):

$$
\begin{aligned}
&(1 + \delta_z^{(t)}) \rho^{(t)} \gamma_z^{(t)} \frac{4L_R^2}{\mu_g} \sum_{i=1}^n \frac{w_i}{\|a_i^{(t)}\|_1} \sum_{k=0}^{\tau_i - 1} a_i^{(t,k)} \mathbb{E} \left[ \left\| x^{(t)} - x_i^{(t,k)} \right\|^2 + \left\| y^{(t)} - y_i^{(t,k)} \right\|^2 + \left\| z^{(t)} - z_i^{(t,k)} \right\|^2 \right] \\
&\overset{(a)}{\leq} (1 + \delta_z^{(t)}) \rho^{(t)} \gamma_z^{(t)} \frac{4L_R^2}{\mu_g} \left( 2\eta_x^2 \bar{\tau} \sigma_{M1}^2 + 2\eta_y^2 \bar{\tau} \sigma_{M2}^2 + 4\eta_y^2 \bar{\tau} c_a \alpha_{\max} L_1^2 \mathbb{E} \left[ \left\| y^{(t)} - \widetilde{y}^*(x^{(t)}) \right\|^2 \right] \right),
\end{aligned}
\tag{156}
$$

reflects the impact of the deviation between local variables $(x_i^{(t,k)}, y_i^{(t,k)}, z_i^{(t,k)})$ and global variables $(x^{(t)}, y^{(t)}, z^{(t)})$ on the convergence of $z^{(t)}$. Here, $\tau_i$ denotes the number of local iterations on client $i$, and the deviation accumulates with increasing local iterations. A large step size $\rho^{(t)} \gamma_z^{(t)}$ will amplify this error. Step $(a)$ applies inequalities (128a), (128b), and (132).

The fourth term on the right-hand side of inequality (151) (**Lower-Level Variable Bias Term**):

$$(1 + \delta_z^{(t)})\rho^{(t)}\gamma_z^{(t)}\frac{4L_R^2}{\mu_g}\mathbb{E}\left[\left\|y^{(t)} - \widetilde{y}^*(x^{(t)})\right\|^2\right], \tag{157}$$

quantifies the impact of the deviation between the lower-level variable $y^{(t)}$ and its optimal solution $\widetilde{y}^*(x^{(t)})$ on the convergence of $z^{(t)}$. This reveals that instability in lower-level optimization can interfere with the update of the auxiliary variable. An excessively large step size $\rho^{(t)}\gamma_z^{(t)}$ can amplify this error.

The fifth term on the right-hand side of inequality (151) (**Upper-Level Update Interference Term**):

$$\left(\rho^{(t)}\gamma_x^{(t)}\right)^2\left(L_z^2 + \frac{L_{zx}}{2}\right)\mathbb{E}\left[\left\|\sum_{i \in C^{(t)}}\widetilde{w}_i h_{x,i}^{(t)}\right\|^2\right], \tag{158}$$

captures the interference effect of the upper-level variable $x^{(t)}$'s update on the convergence of $z^{(t)}$. This error is amplified if the step size $\rho^{(t)}\gamma_x^{(t)}$ is too large.

The sixth term on the right-hand side of inequality (151) (**Global Gradient Estimation Error Term**):

$$\left(\rho^{(t)}\gamma_x^{(t)}\right)^2\frac{L_z^2}{\delta_{z,1}^{(t)}}\mathbb{E}\left[\left\|\sum_{i=1}^n w_i\widetilde{h}_{x,i}^{(t)}\right\|^2\right], \tag{159}$$

quantifies the influence of the global gradient estimation error for the upper-level variable $x$ from all clients on the convergence of $z^{(t)}$. Again, a large update step $\rho^{(t)}\gamma_x^{(t)}$ will amplify this error.

### C.9  Descent of the Lyapunov Function

In this subsection, we analyze and prove the driving and resisting forces behind the descent of the Lyapunov function associated with Algorithm 1 during the global iteration process.

**Lemma 21** (**Descent of the Lyapunov Function**). *Under Assumptions 1–9, suppose that the global learning rates satisfy*

$$\gamma_x^{(t)} = \mathcal{O}\left(\sqrt{\frac{P}{\bar{\tau}T}}\right), \tag{160a}$$

$$\gamma_{y,\max} = c_{\gamma_y}\gamma_{x,\min}, \tag{160b}$$

$$\gamma_{z,\max} = c_{\gamma_z}\gamma_{x,\min}, \tag{160c}$$

*and the constraint introduced in (Yang et al., 2023) holds*

$$\rho^{(t)}\gamma_x^{(t)} \leq \rho^{(t)}\gamma_{x,\max}$$

$$\leq \min\left\{\frac{\mu_g}{12L_1^2c_{\gamma_z}}, \ \frac{\mu_g}{36L_1^2c_{\gamma_z}}\left(\frac{2\beta_{\max}}{P} + 3\right)^{-1}, \ \frac{P\mu_g c_{\gamma_z}}{36L_1^2\beta_{\max}}\left(L_z^2 + \frac{L_{zx}}{2}\right)^{-1},\right.$$

$$\frac{P}{6L_\Phi^2\beta_{\max}}, \ \frac{4}{L_\Phi^2}, \ \frac{4}{c_{\gamma_y}}, \ \frac{4}{c_{\gamma_z}}, \ \frac{\mu_g}{12c_{\gamma_y}}\left[\left(\frac{2\beta_{\max}}{P} + 3\right)L_1^2 + \frac{2\beta_{\max}\beta_{gh}^2}{P} + 3L_1^2\right]^{-1},$$

$$\frac{Pc_{\gamma_y}\mu_g}{36\beta_{\max}(L_1^2 + r^2L_2^2)}\left(L_z^2 + \frac{L_{yx}}{2}\right)^{-1}, \ \frac{c_{\gamma_y}\mu_g}{4L_{yx}(L_f^2 + r_{\max}^2L_1^2)},$$

$$\left.\frac{c_{\gamma_z}\mu_g}{4L_{zx}(L_f^2 + r_{\max}^2L_1^2)}, \ \frac{1}{8}\left(K_y(L_y^2 + \frac{L_{yx}}{2}) + K_z(L_z^2 + \frac{L_{zx}}{2})\right)^{-1}\right\}, \tag{161}$$

and the local learning rates satisfy

$$\eta_x = \mathcal{O}\left(\frac{1}{\bar{\tau}\sqrt{T}}\right), \tag{162a}$$

$$\eta_y = \mathcal{O}\left(\frac{1}{\bar{\tau}\sqrt{T}}\right), \tag{162b}$$

$$\eta_z = \mathcal{O}\left(\frac{1}{\bar{\tau}\sqrt{T}}\right), \tag{162c}$$

and the constraints introduced in inequality (130), i.e.,

$$\eta_x^2 \leq \frac{1}{2\bar{\tau}c_a\alpha_{\max}L_1^2}, \tag{163a}$$

$$\eta_y^2 \leq \frac{1}{4\bar{\tau}c_a\alpha_{\max}L_1^2}. \tag{163b}$$

Consider the Lyapunov function defined as

$$\Psi(x^{(t)}) := \mathbb{E}\left[\widetilde{\Phi}(x^{(t)})\right] + K_y\mathbb{E}\left[\left\|y^{(t)} - \widetilde{y}^*(x^{(t)})\right\|^2\right] + K_z\mathbb{E}\left[\left\|z^{(t)} - \widetilde{z}^*(x^{(t)})\right\|^2\right] \tag{164}$$

with coefficients $K_y$ and $K_z$ set as:

$$K_y = \left[\frac{40(L_1^2 + r^2L_2^2)}{\mu_g} + \frac{384L_R^2L_1^2}{\mu_g^3}\right]\frac{1}{c_{\gamma_y}}, \tag{165a}$$

$$K_z = \frac{6L_1^2}{\mu_g c_{\gamma_z}}, \tag{165b}$$

and parameters $\delta_y^{(t)}$ and $\delta_z^{(t)}$, defined in equations (145) and (153), are set as:

$$\delta_y^{(t)} = \frac{\rho^{(t)}\gamma_y^{(t)}\mu_g}{4}, \tag{166a}$$

$$\delta_z^{(t)} = \frac{\rho^{(t)}\gamma_z^{(t)}\mu_g}{4}, \tag{166b}$$

and constants $\bar{\tau}$ and $\bar{\rho}$ are defined as

$$\bar{\tau} := \frac{1}{n}\sum_{i=1}^{n}\tau_i, \tag{167a}$$

$$\bar{\rho} := \frac{1}{T}\sum_{t=0}^{T-1}\rho^{(t)}. \tag{167b}$$

Then, for each global iteration round $t \in \{0, 1, \cdots, T-1\}$, the Lyapunov function satisfies

$$\Psi(x^{(t+1)}) - \Psi(x^{(t)}) \leq -\frac{\rho^{(t)}\gamma_x}{2}\mathbb{E}\left[\left\|\nabla\widetilde{\Phi}(x^{(t)})\right\|^2\right] + \epsilon_{sync}^{(t)} + \epsilon_{part}^{(t)} + \epsilon_{drift}^{(t)}, \tag{168}$$

where the error terms $\epsilon_{sync}^{(t)}, \epsilon_{part}^{(t)}, \epsilon_{drift}^{(t)}$ are defined as

$$\epsilon_{sync}^{(t)} := (\rho^{(t)}\gamma_x^{(t)})^2\left[L_\Phi^2 + 2K_y\left(L_y^2 + \frac{L_{yx}}{2}\right)\right.$$

$$\left. + 2K_z\left(L_z^2 + \frac{L_{zx}}{2}\right) + 4K_zc_{\gamma_z}^2\right]\frac{n}{P}\sum_{i=1}^{n}\frac{w_i^2\|a_i^{(t)}\|_2^2}{\|a_i^{(t)}\|_1^2}(\sigma_f^2 + r_i^2\sigma_{gg}^2)$$

$$+ (\rho^{(t)}\gamma_x^{(t)})^2 K_y c_{\gamma_y}^2 \frac{n}{P} \sum_{i=1}^{n} \frac{w_i^2 \|a_i^{(t)}\|_2^2}{\|a_i^{(t)}\|_1^2} \sigma_g^2, \tag{169a}$$

$$\epsilon_{part}^{(t)} := 2(\rho^{(t)}\gamma_x^{(t)})^2 \left[ L_\Phi^2 + 2K_y \left( L_y^2 + \frac{L_{yx}}{2} \right) \right.$$

$$\left. + 2K_z \left( L_z^2 + \frac{L_{zx}}{2} \right) + 4K_z c_{\gamma_z}^2 \right] \frac{(n-P)\beta_{\max}}{P(n-1)} (L_f^2 + r_{\max}^2 L_1^2)$$

$$+ 2(\rho^{(t)}\gamma_x^{(t)})^2 K_y c_{\gamma_y}^2 \frac{(n-P)\beta_{\max}}{P(n-1)} \sigma_{gh}^2, \tag{169b}$$

$$\epsilon_{drift}^{(t)} := 3(\rho^{(t)}\gamma_x^{(t)}) \left[ \frac{3}{2} + 24K_z c_{\gamma_z} \right.$$

$$\left. + \frac{\beta_{\max}}{P} \left( \frac{17}{4} + 16K_z c_{\gamma_z} \right) \right] (L_1^2 + r^2 L_2^2) \left( (\eta_x^2 + \eta_z^2)\bar{\tau}\sigma_{M1}^2 + 2\eta_y^2\bar{\tau}\sigma_{M2}^2 \right)$$

$$+ 4(\rho^{(t)}\gamma_x^{(t)}) K_y \left[ c_{\gamma_y} \frac{L_1^2}{\mu_g} + 2c_{\gamma_z} \left( \frac{2\beta_{\max}}{P} + 3 \right) L_1^2 \right] \left( \eta_x^2\bar{\tau}\sigma_{M1}^2 + 2\eta_y^2\bar{\tau}\sigma_{M2}^2 \right)$$

$$+ 4(\rho^{(t)}\gamma_x^{(t)}) K_z c_{\gamma_z} \frac{L_R^2}{\mu_g} \left( (\eta_x^2 + \eta_z^2)\bar{\tau}\sigma_{M1}^2 + 2\eta_y^2\bar{\tau}\sigma_{M2}^2 \right). \tag{169c}$$

*Proof.* From the definition of the Lyapunov function in (164), we have:

$$\Psi(x^{(t+1)}) - \Psi(x^{(t)})$$

$$= \mathbb{E}\left[ \widetilde{\Phi}(x^{(t+1)}) \right] - \mathbb{E}\left[ \widetilde{\Phi}(x^{(t)}) \right]$$

$$+ K_y \left( \mathbb{E}\left[ \left\| y^{(t+1)} - \widetilde{y}^*(x^{(t+1)}) \right\|^2 \right] - \mathbb{E}\left[ \left\| y^{(t)} - \widetilde{y}^*(x^{(t)}) \right\|^2 \right] \right)$$

$$+ K_z \left( \mathbb{E}\left[ \left\| z^{(t+1)} - \widetilde{z}^*(x^{(t+1)}) \right\|^2 \right] - \mathbb{E}\left[ \left\| z^{(t)} - \widetilde{z}^*(x^{(t)}) \right\|^2 \right] \right)$$

$$\overset{(a)}{\leq} -\frac{\rho^{(t)}\gamma_x^{(t)}}{2} \mathbb{E}\left[ \left\| \nabla\widetilde{\Phi}(x^{(t)}) \right\|^2 \right]$$

$$+ \epsilon_{\text{sync}}^{(t)} + \epsilon_{\text{part}}^{(t)} + \epsilon_{\text{drift}}^{(t)}.$$

In step $(a)$, we substitute inequalities (138), (143), and (151) into the Lyapunov expression (164), and then expand the corresponding terms using inequalities (137), (128), and (134).

In step $(b)$, combining Assumption 9 with the upper bounds on global learning rates in (160b) and (160c), we obtain:

$$\gamma_y^{(t)} \leq \gamma_{y,\max} = c_{\gamma_y}\gamma_{x,\min} \leq c_{\gamma_y}\gamma_x^{(t)}, \tag{170a}$$

$$\gamma_z^{(t)} \leq \gamma_{z,\max} = c_{\gamma_z}\gamma_{x,\min} \leq c_{\gamma_z}\gamma_x^{(t)}, \tag{170b}$$

which implies:

$$\rho^{(t)}\gamma_y^{(t)} \leq c_{\gamma_y} \cdot \rho^{(t)}\gamma_x^{(t)}, \tag{171a}$$

$$(\rho^{(t)}\gamma_y^{(t)})^2 \leq c_{\gamma_y}^2 \cdot (\rho^{(t)}\gamma_x^{(t)})^2, \tag{171b}$$

$$\rho^{(t)}\gamma_z^{(t)} \leq c_{\gamma_z} \cdot \rho^{(t)}\gamma_x^{(t)}, \tag{171c}$$

$$(\rho^{(t)}\gamma_z^{(t)})^2 \leq c_{\gamma_z}^2 \cdot (\rho^{(t)}\gamma_x^{(t)})^2. \tag{171d}$$

Combining all the terms and reorganizing leads to inequality (168).

This completes the proof of Lemma 21. $\qquad\square$

In this lemma, the full synchronization error $\epsilon_{\text{sync}}^{(t)}$ reflects the noise introduced by stochastic updates on clients. It is affected by the client sampling ratio $P/n$, and remains nonzero even under full participation $P = n$, depending on the noise levels $\{\sigma_f^2, \sigma_{gg}^2, \sigma_g^2\}$.

The partial participation error $\epsilon_{\text{part}}^{(t)}$ arises from the limited participation of clients ($P < n$). It vanishes when $P = n$, and decreases as $P$ increases.

The client drift error $\epsilon_{\text{drift}}^{(t)}$ captures the deviation of local variables due to data heterogeneity and local updates. It is proportional to the client step sizes $\{\eta_x, \eta_y, \eta_z\}$ and the average number of local iterations $\bar{\tau}$, and is affected by the noise levels $\sigma_{M1}^2$ and $\sigma_{M2}^2$.

## C.10 Convergence Analysis of the Algorithm

This subsection establishes the convergence of Algorithm 1. We then analyze how system heterogeneity and partial client participation impact its convergence and performance.

In the absence of system heterogeneity, i.e., when for every global iteration round $t \in \{0, 1, \cdots, T-1\}$ and every client $i \in \{1, 2, \cdots, n\}$, the local gradient aggregation coefficient vectors $a_i^{(t)}$ satisfy

$$\|a_1^{(t)}\| = \|a_2^{(t)}\| = \ldots = \|a_n^{(t)}\|. \tag{172}$$

Then, based on the definition of the client weights $w_i$ in 69, we obtain

$$w_i = \frac{\|a_i^{(t)}\|}{\rho^{(t)}} p_i = \frac{\|a_i^{(t)}\|}{\sum_{j=1}^n p_j \|a_j^{(t)}\|} p_i = p_i, \quad \forall\, i \in \{1, 2, \cdots, n\}, \tag{173}$$

and the global upper-level optimization problem (1a) reduces to the refactored form defined in (76a)

$$\widetilde{\Phi}(x^{(t)}) = \widetilde{F}(x^{(t)}, \widetilde{y}^*(x^{(t)})) = \sum_{i=1}^n w_i f_i(x^{(t)}, \widetilde{y}^*(x^{(t)})). \tag{174}$$

However, if system heterogeneity is present, meaning that for some global iteration round $t \in \{0, 1, \cdots, T-1\}$ and clients $i \in \{1, 2, \cdots, n\}$, the local gradient aggregation coefficient vectors $a_i^{(t)}$ may differ, we may have

$$\|a_i^{(t)}\| \neq \|a_j^{(t)}\|, \quad \exists\, i \neq j \in \{1, 2, \cdots, n\}, \tag{175}$$

and the definition of $w_i$ in 69 leads to

$$w_i = \frac{\|a_i^{(t)}\|}{\rho^{(t)}} p_i = \frac{\|a_i^{(t)}\|}{\sum_{j=1}^n p_j \|a_j^{(t)}\|} p_i \neq p_i, \quad \exists\, i \in \{1, 2, \cdots, n\}, \tag{176}$$

which implies that the global upper-level optimization problem (1a) must be expressed in its original form

$$\Phi(x^{(t)}) = F(x^{(t)}, y^*(x^{(t)})) = \sum_{i=1}^n p_i f_i(x^{(t)}, y^*(x^{(t)})). \tag{177}$$

**Theorem 2** (**Convergence of Algorithm 1**). *Under Assumptions 1–9, the algorithms ASFBO and LA-ASFBO satisfy the following convergence guarantees.*

*In the absence of system heterogeneity, the algorithms satisfy*

$$\min_t \left\| \nabla \widetilde{\Phi}(x^{(t)}) \right\|^2 \le \frac{1}{T} \sum_{t=0}^{T-1} \mathbb{E}\left[\left\| \nabla \widetilde{\Phi}(x^{(t)}) \right\|^2\right]$$

$$= \mathcal{O}\left( M_{sync} \cdot \frac{\beta_{\max}^2 \alpha_{\max}^2}{(c_a')^2 \alpha_{\min}^2} \cdot \sqrt{\frac{1}{P\bar{\tau}T}} \right)$$

$$+ \mathcal{O}\left( M_{part} \cdot \frac{n-P}{n-1} \cdot \sqrt{\frac{\bar{\tau}}{PT}} \right)$$

$$+ \mathcal{O}\left( M_{drift} \cdot (\sigma_{M1}^2 + \sigma_{M2}^2) \cdot \frac{1}{\bar{\tau}T} \right), \tag{178}$$

*and in the presence of system heterogeneity, the algorithms satisfy*

$$\min_t \left\| \nabla \Phi(x^{(t)}) \right\|^2 \le \frac{1}{T} \sum_{t=0}^{T-1} \mathbb{E}\left[\left\| \nabla \Phi(x^{(t)}) \right\|^2\right]$$

$$= \mathcal{O}\left( M \cdot M_{sync} \cdot \frac{\beta_{\max}^2 \alpha_{\max}^2}{(c_a')^2 \alpha_{\min}^2} \cdot \sqrt{\frac{1}{P\bar{\tau}T}} \right)$$

$$+ \mathcal{O}\left( M \cdot M_{part} \cdot \frac{n-P}{n-1} \cdot \sqrt{\frac{\bar{\tau}}{PT}} \right)$$

$$+ \mathcal{O}\left( M \cdot M_{drift} \cdot (\sigma_{M1}^2 + \sigma_{M2}^2) \cdot \frac{1}{\bar{\tau}T} \right), \tag{179}$$

*where the constants $M, M_{part}, M_{sync}, M_{drift}$ are defined as follows:*

$$M := 1 + 2\left( \frac{\beta_{\max}' - \beta_{\min}}{\beta_{\min}} \right)^2, \tag{180a}$$

$$M_{sync} := \left[ L_\Phi^2 + K_y\left(2L_y^2 + L_{yx}\right) + K_z\left(2L_z^2 + L_{zx}\right) + 4K_z c_{\gamma_z}^2 \right]\left(\sigma_f^2 + r_{\max}^2 \sigma_{gg}^2\right)$$
$$+ K_y c_{\gamma_y}^2 \sigma_g^2, \tag{180b}$$

$$M_{part} := 2\left[ L_\Phi^2 + K_y\left(2L_y^2 + L_{yx}\right) + K_z\left(2L_z^2 + L_{zx}\right) + 4K_z c_{\gamma_z}^2 \right]\beta_{\max}(L_f^2 + r_{\max}^2 L_1^2)$$
$$+ 2K_y c_{\gamma_y}^2 \beta_{\max} \sigma_{gh}^2, \tag{180c}$$

$$M_{drift} := 3\left[ \frac{3}{2} + 24K_z c_{\gamma_z} + \frac{\beta_{\max}}{P}\left( \frac{17}{4} + 16K_z c_{\gamma_z} \right) \right](L_1^2 + r^2 L_2^2)$$
$$+ 4K_y\left[ c_{\gamma_y} \frac{L_1^2}{\mu_g} + 2c_{\gamma_z}\left( \frac{2\beta_{\max}}{P} + 3 \right) L_1^2 \right] + 4K_z c_{\gamma_z} \frac{L_R^2}{\mu_g}. \tag{180d}$$

*Proof.* The proof begins with the case without system heterogeneity and is later extended to the case with system heterogeneity.

**(1)** In the absence of system heterogeneity, conditions (172) and (173) are satisfied. To establish the convergence of the proposed method, we aim to show that the expected squared norm of the gradient of the objective function $\widetilde{\Phi}(x)$ after $T$ iterations (see Definition 2), i.e.,

$$\min_t \mathbb{E}\left[\left\| \nabla \widetilde{\Phi}(x^{(t)}) \right\|^2\right] \le \frac{1}{T} \sum_{t=0}^{T-1} \mathbb{E}\left[\left\| \nabla \widetilde{\Phi}(x^{(t)}) \right\|^2\right]$$

$$\overset{(a)}{\le} \frac{4}{T\bar{\rho}\gamma_{x,\min}}\left[ \Psi(x^{(0)}) - \Psi(x^{(T)}) \right] + \frac{4}{T} \sum_{t=0}^{T-1} \frac{1}{\bar{\rho}\gamma_{x,\min}}\left( \epsilon_{\text{sync}}^{(t)} + \epsilon_{port}^{(t)} + \epsilon_{cd}^{(t)} \right)$$

$$\stackrel{(b)}{=} \mathcal{O}\left(M_{\text{sync}} \cdot \frac{\beta_{\max}^2 \alpha_{\max}^2}{(c_a')^2 \alpha_{\min}^2} \cdot \sqrt{\frac{1}{P\bar{\tau}T}}\right) + \mathcal{O}\left(M_{\text{part}} \cdot \frac{n-P}{n-1} \cdot \sqrt{\frac{\bar{\tau}}{PT}}\right)$$
$$+ \mathcal{O}\left(M_{\text{drift}} \cdot (\sigma_{M1}^2 + \sigma_{M2}^2) \cdot \frac{1}{\bar{\tau}T}\right)$$

We first analyze step $(a)$. By summing inequality (168) from $t = 0$ to $T-1$, we obtain:

$$\Psi(x^{(T)}) - \Psi(x^{(0)}) \leq -\sum_{t=0}^{T-1} \frac{\rho^{(t)} \gamma_x^{(t)}}{2} \mathbb{E}\left[\left\|\nabla\widetilde{\Phi}(x^{(t)})\right\|^2\right] + \sum_{t=0}^{T-1} \left(\epsilon_{\text{sync}}^{(t)} + \epsilon_{\text{part}}^{(t)} + \epsilon_{\text{drift}}^{(t)}\right) \tag{181}$$

Combining with Assumption 9 and the fact that $\rho^{(t)} \in [\frac{1}{2}\bar{\rho}, \frac{3}{2}\bar{\rho}]$, we have:

$$\frac{\bar{\rho}\gamma_{x,\min}}{4} \sum_{t=0}^{T-1} \mathbb{E}\left[\left\|\nabla\widetilde{\Phi}(x^{(t)})\right\|^2\right] \leq \sum_{t=0}^{T-1} \frac{\rho^{(t)}\gamma_x^{(t)}}{2} \mathbb{E}\left[\left\|\nabla\widetilde{\Phi}(x^{(t)})\right\|^2\right]$$
$$\leq \Psi(x^{(0)}) - \Psi(x^{(T)}) + \sum_{t=0}^{T-1} \left(\epsilon_{\text{part}}^{(t)} + \epsilon_{\text{sync}}^{(t)} + \epsilon_{cd}^{(t)}\right), \tag{182}$$

which leads to:

$$\sum_{t=0}^{T-1} \mathbb{E}\left[\left\|\nabla\widetilde{\Phi}(x^{(t)})\right\|^2\right] \leq \frac{4}{\bar{\rho}\gamma_{x,\min}}\left[\Psi(x^{(0)}) - \Psi(x^{(T)})\right] + 4\sum_{t=0}^{T-1} \frac{1}{\bar{\rho}\gamma_{x,\min}}\left(\epsilon_{\text{sync}}^{(t)} + \epsilon_{\text{part}}^{(t)} + \epsilon_{\text{drift}}^{(t)}\right). \tag{183}$$

We next analyze the term $\frac{4}{T\bar{\rho}\gamma_{x,\min}}\left[\Psi(x^{(0)}) - \Psi(x^{(T)})\right]$ in step $(b)$. Since the Lyapunov function is generally non-negative, i.e., $\Psi(x^{(T)}) \geq 0$, we have:

$$\Psi(x^{(0)}) - \Psi(x^{(T)}) \leq \Psi(x^{(0)}), \tag{184}$$

where $\Psi(x^{(0)})$ is a constant. From Assumption 9 and (160a), we know that $\gamma_{x,\min} \leq \gamma_x^{(t)} = \mathcal{O}\left(\sqrt{\frac{P}{\bar{\tau}T}}\right)$, and from the definitions of $\rho^{(t)}$ in (69) and $\bar{\rho}$ in (167b), it follows that $\bar{\rho} = \mathcal{O}(\bar{\tau})$. Therefore,

$$\frac{4}{T\bar{\rho}\gamma_{x,\min}}\left[\Psi(x^{(0)}) - \Psi(x^{(T)})\right] = \mathcal{O}\left(\sqrt{\frac{1}{P\bar{\tau}T}}\right). \tag{185}$$

We then analyze the term $\frac{4}{T}\sum_{t=0}^{T-1}\frac{1}{\bar{\rho}\gamma_{x,\min}}\epsilon_{\text{sync}}^{(t)}$ in step $(b)$. From Assumptions 6, 7, and 8, it follows that the term $w_i^2\|a_i^{(t)}\|_2^2/\|a_i^{(t)}\|_1^2$ in the definition of $\epsilon_{\text{sync}}^{(t)}$ (see (169a)) satisfies:

$$\frac{w_i^2\|a_i^{(t)}\|_2^2}{\|a_i^{(t)}\|_1^2} \leq \frac{\beta_{\max}^2\alpha_{\max}^2}{\bar{\tau}n^2(c_a')^2\alpha_{\min}^2}, \tag{186}$$

and by the definition of $M_{\text{sync}}$ in (180b), we obtain:

$$\epsilon_{\text{sync}}^{(t)} \leq (\rho^{(t)}\gamma_x^{(t)})^2 \frac{\beta_{\max}^2\alpha_{\max}^2}{P\bar{\tau}(c_a')^2\alpha_{\min}^2} M_{\text{sync}} \leq \left(\frac{3}{2}\bar{\rho}\gamma_{x,\max}\right)^2 \frac{\beta_{\max}^2\alpha_{\max}^2}{P\bar{\tau}(c_a')^2\alpha_{\min}^2} M_{\text{sync}}. \tag{187}$$

Therefore,

$$\frac{4}{T}\sum_{t=0}^{T-1}\frac{1}{\bar{\rho}\gamma_{x,\min}}\epsilon_{\text{sync}}^{(t)} \leq \frac{4}{T}\sum_{t=0}^{T-1}\frac{1}{\bar{\rho}\gamma_{x,\min}} \cdot \left(\frac{3}{2}\bar{\rho}\gamma_{x,\max}\right)^2 \frac{\beta_{\max}^2\alpha_{\max}^2}{P\bar{\tau}(c_a')^2\alpha_{\min}^2} M_{\text{sync}}$$
$$= M_{\text{sync}} \cdot \frac{9\beta_{\max}^2\alpha_{\max}^2}{(c_a')^2\alpha_{\min}^2} \cdot \frac{\bar{\rho}\gamma_{x,\max}^2}{P\bar{\tau}\gamma_{x,\min}}$$

$$\stackrel{(c)}{=} \mathcal{O}\left(M_{\text{sync}} \cdot \frac{\beta_{\max}^2 \alpha_{\max}^2}{(c_a')^2 \alpha_{\min}^2} \cdot \sqrt{\frac{1}{P\bar{\tau}T}}\right), \tag{188}$$

where in step $(c)$ we used $\bar{\rho} = \mathcal{O}(\bar{\tau})$ and $\frac{\gamma_{x,\max}^2}{\gamma_{x,\min}} = \mathcal{O}\left(\sqrt{\frac{P}{\bar{\tau}T}}\right)$.

We now analyze the term $\frac{4}{T}\sum_{t=0}^{T-1}\frac{1}{\bar{\rho}\gamma_{x,\min}}\epsilon_{\text{part}}^{(t)}$ in step $(b)$. From the definition of $\epsilon_{\text{part}}^{(t)}$ in (169b) and the definition of $M_{\text{part}}$ in (180c), we have:

$$\epsilon_{\text{part}}^{(t)} \le (\rho^{(t)}\gamma_x^{(t)})^2 \cdot \frac{n-P}{P(n-1)}M_{\text{part}} \le \left(\frac{3}{2}\bar{\rho}\gamma_{x,\max}\right)^2 \cdot \frac{n-P}{P(n-1)}M_{\text{part}}. \tag{189}$$

Then,

$$\begin{aligned}
\frac{4}{T}\sum_{t=0}^{T-1}\frac{1}{\bar{\rho}\gamma_{x,\min}}\epsilon_{\text{part}}^{(t)} &\le \frac{4}{T}\sum_{t=0}^{T-1}\frac{1}{\bar{\rho}\gamma_{x,\min}}\cdot\left(\frac{3}{2}\bar{\rho}\gamma_{x,\max}\right)^2 \cdot \frac{n-P}{P(n-1)}M_{\text{part}} \\
&= M_{\text{part}}\cdot\frac{9(n-P)}{n-1}\cdot\frac{\bar{\rho}\gamma_{x,\max}^2}{P\gamma_{x,\min}} \\
&\stackrel{(d)}{=} \mathcal{O}\left(M_{\text{part}}\cdot\frac{n-P}{n-1}\cdot\sqrt{\frac{\bar{\tau}}{PT}}\right),
\end{aligned} \tag{190}$$

where in step $(d)$ we used $\bar{\rho} = \mathcal{O}(\bar{\tau})$ and $\frac{\gamma_{x,\max}^2}{\gamma_{x,\min}} = \mathcal{O}\left(\sqrt{\frac{P}{\bar{\tau}T}}\right)$.

We next analyze the term $\frac{4}{T}\sum_{t=0}^{T-1}\frac{1}{\bar{\rho}\gamma_{x,\min}}\epsilon_{\text{drift}}^{(t)}$ in step $(b)$. From the definition of $\epsilon_{\text{drift}}^{(t)}$ in (169c), we consider the quantity $\eta_x^2\bar{\tau}\sigma_{M1}^2 + 2\eta_y^2\bar{\tau}\sigma_{M2}^2$, which satisfies

$$\eta_x^2\bar{\tau}\sigma_{M1}^2 + 2\eta_y^2\bar{\tau}\sigma_{M2}^2 \le (\eta_x^2 + \eta_z^2)\bar{\tau}\sigma_{M1}^2 + 2\eta_y^2\bar{\tau}\sigma_{M2}^2. \tag{191}$$

Then from the definition of $M_{\text{drift}}$ in (180d), we have

$$\begin{aligned}
\epsilon_{\text{drift}}^{(t)} &\le \rho^{(t)}\gamma_x^{(t)}\cdot\left[(\eta_x^2 + \eta_z^2)\bar{\tau}\sigma_{M1}^2 + 2\eta_y^2\bar{\tau}\sigma_{M2}^2\right]M_{\text{drift}} \\
&\le \left(\frac{3}{2}\bar{\rho}\gamma_{x,\max}\right)\cdot\left[(\eta_x^2 + \eta_z^2)\bar{\tau}\sigma_{M1}^2 + 2\eta_y^2\bar{\tau}\sigma_{M2}^2\right]M_{\text{drift}}.
\end{aligned} \tag{192}$$

Therefore,

$$\begin{aligned}
\frac{4}{T}\sum_{t=0}^{T-1}\frac{1}{\bar{\rho}\gamma_{x,\min}}\epsilon_{\text{drift}}^{(t)} &\le \frac{4}{T}\sum_{t=0}^{T-1}\frac{1}{\bar{\rho}\gamma_{x,\min}}\cdot\left(\frac{3}{2}\bar{\rho}\gamma_{x,\max}\right)\cdot\left[(\eta_x^2 + \eta_z^2)\bar{\tau}\sigma_{M1}^2 + 2\eta_y^2\bar{\tau}\sigma_{M2}^2\right]M_{\text{drift}} \\
&= M_{\text{drift}}\cdot\frac{3}{2}\cdot\frac{\bar{\rho}\gamma_{x,\max}}{\gamma_{x,\min}}\cdot\left[(\eta_x^2 + \eta_z^2)\bar{\tau}\sigma_{M1}^2 + 2\eta_y^2\bar{\tau}\sigma_{M2}^2\right] \\
&\stackrel{(e)}{=} \mathcal{O}\left(M_{\text{drift}}\cdot(\sigma_{M1}^2 + \sigma_{M2}^2)\cdot\frac{1}{\bar{\tau}T}\right),
\end{aligned} \tag{193}$$

where in step $(e)$ we use $\bar{\rho} = \mathcal{O}(\bar{\tau})$, $\frac{\gamma_{x,\max}}{\gamma_{x,\min}} = \mathcal{O}(1)$, and the local step sizes $\eta_x = \mathcal{O}\left(\frac{1}{\bar{\tau}\sqrt{T}}\right)$, $\eta_y = \mathcal{O}\left(\frac{1}{\bar{\tau}\sqrt{T}}\right)$, $\eta_z = \mathcal{O}\left(\frac{1}{\bar{\tau}\sqrt{T}}\right)$.

**(2)** We now consider the case with system heterogeneity, i.e., conditions (175) and (176) hold. Based on the objective functions without and with heterogeneity given in (174) and (177), the gradient difference satisfies

$$\begin{aligned}
\nabla\Phi(x^{(t)}) &- \nabla\widetilde{\Phi}(x^{(t)}) \\
&= \nabla\bar{F}(x^{(t)}, y^*(x^{(t)})) - \nabla\widetilde{\bar{F}}(x^{(t)}, \widetilde{y}^*(x^{(t)}))
\end{aligned}$$

$$= \sum_{i=1}^{n} p_i \nabla \bar{f}_i(x^{(t)}, y^*(x^{(t)})) - \sum_{i=1}^{n} w_i \nabla \bar{f}_i(x^{(t)}, \widetilde{y}^*(x^{(t)}))$$

$$= \sum_{i=1}^{n} p_i \left[ \nabla \bar{f}_i(x^{(t)}, y^*(x^{(t)}), z^*(x^{(t)})) - \nabla \bar{f}_i(x^{(t)}, \widetilde{y}^*(x^{(t)}), \widetilde{z}^*(x^{(t)})) \right]$$

$$+ \sum_{i=1}^{n} \frac{p_i - w_i}{w_i} w_i \nabla \bar{f}_i(x^{(t)}, \widetilde{y}^*(x^{(t)}), \widetilde{z}^*(x^{(t)})). \tag{194}$$

Taking squared norms on both sides, we get

$$\left\| \nabla \Phi(x^{(t)}) - \nabla \widetilde{\Phi}(x^{(t)}) \right\|^2 \le 2 \left\| \nabla \bar{F}(x^{(t)}, y^*(x^{(t)}), z^*(x^{(t)})) - \nabla \bar{F}(x^{(t)}, \widetilde{y}^*(x^{(t)}), \widetilde{z}^*(x^{(t)})) \right\|^2$$

$$+ 2 \left\| \sum_{i=1}^{n} \frac{p_i - w_i}{w_i} w_i \nabla \bar{f}_i(x^{(t)}, \widetilde{y}^*(x^{(t)}), \widetilde{z}^*(x^{(t)})) \right\|^2$$

$$\overset{(a)}{\le} 6(L_1^2 + r^2 L_2^2) \|y^*(x^{(t)}) - \widetilde{y}^*(x^{(t)})\|^2$$

$$+ 6L_1^2 \|z^*(x^{(t)}) - \widetilde{z}^*(x^{(t)})\|^2$$

$$+ 2 \left( \frac{\beta'_{\max} - \beta_{\min}}{\beta_{\min}} \right)^2 \left\| \nabla \widetilde{\Phi}(x^{(t)}) \right\|^2, \tag{195}$$

where step $(a)$ follows from the surrogate gradient expansion (48), norm inequalities, Lipschitz continuity in Assumption 3, and boundedness results in Lemma 4. The weight coefficient bound in Assumption 8 gives

$$\frac{p_i - w_i}{w_i} \le \frac{\beta'_{\max} - \beta_{\min}}{\beta_{\min}}. \tag{196}$$

**(3)** Substituting inequality (195) into (179), we obtain

$$\min_t \left\| \nabla \Phi(x^{(t)}) \right\|^2 \le \frac{2}{T} \sum_{t=0}^{T-1} \left\| \nabla \widetilde{\Phi}(x^{(t)}) \right\|^2 \cdot \left[ 1 + \left( \frac{\beta'_{\max} - \beta_{\min}}{\beta_{\min}} \right)^2 \right]$$

$$+ 12(L_1^2 + r^2 L_2^2) \cdot \frac{1}{T} \sum_{t=0}^{T-1} \|y^*(x^{(t)}) - \widetilde{y}^*(x^{(t)})\|^2$$

$$+ 12L_1^2 \cdot \frac{1}{T} \sum_{t=0}^{T-1} \|z^*(x^{(t)}) - \widetilde{z}^*(x^{(t)})\|^2$$

$$\overset{(f)}{\approx} M \cdot \frac{1}{T} \sum_{t=0}^{T-1} \left\| \nabla \widetilde{\Phi}(x^{(t)}) \right\|^2$$

$$\overset{(g)}{=} \mathcal{O} \left( M \cdot M_{\text{sync}} \cdot \frac{\beta_{\max}^2 \alpha_{\max}^2}{(c_a')^2 \alpha_{\min}^2} \cdot \sqrt{\frac{1}{P\bar{\tau}T}} \right)$$

$$+ \mathcal{O} \left( M \cdot M_{\text{part}} \cdot \frac{n - P}{n - 1} \cdot \sqrt{\frac{\bar{\tau}}{PT}} \right)$$

$$+ \mathcal{O} \left( M \cdot M_{\text{drift}} \cdot (\sigma_{M1}^2 + \sigma_{M2}^2) \cdot \frac{1}{\bar{\tau}T} \right). \tag{197}$$

In step $(f)$, since the drift error $\epsilon_{\text{drift}}^{(t)}$ is independent of system heterogeneity as long as local step sizes $\{\eta_x, \eta_y, \eta_z\}$ and average local iteration number $\bar{\tau}$ are unaffected, we can approximate

$$y^*(x^{(t)}) \approx \widetilde{y}^*(x^{(t)}), \tag{198a}$$

$$z^*(x^{(t)}) \approx \widetilde{z}^*(x^{(t)}). \tag{198b}$$

In step $(g)$, we apply inequality (181), and obtain

$$M \cdot \frac{1}{T} \sum_{t=0}^{T-1} \left\| \nabla \widetilde{\Phi}(x^{(t)}) \right\|^2 = M \cdot \mathcal{O}\left( M_{\text{sync}} \cdot \frac{\beta_{\max}^2 \alpha_{\max}^2}{(c_a')^2 \alpha_{\min}^2} \cdot \sqrt{\frac{1}{P\bar{\tau}T}} \right)$$

$$+ M \cdot \mathcal{O}\left( M_{\text{part}} \cdot \frac{n-P}{n-1} \cdot \sqrt{\frac{\bar{\tau}}{PT}} \right)$$

$$+ M \cdot \mathcal{O}\left( M_{\text{drift}} \cdot (\sigma_{M1}^2 + \sigma_{M2}^2) \cdot \frac{1}{\bar{\tau}T} \right) \tag{199}$$

This concludes the proof of Theorem 2. $\qquad \square$

According to the client participation rate, we distinguish between two cases: $P \approx n$ and $P < n$, and analyze the communication and computational complexity of the ASFBO and LA-ASFBO algorithms respectively.

**Corollary 2** (**Complexity of Algorithm 1**). *When $P \approx n$, to achieve $\epsilon$-accuracy, the communication complexity is $T = \mathcal{O}(\epsilon^{-2})$, and the per-client computation complexity is $\bar{\tau}T = \mathcal{O}(n^{-1}\epsilon^{-2})$. By setting the local iteration count to $\bar{\tau} = \mathcal{O}(\frac{T}{n})$, the communication complexity can be improved to $T = \mathcal{O}(\epsilon^{-1})$. When $P < n$, to achieve $\epsilon$-accuracy, the communication complexity becomes $T = \mathcal{O}(P^{-1}\epsilon^{-2})$, and the per-client computation complexity is $\bar{\tau}T = \mathcal{O}(P^{-1}\epsilon^{-2})$.*

*Proof.* **(1)** When nearly all clients participate in the optimization (i.e., $P \approx n$), we have $\frac{n-P}{n-1} \approx 0$, in which case the partial participation error satisfies

$$\mathcal{O}\left( M \cdot M_{\text{part}} \cdot \frac{n-P}{n-1} \cdot \sqrt{\frac{\bar{\tau}}{PT}} \right) \approx 0 \tag{200}$$

Hence, inequality (179) simplifies to

$$\min_t \left\| \nabla \Phi(x^{(t)}) \right\|^2 = \mathcal{O}\left( M \cdot M_{\text{sync}} \cdot \frac{\beta_{\max}^2 \alpha_{\max}^2}{(c_a')^2 \alpha_{\min}^2} \cdot \sqrt{\frac{1}{P\bar{\tau}T}} \right)$$

$$+ \mathcal{O}\left( M \cdot M_{\text{drift}} \cdot (\sigma_{M1}^2 + \sigma_{M2}^2) \cdot \frac{1}{\bar{\tau}T} \right)$$

$$\leq \epsilon. \tag{201}$$

In this case, to achieve $\epsilon$-accuracy, the total number of global iterations (i.e., communication rounds) must satisfy $T = \mathcal{O}(\epsilon^{-2})$. The per-client computational complexity is then $\bar{\tau}T = \mathcal{O}(n^{-1}\epsilon^{-2})$. Furthermore, by setting $\bar{\tau} = \mathcal{O}(\frac{T}{n})$, the communication complexity can be improved to $T = \mathcal{O}(\epsilon^{-1})$.

**(2)** When only a subset of clients participate (i.e., $P < n$), the partial participation error is non-negligible. However, if we set $\bar{\tau} = \mathcal{O}\left( \frac{n-1}{n-P} \right)$, inequality (179) can be simplified to

$$\min_t \left\| \nabla \Phi(x^{(t)}) \right\|^2 = \mathcal{O}\left( M \cdot \left( M_{\text{sync}} \cdot \frac{\beta_{\max}^2 \alpha_{\max}^2}{(c_a')^2 \alpha_{\min}^2} + M_{\text{part}} \right) \cdot \sqrt{\frac{n-P}{(n-1)PT}} \right)$$

$$+ \mathcal{O}\left( M \cdot M_{\text{drift}} \cdot (\sigma_{M1}^2 + \sigma_{M2}^2) \cdot \frac{1}{\bar{\tau}T} \right)$$

$$\leq \epsilon. \tag{202}$$

Under this setting, to achieve $\epsilon$-accuracy, the number of global iterations (communication rounds) must satisfy $T = \mathcal{O}(P^{-1}\epsilon^{-2})$, and the per-client computational complexity is $\bar{\tau}T = \mathcal{O}(P^{-1}\epsilon^{-2})$.

This completes the proof of Corollary 2. $\qquad \square$

# D  Experiment Details

This section conducts numerical experiments to evaluate the performance of the proposed ASFBO and LA-ASFBO algorithms. We begin by introducing the federated hyper-representation learning (FHRL) task used in the experiments. Then, we describe the experimental setup, including datasets, models, and algorithmic hyperparameters. Finally, we evaluate the performance of the proposed algorithms under both IID and Non-IID settings, and compare them with the baseline methods SimFBO Yang et al. (2023), ShroFBO Yang et al. (2023), FedNest Tarzanagh et al. (2022), and LFedNest Tarzanagh et al. (2022).

All experiments are conducted on a Linux system equipped with an Intel i5-12500H CPU and an Nvidia GeForce RTX 3050 GPU using Python 3.12.

## D.1  Federated Hyper-Representation Learning

This subsection introduces the federated hyper-representation learning (FHRL) task and formulates it as a federated bilevel optimization problem.

FHRL is derived from the concept of hyper-representation learning Schürholt et al. (2022), which aims to improve generalization across downstream tasks by learning a shared representation. It is widely used in meta-learning and multi-task learning scenarios. Hyper-representation learning is essentially a class of bilevel optimization problems Franceschi et al. (2018), whose objective is to learn a general-purpose feature extractor (referred to as the "representation" or backbone) such that task-specific classifiers or lightweight modules can be efficiently and effectively trained atop this shared representation.

In the federated setting, the FHRL task can be modeled as the following federated bilevel optimization problem:

$$\min_{x \in \mathbb{R}^p} F(x) = \frac{1}{n} \sum_{i=1}^{n} \frac{1}{|\mathcal{D}_{i,\text{val}}|} \sum_{\xi \in \mathcal{D}_{i,\text{val}}} \mathcal{L}\left(x, y^*(x); \xi\right), \tag{203a}$$

$$\text{s.t.} \quad y^*(x) = \arg\min_{y \in \mathbb{R}^d} \frac{1}{n} \sum_{i=1}^{n} \frac{1}{|\mathcal{D}_{i,\text{train}}|} \sum_{\zeta \in \mathcal{D}_{i,\text{train}}} \mathcal{L}\left(x, y; \zeta\right), \tag{203b}$$

where $\mathcal{D}_{\text{train}}$ and $\mathcal{D}_{\text{val}}$ denote the complete training and validation sets, and $\zeta$ and $\xi$ represent individual data samples. $\mathcal{D}_{i,\text{train}}$ and $\mathcal{D}_{i,\text{val}}$ denote the local training and validation subsets on client $i$, respectively. The variable $x$ corresponds to the shared representation parameters, such as the backbone of a multilayer perceptron (MLP) Rumelhart et al. (1986) or a convolutional neural network (CNN) LeCun et al. (1989). The variable $y^*(x)$ represents the task-specific classifier (head) trained optimally on $\mathcal{D}_{\text{train}}$ given a fixed $x$. $\mathcal{L}(\cdot)$ denotes the loss function, typically the cross-entropy loss.

The upper-level objective focuses on generalization to the validation set, while the lower-level problem seeks optimal task-specific adaptation on the local training data. Since the output layer in neural networks is typically a linear transformation without nonlinear activation and the loss function is usually convex (e.g., cross-entropy or mean squared error), the optimization over the output layer satisfies strong convexity as long as the output of the preceding layer is full-rank.

## D.2  Experimental Setup

To ensure fair comparison with existing methods, the dataset, model, and part of the framework settings used in our experiments follow those in (Tarzanagh et al., 2022; Yang et al., 2023).

### D.2.1  Dataset

We use the MNIST dataset LeCun et al. (1998), consisting of grayscale images of handwritten digits. Each image is of size $28 \times 28$ pixels. The training set contains 60,000 labeled samples, and the test set contains another 10,000 labeled samples.

In the IID setting, each client randomly selects 600 non-overlapping samples from the full training set without replacement, simulating data homogeneity. Each client then splits these 600 samples into 450 for training and 150 for validation.

In the Non-IID setting, the samples are first sorted by label. The sorted dataset is partitioned into multiple subsets of 300 samples each, from which each client randomly selects two subsets (i.e., 600 samples total) without replacement, simulating high data heterogeneity. Each client again splits the 600 samples into 450 for training and 150 for validation. For evaluation, all 10,000 test samples are used.

### D.2.2   Model

We adopt a multilayer perceptron (MLP) Rumelhart et al. (1986) as the model for the FHRL task. The MLP contains one input layer, one hidden layer, and one output layer. The input layer maps the 784-dimensional input to 200 dimensions. The hidden layer contains 200 neurons and uses the ReLU (Rectified Linear Unit) activation Nair & Hinton (2010). The output layer maps from 200 to 10 dimensions, and the loss function is L1-regularized cross-entropy loss.

We designate the 157,000 parameters between the input and hidden layer as upper-level variables, and the 2,010 parameters between the hidden and output layer as lower-level variables.

### D.2.3   Algorithm Hyperparameters

All algorithms considered support partial client participation in each round. We set the total number of clients to $n = 100$, and randomly sample $P = 10$ clients in each round.

For the server-side learning rate scaling factor update (71), we set the decay factor $\varsigma^{(t)} = 0.75$. The base global learning rates in (72) are set to $\{\gamma_x = 0.03, \gamma_y = 0.03, \gamma_z = 0.05\}$, with the small constant $\rho = 0.001$.

In the learning rate bound conditions (88), we set the lower bounds as $\{\gamma_{x,\min} = 0.01, \gamma_{y,\min} = 0.03, \gamma_{z,\min} = 0.02\}$, and the upper bounds as $\{\gamma_{x,\max} = 0.1, \gamma_{y,\max} = 0.3, \gamma_{z,\max} = 0.2\}$.

The average number of local iterations is $\bar{\tau} = 10$, so each client's local iteration count $\tau_i^{(t)}$ is uniformly drawn from $[5, 15]$.

In the client-side variable update step (64), the local learning rates are set to $\{\eta_x = 0.01, \eta_y = 0.03, \eta_z = 0.02\}$.

In the momentum updates (65) and (66), the momentum coefficient is set to $\beta = 0.25$.

