# OpenReview forum: "Communication-Efficient Adaptive Federated Bi-level Optimization with Data and System Heterogeneity"
_TMLR — Rejected by TMLR_

### Review · Reviewer_VJz2 · 2025-12-18

**Summary Of Contributions:**

This submission studies federated stochastic bilevel optimization under non-IID data, partial participation, and system heterogeneity (clients performing different numbers of local steps). It proposes two algorithms (ASFBO and LA-ASFBO) that combine (i) a single-loop surrogate hypergradient estimator, (ii) STORM-style momentum/variance reduction, (iii) a reweighted aggregation mechanism intended to correct bias from heterogeneous local computation, and (iv) a server-side adaptive scaling reminiscent of AdaGrad-Norm. The paper provides convergence/complexity statements and experiments primarily on MNIST/MLP, comparing to several baselines.

*Strengths*

- Targets an important and nontrivial problem (bilevel + FL + system heterogeneity).

- Algorithmic design is internally coherent (variance reduction + reweighting + server adaptivity).

- Includes both theory and empirical comparisons.

*Weaknesses*

- A central theoretical argument appears to rely on an unjustified approixmation, undermining the rigor of the guarantees.

- Communication-complexity claims appear overstated without a careful accounting of the communication–computation trade-off.

- Empirical evlauation and ablations are too narrow to support the breadth of the claims.

- Notation/terminology inconsistencies (including a confusing “PP” usage) reduce clarity and confidence.

**Audience:**

Yes

**Audience Explanation:**

Yes, the topic is relevant to TMLR readers interested in federated optimization, bilevel methods (hyperparameter tuning / meta-learning in distributed settings), and robustness to system heterogeneity. However, interest does not translate to publishability here, because the current theoretical and empirical support is not yet at the level expected for TMLR. With substantial revisions, it could become a useful reference point.

**Broader Impact Concerns:**

I do not see severe ethical concerns intrinsic to the algorithms, but a few points should be addressed more clearly:

- Privacy implications should not be overstated. If the paper uses language that could be interpreted as “privacy protection,” it should explicitly clarify that federated learning is not inherently private, and that no formal privacy guarantee (e.g., differential privacy) is provided unless it truly is.

- Downstream use risks: Bilevel optimization can tune models in ways that may amplify biases or degrade performance for minority users if objectives are misspecified. A brief statement acknowledging this risk and recommending evaluation across sub-populations would strengthen the broader impact discussion.

**Claims And Evidence:**

No

**Claims Explanation:**

Not sufficiently, for the following reasons:

- Theoretical guarantees are not fully convincing as written.
A key part of the analysis seems to invoke an approximation (e.g., treating certain optimal quantities like
$y^∗(x)$ and $z^∗(x)$ as interchangeable with approximations) to obtain drift/heterogeneity-independent bounds. Unless this step is made rigorous (with explcit error control and assumptions) or the claims are weakened accordingly, the presented convergence/complexity story is not adequately supported.

- The “$\tilde{O}(ε^{−1})$ communication rounds” narrative is not clearly justified.
The improvement appears to rely on parameter choices that incraese local computation (e.g., scaling the average number of local steps with the horizon). Without a transparent statement of total cost (rounds, transmitted bytes, local gradient computations, and wall-clock), the evidence does not convincingly support the headline claim that communication is “significantly reduced.”

- Empirical evidence is too limited for the scope of the claims.
Experiments are largely confined to MNIST/MLP and relatively constrained settings. This is not enough to substantiate broad statements about robustness to heterogeneity and generalization, especialy given the complexity of bilevel FL dynamics.

- Attribution to components (STORM vs reweighting vs adaptivity) is not established.
The paper often implies specific components are responsible for gains, but without clean, component-wise ablations, this remains suggestive rather than convincing.

Overall: while the direction is plausible, the evidence (as currently presented) is not yet accurate/convincing/clear enough to support the strongest claims.

**Requested Changes:**

Below are proposed adjustments, each marked as Critical (needed to change my recommendation) or Strengthening (would improve the work but not strictly required on its own).

*Critical*

- Fix the theoretical gap / remove unproven approximations.
Make all key steps in the proofs fully rigorous, with explicit bounds on approximation error and clearly stated assumptions; or weaken claims to match what is actually proven.

- Recalibrate and clarify communication claims. State explicitly what is being counted (rounds, bytes, gradients communicated); Provide a clear communication–computation trade-offf analysis (including regimes where gains hold); Align the abstract/introduction claims with the formal complexity results.

- Substantially broaden empirical validation. Add at least 1–2 more realistic benchmarks (larger datasets/models) relevant to federated learning; Report additional metrics: communication volume, wall-clock time (if feasible), sensitivity to local steps distribution, and stability across random seeds.

- Add clean ablations isolating each component.
Evaluate (i) no reweighting, (ii) no STORM/variance reduction, (iii) no server adaptivity, and combinations, to justify design choices and causal attribution.

- Resolve clarity/consistency issues. Fix notation overload and inconsistencies (e.g., reused symbols); Clarify terminlogy around “PP” (partial participation vs any privacy-related meaning); Improve readability with a notation table summarizing local/global variables and indexing.

*Strengthening*

- Improve positioning and comparisons to closely related FL optimization methods.
Strengthen the related-work discussion and, where feasible, add comparisons or discussion against standard non-IID/drift and server-optimizer baselines (e.g., FedOpt family, drift-correction methods).

- Reproducibility upgrades.
Provide implementation details, hyperparameter sensitivity studies, and ideally public code (or at least a detailed reproducibility checklist).

---

### Review · Reviewer_76FL · 2025-12-31

**Summary Of Contributions:**

The paper proposes adaptive single-loop federated bilevel optimization algorithms (ASFBO and a variance-reduced variant LA-ASFBO) that address data heterogeneity and system heterogeneity, via reweighted aggregation, momentum updates, and adaptive server-side step sizes. It provides rigorous convergence guarantees with communication-efficient rates under standard bilevel assumptions and demonstrates empirical improvements over existing methods. Key strengths include sound theory and explicit handling of heterogeneous local steps, while the main weakness is limited empirical validation, relying solely on MNIST without ablations or large-scale benchmarks.

**Audience:**

Yes

**Audience Explanation:**

Researchers would be interested in this paper’s treatment of system heterogeneity, particularly the explicit handling of heterogeneous local computation. The single-loop framework and rigorous analysis are relevant to the TMLR audience.

**Broader Impact Concerns:**

I do not identify any major ethical concerns for this work. The paper focuses on optimization algorithms for federated learning and does not introduce new data collection practices, deployment scenarios, or explicit social decision-making applications.

**Claims And Evidence:**

No

**Claims Explanation:**

Theoretical claims are well supported. Empirical claims are only partially supported due to limited experimentation.

1. Robustness to system heterogeneity - Experiments in Section 6.2 with varying local iterations do show ASFBO outperforming FedNest and LFedNest in these settings. However, without ablation studies isolating the re-weighted aggregation from the momentum/STORM components, it is difficult to confirm that the robustness comes specifically from the theoretical heterogeneity handling rather than just the inclusion of variance reduction.

2. Superiority of the proposed algorithm - The authors provide convergence plots on the MNIST dataset for a hyper-representation learning task. Relying solely on MNIST is insufficient to support a broad claim of superiority, as MNIST does not sufficiently stress optimization instability and scalability issues common in bilevel learning with deep networks.

3. Adaptive learning - Theoretically supported, but no experiments isolate adaptivity from fixed learning rates.

4. Personalized privacy protection - No privacy analysis, threat model, or experiments are provided. This claim should be removed or clearly qualified.

5. Computational cost clarification - The method requires maintaining multiple auxiliary variables and computing mixed second-order derivatives (Hessian-vector products). While the authors claim efficiency, there is no explicit analysis or comparison of the wall-clock time or memory footprint per client compared to baselines like FedNest or simple FedAvg-based heuristics.

**Requested Changes:**

1. Add ablation studies that isolate the effect of the re-weighted aggregation mechanism under heterogeneous local iteration counts, e.g., comparing LA-ASFBO with re-weighting vs. without re-weighting, while keeping STORM/momentum fixed. (critical)

2. Include experiments on at least one more challenging and standard federated benchmark, such as CIFAR-10, FEMNIST, Fashion-MNIST, with a CNN. (critical)

3. Provide an ablation comparing AdaGrad-Norm vs. fixed learning rates, even on the existing MNIST setup. (critical)

4. Either remove references to “personalized privacy protection” or clearly qualify the claim as speculative and outside the scope of the current work. (critical)

5. Add a brief discussion or table comparing ASFBO with FedNest and/or FBO-AggITD in terms of per-round gradient/HVP evaluations, communication payload size, client-side memory requirements of auxiliary variables. (recommended)

6. Include a small sensitivity study for one or two key hyperparameters, e.g., momentum coefficient or local steps. (optional)

---

### Review · Reviewer_19UC · 2026-01-02

**Summary Of Contributions:**

This paper addresses the challenges of data and system heterogeneity in Federated Bi-level Optimization (FBO) by proposing a Communication-Efficient Adaptive Single-loop Federated Bi-level Optimization algorithm (ASFBO). ASFBO replaces nested sub-loops with a single-loop architecture to reduce communication and computational costs and employs adaptive learning rates for the upper-level variables to speed convergence. A locally accelerated version, LA-ASFBO, is also introduced, which incorporates momentum-based variance reduction to mitigate hypergradient estimation bias. The theoretical analysis shows that both algorithms achieve convergence to an ϵ-stationary point with O(ϵ−2) sample complexity and O(ϵ −1) communication complexity under non-convex upper-level and strongly convex lower-level settings. Experimental results on federated hyper-representation learning tasks demonstrate superior performance over existing methods.

Strength:
- The proposed algorithms, ASFBO and LA-ASFBO, adopt a fully single-loop architecture, replacing complex nested sub-iterations to significantly reduce communication frequency and computational costs, which is crucial for real-world Federated Learning  settings.
- The algorithms achieve O(ϵ−1) communication complexity and O(P−1ϵ−2) sample complexity to reach an ϵ-stationary point, which are notability efficient.

Weakness:
- The notations are confusing in method part. For example, in problem formulation, x and y are used as variables, but in section 4.1, x, y and z are introduced as global variables, the relation between x,y in sec 3.1 and that in sec 4.1 is not clear. The physical meanings of these variables are also not clearly explained.
- The experiment part is too simple. As a method focus on optimization algorithm, it is necessary to present time complexity of the proposed method compared with baselines.

**Audience:**

Yes

**Audience Explanation:**

The audiences that work on Federated learning and bi-level optimization will be interested in the findings of this paper.

**Broader Impact Concerns:**

There are not concerns that that would require adding a Broader Impact Statement.

**Claims And Evidence:**

Yes

**Claims Explanation:**

Generally, the claims and contributions in the paper is supported by accurate, convincing and clear evidence.

**Requested Changes:**

A notation table to make complex notations clearer in the paper.
Experiments on time cost to verify the effectiveness of the proposed algorithm.

---

### Decision · Action_Editor_otGS · 2026-02-23

**Recommendation:** Reject

**Additional Comments:**

Following the agreement among all reviewers, the final recommendation for this work is Reject, but I strongly encourage the authors to submit a major revision as a new submission once the claims are fully supported. A resubmission (per the feedback received by the reviewers) should:

- Substantially strengthen the experimental evaluation beyond MNIST/MLP using additional standard federated benchmarks and more realistic models, and assess robustness across heterogeneous settings.

- Include clear component-wise ablation studies to isolate the contributions of the main proposed mechanisms (e.g., reweighted aggregation, variance reduction/momentum, and server-side adaptivity), so the source of improvements/robustness can be convincingly attributed.

- Provide transparent computation–communication cost accounting consistent with any “communication efficiency” claims, reporting not only communication rounds but also communication volume (e.g., bytes) and local computation overhead (and ideally runtime when appropriate).

- Tighten and clarify the theoretical presentation in the main text so that the key steps, assumptions, and how approximation/tracking errors are handled are fully rigorous and easy to verify (with the appendix serving as supporting detail rather than the primary source of clarity).

**Audience:**

Yes

**Audience Explanation:**

The topic of federated bilevel optimization under both data and system heterogeneity is timely and relevant to the TMLR audience, and the overall algorithmic direction (single-loop hypergradient estimation with mechanisms aimed at heterogeneity) is of clear interest. With stronger empirical validation and clearer cost reporting, the work could become a valuable contribution for researchers working on federated optimization and bilevel methods.

**Claims And Evidence:**

No

**Claims Explanation:**

Following the reviewers' final recommendations (leaning toward rejection), the authors' rebuttal, and my own evaluation of this submission, I believe the paper proposes promising ideas and algorithms and includes theoretical analysis. However, the submission’s strongest claims (notably around superiority under heterogeneity, communication efficiency, and computational costs) are not convincingly supported by the current evidence.

As the reviewers pointed out, the empirical evaluation is too narrow (largely MNIST/MLP), lacks component-wise ablations to attribute gains to reweighting vs. variance reduction vs. server-side adaptivity, and does not provide a transparent computation–communication cost accounting (e.g., rounds/bytes alongside local compute). In addition, key theoretical steps (e.g., how tracking/approximation errors are bounded and absorbed in the Lyapunov analysis) need clearer, more transparent exposition for readers to verify the rigor in the main text.

**Resubmission Of Major Revision:**

The authors may consider submitting a major revision at a later time.